# Reliably Detecting Model Failures in Deployment Without Labels

Viet Nguyen[1,3,†], Changjian Shui[2,†], Vijay Giri[4],
Siddarth Arya[1,3], Amol Verma[1,5], Fahad Razak[1,5], Rahul G. Krishnan[1,3]

[1]University of Toronto [2]University of Ottawa [3]Vector Institute
[4]University of Pennsylvania [5]Unity Health Toronto

## Abstract

The distribution of data changes over time; models operating in dynamic environments need retraining. But knowing when to retrain, without access to labels, is an open challenge since some, but not all shifts degrade model performance. This paper formalizes and addresses the problem of post-deployment deterioration (PDD) monitoring. We propose D3M, a practical and efficient monitoring algorithm based on the disagreement of predictive models, achieving low false positive rates under non-deteriorating shifts and provides sample complexity bounds for high true positive rates under deteriorating shifts. Empirical results on both standard benchmark and a real-world large-scale internal medicine dataset demonstrate the effectiveness of the framework and highlight its viability as an alert mechanism for high-stakes machine learning pipelines.

## 1 Introduction

Performance guarantees of conventional machine learning (ML) models hinge on the belief that the distribution of data with which these models train is identical to the distribution on which they are deployed [1, 2, 3]. In many real-world scenarios such as healthcare, however, this assumption fails due to distribution shift during model deployment. Benchmarks such as WILDS [4] and WILD-Time [5] have encouraged machine learning researchers to study and better understand how data shifts influence predictive systems. Yet, the number of tools at a practitioner's disposal for building predictive models far exceeds those to monitor model failures. There is a need to create *guardrails* that *self-detect* and *alert* end-users to critical changes in the model when its performance drops below acceptable thresholds [6, 7].

We define post-deployment deterioration (PDD) as the scenario where a trained ML model underperforms on a distributionally shifted deployment query with respect to its validation performance. PDD presents a distinct set of systemic challenges stemming from considerations over the feasibility of deployment in real-world ML pipelines. Predominant is the scarcity of labels during deployment: for many downstream tasks such as in healthcare, labels are expensive to obtain [8] or require human intervention [9]. Due to deployed models predicting events temporally extended in the future [10, 11], labels might even be unavailable. Another is the robustness of the monitoring system: it should flag critical changes in model deterioration early, using few samples, while remaining resilient to non-deteriorating changes to minimize unnecessary interruptions of service among other practical considerations.

To address these challenges, we conceive a set of desiderata for any algorithm monitoring PDD, targeting their practicality and effectiveness as plug-ins to ML pipelines. To address the scarcity of

---

[†]Equal contribution, correspondence to Viet Nguyen: `viet@cs.toronto.edu`

39th Conference on Neural Information Processing Systems (NeurIPS 2025).

labels, PDD monitoring algorithms should operate on unlabeled data from the test distribution to ascertain potential deterioration of the deployed model. Further, PDD monitoring algorithms should not depend on training data during deployment, as continuous (even indefinite) access to sensitive or personally identifiable training data might violate certain regulations protecting the privacy of data subjects [12]. An algorithm satisfying this desideratum is scalable, as it only audits a model's input stream during monitoring with minimum data storage and regulatory concerns. Finally, PDD monitoring mechanisms should be robust to flagging non-deteriorating changes and effective even when samples from the deployment distribution are scarce.

When it comes to designing monitoring protocols that satisfy the above desiderata, recent related works only partially attend to individual desiderata. The literature on distribution shifts while achieving strong empirical performance on unlabeled deployment data [13, 14, 15], are not robust to false positives when the distribution shift is non-deteriorating. The model disagreement framework [16, 17, 18, 19, 20] emerges as a competitive setup for monitoring with downstream performance considerations via the tracking of disagreement statistics, while foregoing explicit distribution shift computations. However, shift-based and disagreement-based monitoring methods all depend on the *presence of training data post-deployment*, and do not provide any guarantees on robustness against false positives in the monitoring of non-deteriorating shifts.

In this work, we answer all desiderata for PDD monitoring via the disagreement framework by proposing **D**isagreement-**D**riven **D**eterioration **M**onitoring **(D3M)**. Our contributions are as follows:

**Answering desiderata.** D3M is a novel algorithm operating in the label-free deployment setting (1), requiring no training data during monitoring (2), and is provably robust in flagging deteriorating shifts as well as resilient to flagging non-deteriorating shifts (3). A comparison of the satisfaction of the PDD desiderata of our method with other related work in the literature is in Table 1.

**Practical and scalable.** D3M is model agnostic so long as the base model's feature extractor can be optimized via gradient descent. This flexibility allows D3M to monitor various modalities of high-dimensional data. Unlike previous work, D3M avoids the retraining or finetuning of the base model via posterior sampling, crucial for the efficient monitoring of large models. Finally, owing to the decoupling of the algorithmic protocol into two distinct stages, D3M is *efficiently scalable in the size of the training dataset*, a critical consideration on the feasibility of its application onto current ML pipelines that is not enjoyed by standard baselines.

**Empirical validation.** We showcase experimental results on various shift scenarios in the UCI Heart Disease dataset [21], CIFAR-10/10.1 [2], Camelyon17 (WILDS) [4], and the GEneral Medicine INpatient Initiative (GEMINI) dataset [22, 23] to demonstrate its effectiveness in monitoring models of various modalities. Our method effectively detects deployment-time deterioration with low false positive rates (FPR) when shifts are non deteriorating, and achieves competitive true positive rates (TPR) when shifts are deteriorating compared to standard baselines. In particular, we discuss how results on the internal medicine dataset suggest D3M to be well-suited for integration into real-world clinical monitoring pipelines.

**Provable algorithm.** Under certain assumptions about the underlying distribution changes, D3M provably monitors model deterioration when a deteriorating shift is present. In the presence of non-deteriorating shifts, D3M provably resists detection, thereby achieving low false positive rates.

**Table 1:** Comparisons between related work. *Training data-free*: whether post-deployment monitoring requires training data; *Deteriorating*: whether the method provably monitors the deteriorating shift; *Non-deteriorating*: whether the method is provably robust in the non-deteriorating shift; *Disagreement*: whether the method is based on the disagreement framework.

| | Training data-free | Deteriorating | Non-deteriorating | Disagreement |
|---|---|---|---|---|
| Yu and Aizawa, 2019 [16] | ✓ | ✗ | ✗ | ✓ |
| Liu et. al., 2020 [13] | ✗ | ✗ | ✗ | ✗ |
| Jiang et. al., 2021 [18] | ✗ | ✗ | ✗ | ✓ |
| Zhao et. al., 2022 [14] | ✗ | ✗ | ✗ | ✗ |
| Rosenfeld and Garg, 2023 [20] | ✗ | ✓ | ✗ | ✓ |
| Ginsberg et. al., 2023 [19] | ✗ | ✓ | ✗ | ✓ |
| **D3M (ours)** | ✓ | ✓ | ✓ | ✓ |

## 2 Background and Algorithm

### 2.1 Problem setup

Assume a base model $f$ is supervisedly trained to classify inputs $x \in \mathcal{X}$ into finite discrete classes $\mathcal{Y} = \{1, \ldots, C\}$ from training examples $\mathcal{D}^n = \{x_i, y_i\}_{i=1:n}$ where tuples $(x_i, y_i)_{i=1:n} \sim \boldsymbol{P}^n$ for all $i \in [n]$. For a joint distribution $\boldsymbol{P}$ over $\mathcal{X} \times \mathcal{Y}$, let $\boldsymbol{P}_x$ denote its marginal distribution over $\mathcal{X}$. We are interested in designing a mechanism such that upon seeing a collection of unlabeled inputs $\{x'_i\}_{i=1:m}$ sampled from a deployment distribution $\boldsymbol{Q}_x$, the mechanism flags model deterioration if $\boldsymbol{Q}_x \neq \boldsymbol{P}_x$ **and** $f$ underperforms on $\boldsymbol{Q}_x$ without being able to observe labels for $\boldsymbol{Q}_x$. On the other hand, if $\boldsymbol{Q}_x \neq \boldsymbol{P}_x$ while $f$ remains performant on $\boldsymbol{Q}_x$, the mechanism should resist flagging. Achieving so ensures that our mechanism only flags deployment-time changes that are truly deteriorating.

*How can we monitor ML models for deployment deterioration due to distribution shift without indiscriminately flagging any changes in the data?*

We require a computable quantity $\phi$, independent of labels, whose value statistically differs in-distribution (ID) and out-of-distribution (OOD) if and only if model deterioration occurs. Monitoring, then, regresses to recording baseline values for $\phi$ evaluated on ID held-out validation samples. Then, upon collecting unsupervised deployment samples from an unknown distribution, the monitoring mechanism computes $\tilde{\phi}$ and compares it to the recorded baseline values, and finally outputs a verdict.

Leveraging insights from [16, 24, 19], the framework of model disagreement possesses this property under certain assumptions about the underlying distribution change (see Appendix A). We say that two models $h_1$ and $h_2$ disagree on an input $x \in \mathcal{X}$ if $h_1(x) \neq h_2(x)$. In particular, models exhibit greater predictive disagreement on unsupervised samples that lead to model deterioration, compared to in-distribution (ID) samples. This is observed through the increased entropy-based discrepancy between classification heads in [16], or maximum disagreement between models in the same ensemble in [24] and [19], as signal for detecting deployment deterioration. Maximizing classification head discrepancy for OOD detection [16] is efficient for monitoring at deployment time, requiring only one forward pass to compute a verdict. However, this trades off classification accuracy as this training procedure alters the original trained decision boundaries. On the other hand, computing model disagreement between ensembles [19] requires finetuning a potentially large network to collect ID and deployment-time disagreement statistics $\phi$. In addition, ID training data is required at deployment, further limiting the scalability of such framework.

### 2.2 Overview of D3M

To avoid needing the original training set at deployment, we replace finetuning with a Bayesian approach that models a posterior predictive distribution (PPD) over logits. This yields a distribution over decision boundaries that remains faithful to ID behavior. By comparing samples from the PPD to the mean prediction, we approximate maximum disagreement without retraining or access to training data. As the PPD is usually intractable, we instead model it with a variational distribution, easily optimizable using standard methods.

Sampling disagreement statistics $\phi$ in this way yields a reference distribution $\Phi$ of ID maximum disagreement rates. At deployment, we compute the same statistic $\tilde{\phi}$ and flag model deterioration when $\tilde{\phi}$ exceeds a high quantile of $\Phi$. This enables unsupervised, training-free monitoring. Our method—**D**isagreement-**D**riven **D**eterioration **M**onitoring (**D3M**)—follows three key steps: **Train**, **Calibrate**, and **Deploy**.

**1. (Train) Base model training.** Firstly, a neural feature extractor $\text{FE}_\theta : \mathcal{X} \to \mathbb{R}^d$ coupled with a Variational Bayesian Last Layers [25] $\text{VBLL}_\theta : \mathbb{R}^d \to \mathcal{P}(\mathbb{R}^C)$ parametrized by $\theta \in \Theta$, are trained on **supervised ID** data $\mathcal{D}^n \sim \boldsymbol{P}^n$ to output posteriors over logits corresponding to $C$ classes. For an input $x \in \mathcal{X}$ and its ground-truth label $y \in \mathcal{Y}$:

$$\psi = \text{FE}_\theta(x)$$
$$q_\theta(\cdot|x) = \text{VBLL}_\theta(\psi)$$

We define our **base model** as the variational **posterior predictive distribution (PPD)** given by integrating with respect to $q_\theta$:

$$\mathbb{P}(y|x, \mathcal{D}^n) = \mathbb{E}_{z \sim q_\theta(\cdot|x)} [\text{softmax}(z)_y]$$

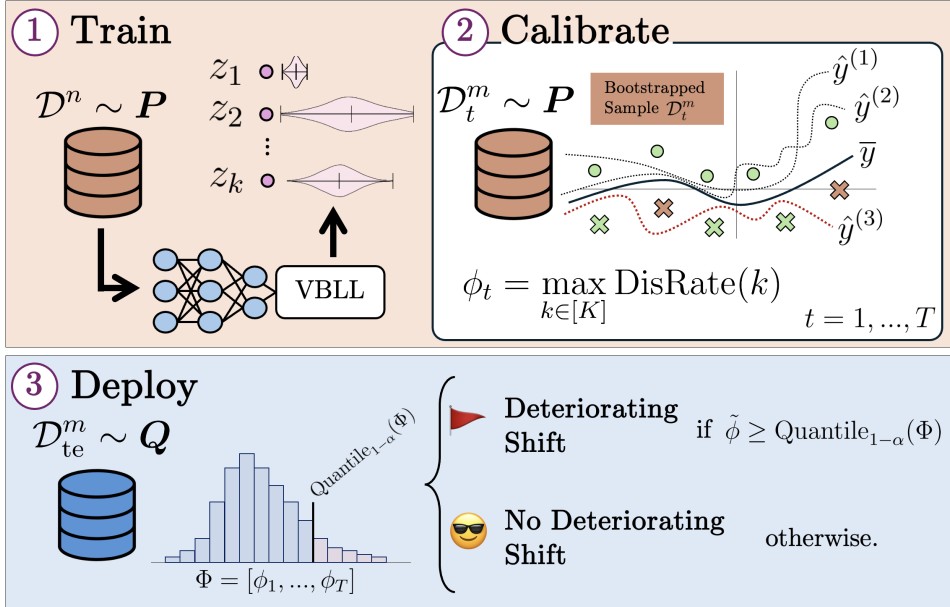

**Figure 1: Overview of D3M. (1) Train:** a feature extractor (FE) and a Variational Bayesian Last Layer (VBLL) are trained to model a posterior predictive distribution (PPD) over class logits. **(2) Calibrate:** disagreement statistics are computed by bootstrapping held-out ID datasets, sampling from the learned posteriors, and comparing sampled predictions to the base model's outputs to collect a set of maximum disagreement rates $\Phi$. For illustrative purposes, agreements and disagreements between $\hat{y}^{(3)}$ and $\bar{y}$ are colored green and orange, respectively. **(3) Deploy:** at deployment, D3M monitors the model on incoming unlabeled data by computing the maximum disagreement rate $\tilde{\phi}$ and flags deteriorating shift if $\tilde{\phi} \geq \text{Quantile}_{1-\alpha}(\Phi)$.

where predictions are assigned by computing the argmax of the PPD. Classification training is performed by maximizing the ELBO with a standard Gaussian prior $p(z)$:

$$\mathcal{L}_{\text{ELBO}}(\theta; x, y) = \mathbb{E}_{z \sim q_\theta(\cdot|x)}[\log \text{softmax}(z)_y] - \text{KL}[q_\theta(z|x) \,\|\, p(z)]$$

**2. (Calibrate) Training of ID max disagreement rates with respect to the base model.** We are to gather disagreement statistics $\phi$ into $\Phi$ computed on ID samples for deployment time comparison. For rounds $t \in [T]$, on a held-out ID collection, we bootstrap an **unsupervised ID** dataset $\mathcal{D}_t^m = \{x_i\}_{i=1}^m \sim \boldsymbol{P}^m$ and acquire posteriors over logits $(q_\theta(\cdot|x_1), q_\theta(\cdot|x_2), \ldots, q_\theta(\cdot|x_m))$. Instead, pseudolabels $\bar{y}_i$ are assigned by the base model using the mean predictive distribution:

$$\bar{y}_i = \underset{y=1,\ldots,C}{\text{argmax}} \, \mathbb{E}_{z_i \sim q_\theta(\cdot|x_i)}[\text{softmax}(z_i)_y], \quad \forall x_i \in \mathcal{D}_t^m$$

We draw $K$ samples from the variational posteriors $q_\theta(\cdot|x_i)$ **in parallel** using vectorized sampling, apply a temperature-scaled softmax, and sample class labels from the resulting categorical distribution:

$$z_i^{(k)} \sim q_\theta(\cdot|x_i) \implies p_i^{(k)} := \text{softmax}\left(\frac{z_i^{(k)}}{\tau}\right), \quad \forall i \in [m], k \in [K]$$

$$\hat{y}_i^{(k)} \sim \text{Categorical}\left(p_i^{(k)}\right)$$

For each posterior sample $k \in [K]$ and their corresponding predictions $(\hat{y}_1^{(k)}, \ldots, \hat{y}_m^{(k)})$, the disagreement rate of $k$ with respect to the base model predictions $(\bar{y}_1, \ldots, \bar{y}_m)$ is calculated as:

$$\text{DisRate}(k) := \frac{1}{m} \sum_{i=1}^m \mathbb{1}\left\{\hat{y}_i^{(k)} \neq \bar{y}_i\right\}, \quad \forall k \in [K]$$

Finally, for the bootstrapped dataset $\mathcal{D}_t^m$, the maximum disagreement rate is given by:

$$\phi_t := \max_{k \in [K]} \text{DisRate}(k)$$

After $T$ rounds, we store our collection of ID maximum disagreement rates $\Phi := \{\phi_t : t \in [T]\}$.

**3. (Deploy) Deployment monitoring of the base model.** Our base model is now ready to be deployed in a test environment and outputs predictions for an unsupervised input stream. To monitor it for deployment deterioration, we periodically gather inputs into a *unsupervised* deployment dataset $\mathcal{D}_{te}^m \sim \boldsymbol{Q}^m$ and compute its maximum disagreement rate $\tilde{\phi}$ as previous. D3M outputs 1 if $\tilde{\phi} \geq \text{Quantile}_{1-\alpha}(\Phi)$ else 0 for a desired significance level $\alpha$.

The entire protocol is summarized in Figure 1. Assume that the deployment distribution is the same as the training distribution. In this case, we expect that $\tilde{\phi} > \text{Quantile}_{1-a}(\Phi)$ with probability $\alpha$, thus having immediate control over the false positive rate (FPR) of D3M. When there is, however, distribution shift between training and deployment distributions, under certain assumptions about the underlying shift and ground truth labeling, we show that D3M **provably** flags deteriorating changes, i.e. a high true positive rate (TPR) is achieved, while being resistant to flagging non-deteriorating changes, i.e. changes in the input distribution that do not result in the base model underperforming. We defer the presentation and discussion of our theoretical analysis to Appendix A.

## 2.3  Algorithmic insights

**D3M is free from training data.** D3M is designed to monitor deployment-time input streams without requiring the training data of its base model. This is an important advantage shared by [16], however not enjoyed by other baselines in Table 1, whose computation of $\phi$ statistics require maintaining agreement on training data. When the neural feature extractor FE is millions or billions of parameters, trained with a comparably large dataset, it becomes infeasible to store the training data within edge compute nodes in order to run the monitoring mechanism.

**No tradeoff with prediction accuracy.** Furthermore, D3M does not trade-off prediction accuracy as the mean prediction model from the PPD is not modified during the construction of $\Phi$ nor its deployment (Appendix C.1). Therefore, ID generalization theories remain applicable [26, 27, 28, 29, 30, 31]. In addition, recording a collection of ID maximum disagreement rates reinforces the robustness of the method. While single instances of maximum disagreement rate are subject to noise in the choice of the held-out validation set, the sampling of posterior logits $z_i^{(k)}$, and the number of drawn samples $K$, a collection $\Phi$ of maximum disagreement rates coupled with a quantile test softens the effect of noise by leveraging large sample statistics.

**Less diverse samples compared to its fully Bayesian counterpart.** Harrison et. al., 2024 [25] suggests that one may apply a variational Bayesian treatment to the last layer only, saving on computational costs of forward and backward propagating through a Bayesian neural network, and enjoy largely the same uncertainty estimation and OOD detection performance. However, the usage of VBLL in D3M is unorthodox with respect to the original work: by sampling posterior weights from a Bayesian model, we are hoping to land on a set of weights such that the resulting model strongly disagree with the base model. VBLL allows more efficient sampling compared to to a fully Bayesian network, though lacking diversity in the sampled predictions, as the variability comes from the last layer only. This results in $K$ sampled logits mostly agreeing with the mean prediction. To improve the diversity of sampling, we employ (1) temperature scaling and (2) the sampling of predictions from the categorical distribution for the computation of maximum disagreement rather than argmax labeling. We defer the study of this tradeoff between fully Bayesian and VBLL to Appendix C.5.

## 2.4  Implementing D3M

The composition $\text{VBLL}_\theta \circ \text{FE}_\theta : \mathcal{X} \to \mathcal{P}(\mathbb{R}^C)$ is readily trained end-to-end on ID data of size $n$ by maximizing the ELBO. For rounds $t \in [T]$, on a large held-out ID validation set, we sample datasets $\mathcal{D}_t^m$ of size $m \ll n$ with replacement for calibration. $\mathcal{D}_t^m$ is processed in a single forward pass, producing $m$ independent $C$-dimensional Gaussian posteriors:

$$q_\theta(z_i|x_i) = \mathcal{N}(z_i|\mu_\theta(x_i), \text{diag}(\sigma_\theta^2(x_i))), \quad \text{for } i = 1, \ldots, m$$

For each posterior, we draw $2K$ samples, $K$ for our Monte-Carlo estimation of the mean model and $K$ for maximum disagreement computations. We keep the sampling temperature $\tau$, the size of bootstrapped datasets $m$, and the number of posterior samples $K$ as tunable hyperparameters. Importantly, these should be the exact same for the computation of $\Phi$ and the subsequent deployment-

time monitoring test, where the exact same computational procedure is employed on a deployment sample $\mathcal{D}_{\mathbf{te}}^m$. A detailed implementation can be found here: https://github.com/teivng/d3m.

## 2.5 Theoretical foundation and practical approximation

Indeed, D3M monitors ML models for deteriorating shifts. As formulated, D3M is an approximation of Idealized D3M (Algorithms 1 and 2 in Appendix A). Our theory formalizes model deterioration under some conditions into a set of inequalities (Definition 2) where Idealized D3M is designed to track their satisfiability. However, this idealized version requires an oracle that can exactly solve the optimization problem of finding the hypothesis in $\mathcal{H}_p$ (well-performing models on the training distribution, Appendix A) that maximally disagrees with the base model. In practice, this search is computationally intractable for large function classes. D3M addresses this by replacing the intractable optimization with the described sampling procedure to approximate hypotheses that likely belong to $\mathcal{H}_p$, trading off exact optimization for computational efficiency while maintaining the core disagreement-based monitoring principle. Our theoretical analysis demonstrates that Idealized D3M provably detects deteriorating shifts while controlling false positive rates (FPR)—guarantees that D3M inherits approximately through its posterior sampling strategy.

In fact, our ablation results empirically show that D3M oversamples hypotheses beyond $\mathcal{H}_p$, deviating from theoretical guarantees. Despite this, D3M demonstrates remarkable robustness in practice, achieving strong empirical performance across diverse datasets. This suggests that the variational posterior sampling, while not perfectly constrained to $\mathcal{H}_p$, still captures sufficient disagreement signal to effectively detect deteriorating shifts. We refer the reader to Appendix A for the theoretical setup and analysis, Appendix C.4 for our results and discussion on our oversampling ablation, and Appendix C.5 for a comparison between an oversampling VBLL and a fully Bayesian neural network.

## 3 Experiments

In all experiments, at minimum, competitive performance in detecting deteriorating shifts (when present) should be achieved. This would validate D3M's effectiveness in alerting end-users of critical changes in deployment. Results on the vision datasets (CIFAR-10/10.1, Camelyon17) show that D3M is effective in monitoring high-dimensional, structure-rich data, in addition to tabular setups. Finally, we explore D3M in monitoring deteriorating and non-deteriorating changes in the GEMINI dataset, a real-world longitudinal electronic health record dataset to study how well our method aligns with clinically meaningful degradation, bringing forth discussions on our mechanism's practical utility for trustworthy, low-intervention deployment in healthcare settings.

## 3.1 Experimental setup

**Datasets.** (1) The **UCI Heart Disease** prediction dataset [21], where each hospital corresponds to a different domain. Here, the distribution shift is due to differences in patient populations and data collection practices across hospitals. (2) **CIFAR-10/10.1** datasets [32, 33] where shift comes from subtle changes in the dataset creation process. By viewing samples from CIFAR-10 as $\boldsymbol{P}$, we test our models' ability to flag deteriorating shift from samples in $\boldsymbol{Q}$ = CIFAR-10.1. (3) The **Camelyon17** dataset from the WILDS benchmark [4, 5] a histopathology image dataset for detecting metastases in lymph node slides, where distribution shift arises from variations in slide staining and image acquisition between hospitals contributing the data. (4) The General Medicine Inpatient Initiative (**GEMINI**) [22, 23] dataset, a comprehensive repository of standardized clinical and administrative data from hospitalizations within general internal medicine. We focus on predicting patient mortality within a 14-day horizon, leveraging static and longitudinal clinical features. This task is deemed essential for facilitating timely clinical interventions, optimizing the allocation of healthcare resources, and ultimately striving to improve patient outcomes [34, 35, 36]. Detailed data descriptions can be found in B.3. Full sweeping details and hyperparameter configurations are reported in B.5.

**Implementation & Baselines.** For tabular data (UCI, GEMINI 14-day mortality), in our implementation, $\mathrm{FE}_\theta$ corresponds to sequences of affine transformations and nonlinearities with skip-connections. For image datasets (CIFAR-10/10.1, Camelyon17), $\mathrm{FE}_\theta$ is either a trained or a finetuned ResNet [37]. In particular, the D3M mechanism is agnostic to the choice of feature extractor, provided that

the architecture reflects appropriate inductive biases for the data modality and permits gradient flow through the extracted features.

To demonstrate that our D3M enjoys low FPR on non deteriorating shifts and high TPR on deteriorating shifts, we compare it against several distribution divergence-based detection methods from the literature: Deep Kernel MMD (MMD-D) [13], H-divergence [14], Black Box Shift Detection (BBSD) [38], Relative Mahalanobis Distance (RMD) [39], Deep Ensembles [40], CTST [41], Domain Classifier (DC) [1], and Detectron [19]. Details can be found in Appendix B.1.

**Evaluations.** For all experiments, the significance level $\alpha$ is fixed to $0.10$. (2) For UCI Heart Disease, CIFAR-10/10.1, and Camelyon17, where there are known post-deployment deterioration, we evaluate the baselines' and D3M's ability to monitor shift for query sizes $\{10, 20, 50\}$ of the deployment distribution. In doing so, we require that good monitors quickly recognize whether the deployment distribution has deteriorating consequences to the model or not. (3) For the GEMINI dataset, we study the detection rates on temporally-split sub-datasets and mixtures of subpopulation splits incurring deteriorating changes and report TPR/FPR where appropriate. For D3M, all TPRs reported are achieved while maintaining an ID FPR below $\alpha$. This is due to the temperature $\tau$: increasing $\tau$ can cause D3M to overfit its reference disagreement distribution $\Phi$ to the held-out validation set, allowing perfect TPR but also maximal FPR.

| | UCI Heart Disease | | | CIFAR 10.1 | | | Camelyon 17 | | |
|---|---|---|---|---|---|---|---|---|---|
| | 10 | 20 | 50 | 10 | 20 | 50 | 10 | 20 | 50 |
| BBSD | .13±.03 | .22±.04 | .46±.05 | .07±.03 | .05±.02 | .12±.03 | .16±.04 | .38±.05 | .87±.03 |
| Rel. Mahalanobis | .11±.03 | .36±.05 | .66±.05 | .05±.02 | .03±.03 | .04±.02 | .16±.04 | .40±.05 | .89±.03 |
| Deep Ensemble | .13±.03 | .32±.05 | .64±.05 | .33±.05 | .52±.05 | .68±.05 | .14±.03 | .26±.04 | .82±.04 |
| CTST | .15±.04 | .51±.05 | **.98±.01** | .03±.02 | .04±.02 | .04±.02 | .11±.03 | .59±.05 | .59±.05 |
| MMD-D | .09±.03 | .12±.03 | .27±.04 | .24±.04 | .10±.03 | .05±.02 | .42±.05 | .62±.05 | .69±.05 |
| H-Div | .15±.04 | .26±.04 | .37±.05 | .02±.01 | .05±.02 | .04±.02 | .03±.02 | .07±.03 | .23±.04 |
| **Detectron** | .24±.04 | **.57±.05** | .82±.04 | .37±.05 | **.54±.05** | **.83±.04** | **.97±.02** | **1.0±.00** | .96±.02 |
| **D3M (Ours)** | **.38±.19** | .25±.28 | .69±.33 | **.40±.10** | .45±.10 | .74±.12 | .89±.20 | .93±.05 | **.99±.02** |

**Table 2:** True positive rates (TPR) comparison across datasets and query sizes. As models do experience deterioration, the higher TPR the better. **Bold** indicates best in column. We report the means and standard deviations of TPRs obtained from 10 independently seeded runs.

## 3.2 Discussions and Analysis

**UCI Heart, CIFAR-10/10.1, and Camelyon17.** Model deterioration is present, thus the **higher** reported TPR the better. Consistently, across the 3 benchmark datasets, we observe that for each query size, D3M enjoys comparable TPR to the top achieving baselines. Results on the Camelyon17 experiments at all query sizes highlight D3M's ability to detect diagnostically-relevant deterioration.

**High-variance reported TPR and limitations.** However, remarkable is the high variance of the TPRs reported on the UCI benchmark. We believe this is a shortcoming of D3M compared to the other baselines in that our results are noisier and less performant. On the one hand, there is merit in high-risk settings to flag critical changes as soon as possible. But on the other hand, more samples may be required in order to truly ascertain the deteriorating nature of a shift. It is up to users of D3M to decide the batching size of collected deployment samples depending on what provides a meaningful signal for further analyses or actionable items. This also reflects the user's tolerance for alert fatigue as noisier estimates on smaller deployment samples may get flagged more liberally. We discuss how selecting this deployment size implicitly allows control of the FPR of the model with additional results of D3M on larger query sizes in the above benchmarks in B.6.

**Comparisons with Detectron. [19]** While D3M and Detectron achieve similar monitoring performance—both successfully flagging all deteriorating shifts in higher query size scenarios (see B.6)—D3M is significantly more scalable and practical for real-world deployment, especially at the edge (Appendix C.2). Unlike Detectron, which requires persistent access to the original training data and gradient-based finetuning during deployment, D3M operates in a truly "source-free" fashion post-training: it needs neither storage nor access to sensitive training samples, nor does it perform computationally expensive or potentially destabilizing finetuning in production. Furthermore, by

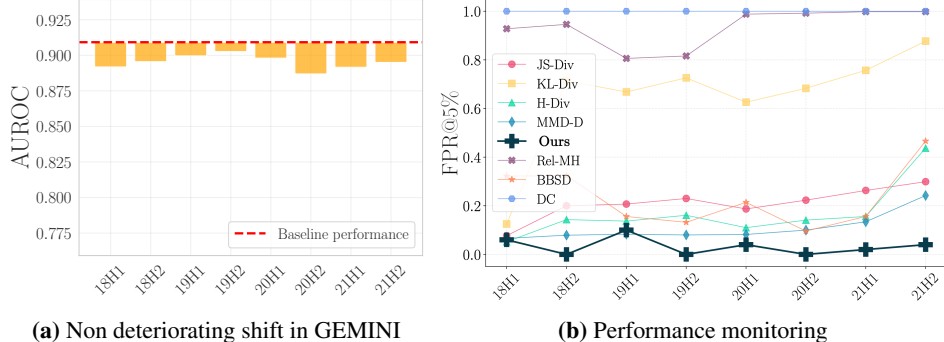

**(a)** Non deteriorating shift in GEMINI

**(b)** Performance monitoring

**Figure 2:** Performances in time evolving shifted test data from GEMINI. **(a)** Performance drop (bar plot) is small, thus a non-deteriorating shift is observed. **(b)** Time evolving shift monitoring. D3M is robust with small False Positive Rate (FPR) at level $\alpha = 0.05$.

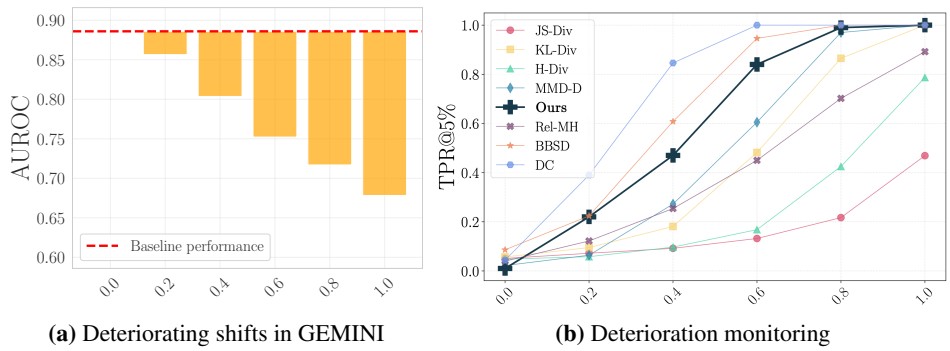

**(a)** Deteriorating shifts in GEMINI

**(b)** Deterioration monitoring

**Figure 3:** Monitoring results on artificially shifted test data from the GEMINI dataset. **(a)** Performance drop (bar plot) is significant when the degree of shift is large ($0.0 \rightarrow 1.0$) **(b)** Results on different monitoring methods, D3M achieves competitive TPRs at level $\alpha = 0.05$.

relying solely on forward passes and simple statistics over deployment data, D3M is inherently robust and less susceptible to the risks associated with continual retraining, such as accidental data contamination or privacy leakage.

**GEMINI Evaluations.** Notably, the GEMINI dataset possesses an *inherent temporal covariate shift*, as patient demographics, laboratory values, and treatment patterns evolve over time, particularly during the COVID-19 period. While we expect distribution shift to occur, whether it represents performance deterioration or benign drift remains unknown a priori, mirroring real-world deployment uncertainty. This makes GEMINI the *closest available approximation to prospective clinical deployment*, where D3M selectively flags shifts those that provably deteriorate performance while ignoring benign drift, ensuring intervention occurs exclusively when warranted. As such, GEMINI serves as a realistic testbed for evaluating both model robustness under distribution shift and D3M's practical utility in settings that closely resemble actual clinical practice.

**GEMINI Temporal Shift Results.** We train the mean model on pre-2018 data and deploy on half-year splits thereafter. As shown in Fig.2(a), there is little to no performance drop over time, indicating a non-deteriorating temporal shift. Accordingly, D3M resists unnecessary alerts, maintaining a low false positive rate compared to baselines (Fig.2(b)).

**GEMINI Age Shift Results.** For deteriorating shifts, we train on adults aged 18–52 and test on various mixtures of age groups. Fig.3(a) shows a clear performance drop as more out-of-distribution samples are included. Here, all methods—including D3M—successfully flag these shifts (Fig.3(b)), with D3M achieving competitive detection across all mixture ratios. This demonstrates that D3M matches the strongest baselines in detecting genuine post-deployment deterioration.

**Clinical integration and flexibility of monitoring.** In real-world clinical settings, the deployment of monitoring systems like D3M must be complemented with clear guidelines for human intervention. One practical integration point is within existing model governance frameworks in hospital information systems, where D3M's alerts could be logged and periodically reviewed by clinical data stewards or model governance committees. *Therein lies the flexibility of detecting versus agnostic adaptation*: monitoring provides a choice on what to do next given a deterioration flag, whether it be retraining, triggering deeper performance audits, shadow deployments, or even adapting. We believe D3M offers a tunable layer of oversight that can flexibly support both real-time and retrospective clinical review processes.

Notably, we view the success of D3M on the real-life D3M as representing a paradigmatic shift in the AI in Healthcare space: current ML models are trained on historical data, validated on held-out test sets, and then deployed directly into clinical practice with minimal ongoing monitoring for performance degradation. While existing approaches relying on heuristic thresholds or requiring expensive retraining cycles have been limitedly employed, our method offers healthcare practitioners a principled mechanism to maintain model reliability in production environments. The validation of the D3M monitoring framework on the GEMINI dataset, which exhibits natural temporal shifts including the COVID-19 disruption, demonstrates the practical utility of this approach in real-world clinical settings where model deterioration can directly impact patient care decisions.

## 4 Related Work

**Adaptation.** Test-time adaptation (TTA) concerns itself more with how one achieves strong deployment performance in spite of a critical change, rather than how one would flag this critical change. To this end, we fundamentally believe that monitoring offers an additional level of flexibility: when a deterioration flag is raised, adaptation is a possible course of action among many others. A few works from this rich body of literature are tangentially related to disagreement-based monitoring. [42, 43, 44] all leverage prediction uncertainty or discrepancy to drive unsupervised domain adaptation. [45], measures and minimizes prediction disagreement at test time, adaptively aligning target-domain features to a domain-invariant feature space, improving performance without requiring access to target domain data during training. In the continual learning literature, [46] also leverages classifier disagreement as a unsupervised proxy of distributional change.

**Performance monitoring of ML models & deteriorating shift.** Evaluating a model's reliability during deployment is crucial for the safety and effectiveness of the machine learning pipeline over time. [47, 48] provided a causal viewpoint wherein the challenge to adapt to diverse scenarios still remain due to a lack of access to the true causal graph. Model disagreement is often used as a monitoring tool for the model generalization [49, 17, 19, 20]. These works align strongly with our method, though the framing is orthogonal to ours as our analysis provides FPR and TPR guarantees whereas they provided sufficient conditions in either ID or other shifts beyond our scope. Several works in the recent literature differentiate shifts in terms of deteriorating or non-deteriorating shifts. [50] studied (deteriorating) shift detection in the continuous monitoring setting using a sequential hypothesis test. Due to the setting being sequential in nature, their method requires true labels from $Q$ immediately after prediction or at the least in a delayed fashion. [51] approached the monitoring problem from a time-continuous anomaly detection perspective, allowing periodic querying of deployment-time ground truths from experts. Other related empirical works along this literature are [52, 53], and the very applied [54].

**Distribution shift detection.** Methods to detect distribution shift arise from different perspectives. In covariate shift detection, [41, 13, 14] treated detection as two-sample tests via classifier, Deep Kernel MMD, and H-divergence. For label shift on the other hand, [38, 55] formulated the problem as a convex optimization problem by solving the label distribution ratio $\alpha = Q(y)/P(y)$. The problem of OOD detection [56, 53] seeks to detect if an individual sample $x$ comes from the training distribution $x \sim P(x)$. Some previous works [57, 58] also adopted the methods in covariate shift detection and generalization by estimating the density ratio for the identification of OOD samples. Whilst these methods detect shifts, they are constrained by their requirement of training data post-deployment and do not consider the extent to which shifts affect model performance.

**Estimating test error with unlabeled data.** Another rich body of research is the estimation of (OOD) test error. This technique and its variants are often inspired by domain adaptation theories [59,

60, 61, 62], seeking guaratess in the form of $\mathrm{err}(f; \boldsymbol{Q}_g) \leq \mathrm{err}(f; \boldsymbol{P}_g) + \Delta(f, \mathcal{H})$, with $\Delta(f, \mathcal{H}) = \sup_{h \in \mathcal{H}} |\mathrm{err}(h; \boldsymbol{P}_f) - \mathrm{err}(h; \boldsymbol{Q}_f)|$. This objective can be alternatively viewed as searching for a critic function $h \in \mathcal{H}$ to maximize a performance gap [20, 18]. One could thus provably estimate the upper bound of the test distribution error. These theories also implicitly assume the availability of training data. Further, they assume that test error should be larger than the training error (granted this is often the case for deteriorating OOD), making them sensitive to non deteriorating shifts as well i.e., a high FPR in detection. Our theoretical analysis addresses this gap.

## 5  Conclusion and Limitations

Two core limitations to D3M are of note when considering deployment in a real-world setting. Firstly, confidence intervals in Table 2 evidences the noisiness of our method compared with other baselines on low query sizes, due to the inherent randomness arising from sampling alternate hypotheses. We do remark, however, that at larger query sizes of 100 and 200, both performance and confidence regions of D3M match those of the strongest baselines (Table 7). We believe that a stronger, more general feature representation helps alleviate this issue as well as enhances performance. Empirically, we observe that by borrowing pretrained ResNet features on ImageNet in the Camelyon17 experiments, D3M achieves high performance at low noise. Future work may explore how performance and stability vary as a function of the quality of representation in the base model. Second, our provided theory only provides provable guarantees when labels are assigned by a ground-truth function. Future work could leverage stronger analytical tools in statistical learning such as PAC-Bayes to derive tighter bounds when labels are noisy, more closely matching real-life deployment scenarios on mislabeled, or even adversarially poisoned labels.

In sum, we study the problem of post-deployment deterioration monitoring of machine learning models in the setting where labels from test distribution are unavailable. We propose a three-stage disagreement-based Bayesian monitoring algorithm, D3M, which monitors and detects deteriorating changes in the deployment dataset while being resilient to flagging non-deteriorating changes. Importantly, our method does not require any training data during deployment monitoring, allowing for efficient out-of-the-box deployment in many machine learning pipelines across various domains. While D3M enjoys increased efficiency, larger confidence intervals demand stricter hyperparameter settings (and thus, bigger initial sweeps) to function effectively. On the theory side, under certain assumptions, statistical guarantees are provided for achieving low FPR in non-deteriorating shifts and high TPR in deteriorating shift.

Empirically, we validate insights from our theory on various synthetic and real-world vision and healthcare datasets evidencing the effective use of D3M. Critically, on the GEMINI dataset—our closest approximation to prospective clinical deployment—D3M demonstrates the precise selectivity required for real-world viability: it resists flagging during periods of non-deteriorating temporal shift while reliably detecting and flagging true performance deterioration. This behavior, validated under realistic conditions with inherent distribution drift, suggests D3M's readiness for practical deployment scenarios where distinguishing actionable degradation from benign evolution is essential. Our work signals a step toward the *robust, scalable, and efficient* deployment of mechanisms to audit and monitor machine learning pipelines in the break of dawn of ubiquitous AI.

## Acknowledgments and Disclosure of Funding

RGK gratefully acknowledges support from the Canada Research Chairs Program (CRC-2022-00049), the Canada CIFAR AI Chairs Program. This research was funded in part by a NFRF Special Call Award (NFRFR-2022-00526). Resources used in preparing this research were provided, in part, by the Province of Ontario, the Government of Canada through CIFAR, and companies sponsoring the Vector Institute (www.vectorinstitute.ai/partnerships/current-partners/). This study was supported by GEMINI, a research program based out of Unity Health Toronto. The development of GEMINI's data platform has been supported by several funding partners, which can be found listed at https://geminimedicine.ca/partners/. All analyses, results, conclusions, and opinions presented in this paper are solely those of the listed authors and are independent from GEMINI's funding sources. No endorsement by GEMINI's funding sources is intended, nor should it be inferred. Amol Verma receives salary support from the Temerty Professorship of AI Research and Education in Medicine at the University of Toronto. Fahad Razak receives salary support from the Canada Research Chairs program.

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

# Appendix

## A  Theoretical Setup and Analysis

### A.1  Overview

In our experiments, we find that the D3M algorithm is effective in monitoring deteriorating changes in multi-class classification. As a proof of concept, we provide sample complexity analyses for the binary classification, where deteriorating shifts are shifts in the covariate distribution of data. We show that an ideal D3M algorithm can achieve strong TPR/FPR guarantees at desirable significance levels under these assumptions. We discuss the extent to which empirical observations from our experiments match with the insights revealed by this analysis.

### A.2  Post-Deployment Deterioration (PDD) and Disagreement-based Post-Deployment Deterioration (D-PDD)

Consider a function class $\mathcal{H}$ of $h : \mathcal{X} \to \mathcal{Y} = \{0, 1\}$ in the binary classification setting. We use $g \in \mathcal{H}$ to denote the ground truth labeling function. We denote the marginal distribution w.r.t $x$ as $\boldsymbol{P}_x(x)$ and the joint distribution with the labeling function in the subscript, that is, for a data distribution $\boldsymbol{P}_x$ over the domain $\mathcal{X}$ and any labeling function $g(x)$, we define the joint distribution as $\boldsymbol{P}_g = \boldsymbol{P}_g(x, y) = \boldsymbol{P}_x(x, g(x))$. For a $f \in \mathcal{H}$, define its generalization error with respect to $\boldsymbol{P}_g$ and its corresponding empirical counterpart with respect to a sample $\mathcal{D}^n = \{(x_i, y_i)\}_{i=1}^n \sim \boldsymbol{P}_g$ as:

$$\mathrm{err}(f; \boldsymbol{P}_g) := \Pr_{(x,y) \sim \boldsymbol{P}_g}[f(x) \neq y], \qquad \widehat{\mathrm{err}}(f; \mathcal{D}^n) := \widehat{\mathrm{err}}(f; \boldsymbol{P}_g) := \frac{1}{n} \sum_{i=1}^n |f(x_i) - y_i|$$

**Training and deployment distribution.**  We denote $\boldsymbol{P}_x$ as the training (marginal) distribution, and $\boldsymbol{Q}_x$ as the deployment distribution. We assume sampled batches of data are I.I.D. with respect to their underlying distribution $\boldsymbol{P}_x$ and/or $\boldsymbol{Q}_x$. We consider that $n$ **labeled samples** from $\boldsymbol{P}_g$ are available before deployment, and $m$ **unlabeled samples** from $\boldsymbol{Q}_x$ are collected during the deployment's input stream.

**Disagreement.**  For any two functions $f$ and $h$ in $\mathcal{H}$, we say that they disagree on any point $x \in \mathcal{X}$ if $f(x) \neq h(x)$. Given the binary classification setting, we can write the disagreement rate of the function $h$ with $f$ on distribution $\boldsymbol{Q}_x$ in terms of error as $\mathrm{err}(h; \boldsymbol{Q}_f)$ or $\mathrm{err}(f; \boldsymbol{Q}_h)$.

Moving forward, $f \in \mathcal{H}$ will be understood to mean our base classifier obtained during training on $\mathcal{D}^n \sim \boldsymbol{P}_g$, while $g$ is the ground truth on $\boldsymbol{P}_x$ and $h \in \mathcal{H}$ denotes auxiliary classifiers in the same hypothesis space, unless otherwise stated. Our goal is to study and monitor the following phenomenon:

**Definition 1** (Post-deployment deterioration, PDD). *Denote $g$ and $g'$ as ground truth labeling functions in the training and deployed distributions $\boldsymbol{P}_x$ and $\boldsymbol{Q}_x$ respectively. We say that PDD has occurred when:*

$$\mathrm{err}(f; \boldsymbol{Q}_{g'}) > \mathrm{err}(f; \boldsymbol{P}_g) \tag{1}$$

Intuitively, Eq. (1) suggests that PDD occurs when a model $f$ experiences higher error during deployment than during its training. Due to the unsupervised nature of the deployment dataset, PDD monitoring is difficult for any arbitrary $g' \neq g$ as we cannot trivially compute empirical errors for the LHS. Though tools from the literature of OOD error estimation may be used, we propose to proxy via a related notion. Def. 2 introduces a new and practical concept—model disagreement-based PDD—equivalent to PDD under specific assumptions.

**Definition 2** (Disagreement based PDD (D-PDD)). *We say that D-PDD has occurred when the following holds for some $\epsilon_f < 1$:*

$$\exists h \in \mathcal{H} \quad s.t. \quad \mathrm{err}(h; \boldsymbol{P}_g) \leq \epsilon_f \ and \ \mathrm{err}(f; \boldsymbol{P}_g) \leq \epsilon_f \ and \ \mathrm{err}(h; \boldsymbol{Q}_f) > \mathrm{err}(h; \boldsymbol{P}_f) \tag{2}$$

D-PDD in Def. 2 is defined as the situation where there exists an auxiliary model $h \in \mathcal{H}$ achieving equally good performance on $\boldsymbol{P}$ (with a small error $\epsilon_f$) but exhibits strong disagreement with $f$ in $\boldsymbol{Q}$. In this case, the distribution $\boldsymbol{Q}$ is further referred to as a **deteriorating shift**. In the following lemma, we demonstrate the conditions for the equivalence of PDD and D-PDD.

**Lemma A.1** (Equivalence condition). *Assume that the ground truth at training and deployment are identical, i.e. $g = g'$, and that $\mathrm{TV}(\boldsymbol{P}, \boldsymbol{Q}) \leq \kappa$, we have that when $\mathrm{err}(f, \boldsymbol{Q}_h) - \mathrm{err}(f, \boldsymbol{P}_h) \geq 2(\kappa + \epsilon)$, i.e. the disagreement gap is large enough, D-PDD and PDD are equivalent.*

*Proof.* To show PDD $\implies$ D-PDD, assume $g = g'$, i.e. identical concepts during training and deployment. Assume our base classifier $f$ is well-trained with $\mathrm{err}(f, \boldsymbol{P}_g) = \epsilon$. We have that

$$\mathrm{err}(f, \boldsymbol{Q}_g) > \mathrm{err}(f, \boldsymbol{P}_g)$$

Let our candidate auxiliary function $h \in \mathcal{H}$ be given by $h = g$. Then, $h$ satisfies all conditions for D-PDD.

We now show that D-PDD $\implies$ PDD. Assume that there is no concept shift, i.e. the ground truth distribution is identical, $g = g'$.

We transport the D-PDD condition $\exists h \in \mathcal{H}$ s.t. $\mathrm{err}(f, \boldsymbol{Q}_h) > \mathrm{err}(f, \boldsymbol{P}_h)$ to the general PDD condition $\mathrm{err}(f, \boldsymbol{Q}_g) > \mathrm{err}(f, \boldsymbol{P}_g)$ by leveraging the proximity of $h$ to $g$ on $\boldsymbol{P}$ and that the total variation between $P$ and $Q$ are constrained by $\kappa$.

We observe that for any $f, g, h \in \mathcal{H}$:

$$|\mathrm{err}(f, \boldsymbol{P}_g) - \mathrm{err}(f, \boldsymbol{P}_h)| < \epsilon$$

Indeed, this is true since:

$$
\begin{aligned}
|\mathrm{err}(f, \boldsymbol{P}_g) - \mathrm{err}(f, \boldsymbol{P}_h)| &= |\boldsymbol{P}(f \neq g) - \boldsymbol{P}(f \neq h)| \\
&= |\mathbb{E}_{\boldsymbol{P}} [\mathbb{1}\{f \neq g\} - \mathbb{1}\{f \neq h\}]| \\
&\leq \mathbb{E}_{\boldsymbol{P}} [|\mathbb{1}\{f \neq g\} - \mathbb{1}\{f \neq h\}|] \\
&\leq \mathbb{E}_{\boldsymbol{P}} [|\mathbb{1}\{g \neq h\}|] \\
&= \boldsymbol{P}(g \neq h) \leq \epsilon
\end{aligned}
$$

where we used Jensen's inequality, and that $|\mathbb{1}\{f \neq g\} + \mathbb{1}\{f \neq h\}| = |\mathbb{1}\{g \neq h\}|$.

Let $\mathrm{TV}(\boldsymbol{P}, \boldsymbol{Q}) \leq \kappa$ for some $\kappa > 0$. We further observe that for any $f, g \in \mathcal{H}$:

$$
\begin{aligned}
|\mathrm{err}(f, \boldsymbol{Q}_g) - \mathrm{err}(f, \boldsymbol{P}_g)| &= |\boldsymbol{Q}(f \neq g) - \boldsymbol{P}(f \neq g)| \\
&\leq \sup_A |\boldsymbol{Q}(A) - \boldsymbol{P}(A)| = \kappa
\end{aligned}
$$

Putting our two observations together yields following decomposition:

$$
\begin{aligned}
|\mathrm{err}(f, \boldsymbol{Q}_h) - \mathrm{err}(f, \boldsymbol{Q}_g)| &\leq |\mathrm{err}(f, \boldsymbol{Q}_h) - \mathrm{err}(f, \boldsymbol{P}_h)| \\
&\quad + |\mathrm{err}(f, \boldsymbol{P}_h) - \mathrm{err}(f, \boldsymbol{P}_g)| + |\mathrm{err}(f, \boldsymbol{P}_g) - \mathrm{err}(f, \boldsymbol{Q}_g)| \\
&\leq 2\kappa + \epsilon
\end{aligned}
$$

For PDD to hold, $\mathrm{err}(f, \boldsymbol{Q}_g)$ needs to be no less than $\mathrm{err}(f, \boldsymbol{P}_h) + \epsilon$ and at most $2\kappa + \epsilon$ less than $\mathrm{err}(f, \boldsymbol{Q}_h)$. Equating yields:

$$\mathrm{err}(f, \boldsymbol{P}_h) + \epsilon \leq \mathrm{err}(f, \boldsymbol{Q}_h) - 2\kappa - \epsilon$$

$$\implies \mathrm{err}(f, \boldsymbol{Q}_h) - \mathrm{err}(f, \boldsymbol{P}_h) \geq 2(\kappa + \epsilon)$$

prescribing the conditions for which D-PDD implies PDD.

$\square$

Thus, if an algorithm monitors D-PDD, then under the assumptions of Lemma A.1, the algorithm also monitors post-deployment deterioration (PDD). In fact, D3M approximately monitors D-PDD. To see this, we show that an ideal version of D3M monitors D-PDD in the above subsection.

## A.3 Idealized D3M and D-PDD Monitoring

Tracking D-PDD in finite samples as formulated requires training data during deployment, which runs counter to the set of desiderata we previously established. To circumvent this, we **decouple** the detection of D-PDD into two **Calibrate** and **Deploy** stages. The **Calibrate** stage finds a subset $\mathcal{H}_p \subset \mathcal{H}$ whose elements satisfy conditions on $\boldsymbol{P}_g$ in Def. 2 as well as approximates $\mathrm{err}(h; \boldsymbol{P}_f)$, while the **Deploy** stage tracks the satisfaction of the last inequality. In this way, information from the training data is compressed into $\mathcal{H}_p$ and the approximation of $\mathrm{err}(h; \boldsymbol{P}_f)$. Meanwhile, the approximation of the disagreement threshold $\mathrm{err}(h, \boldsymbol{P}_f)$ for $h \in \mathcal{H}_p$ can be done via its empirical distribution $\Phi$ computed during the Calibrate phase. We thus present the idealized versions of the Calibrate and Deploy stages of D3M.

---

**Algorithm 1** Idealized D3M: Calibrate

---

**Require:** $\mathcal{D}^n \sim \boldsymbol{P}_g, f, \epsilon, \mathcal{H}$
1: Train a sub hypothesis space $\mathcal{H}_p := \{h \in \mathcal{H}; \; \widehat{\mathrm{err}}(h; \boldsymbol{P}_g) \leq \epsilon\}$
2: $\Phi \leftarrow []$
3: **for** $t \leftarrow 1, 2, \ldots, T$ **do**
4: $\quad \mathcal{D}^m \sim \boldsymbol{P}_f$
5: $\quad h \leftarrow \underset{h \in \mathcal{H}_p}{\mathrm{argmax}} \; \widehat{\mathrm{err}}(h; \mathcal{D}^m)$
6: $\quad$ **append** $\widehat{\mathrm{err}}(h; \mathcal{D}^m)$ to $\Phi$
7: **end for**
8: **return** $\Phi, \mathcal{H}_p$

---

**1. Calibration in $\boldsymbol{P}$.** Given in-distribution training data $\mathcal{D}^n$, a base model $f$ trained on $\mathcal{D}^n$, an error tolerance $\epsilon$, and the hypothesis class $\mathcal{H}$, we formulate the subset of $\mathcal{H}$ achieving the error tolerance, $\mathcal{H}_p = \{h \in \mathcal{H}; \; \mathrm{err}(h; \boldsymbol{P}_g) \leq \epsilon\}$. Then, the disagreement distribution $\Phi$ is trained: for $T$ rounds, $m$ samples pseudo-labeled by $f$ ($\mathcal{D}^m$) is used to train auxiliary models $h$ by maximizing disagreement between $h$ and $f$ under $\mathcal{H}_p$ on $\mathcal{D}^m$ to approximate $\mathrm{dis}_P = \underset{h \in \mathcal{H}_p}{\max} \; \mathrm{err}(h; \boldsymbol{P}_f)$. The empirical disagreement rate achieved by $h$ is appended to $\Phi$. Finally, the pre-training procedure returns $\Phi$ and $\mathcal{H}_p$.

---

**Algorithm 2** Idealized D3M: Deploy

---

**Require:** $\mathcal{H}_p, \Phi, f, \alpha$
1: $\mathcal{D}^m \sim \boldsymbol{Q}_f$
2: $h \leftarrow \underset{h \in \mathcal{H}_p}{\mathrm{argmax}} \; \widehat{\mathrm{err}}(h; \mathcal{D}^m)$
3: **return** $\widehat{\mathrm{err}}(h; \mathcal{D}^m) > (1 - \alpha)$ quantile of $\Phi$

---

**2. Deploy in $\boldsymbol{Q}$.** Given $\Phi$ and $\mathcal{H}_p$, we compute the one-sample approximation of the maximal disagreement with $f$ on $\boldsymbol{Q}$: $\mathrm{dis}_Q = \max_{h \in \mathcal{H}_p} \mathrm{err}(h; \boldsymbol{Q}_f)$. We say D-PDD happens when $\mathrm{dis}_Q$ lies in the top $\alpha$ quantile of $\Phi$.

The idealized algorithms 1 and 2 together track D-PDD in finite samples. Indeed, a deployment on $\boldsymbol{Q}$ is flagged when the last inequality in Def. 2 is detected, while the imposition of the other inequalities are done via formulating the constrained hypothesis space $\mathcal{H}_p$ of hypotheses that already satisfy these inequalities and searching over it. Of note is that at deployment time, the original training set $\mathcal{D}^n$ is not required.

**D3M approximates Idealized D3M.** The primary implementation consideration in the Idealized D3M algorithms is the search over $\mathcal{H}_p$, which for large function classes cannot be done trivially. D3M "Bayesianly" approximates this search by turning the intractable optimization problem into a sampling problem. By drawing from our $(\mathrm{VBLL}_\theta \circ \mathrm{FE}_\theta)$ posterior, we are hoping to sample hypotheses that belong in $\mathcal{H}_p$ with high probability. When viewed this way, the correspondence between D3M and its idealized version above follows. The price of the increased efficiency from avoiding intractable

searches over $\mathcal{H}_p$ is thus the sampling noise from D3M which may return hypotheses beyond the constraints of $\mathcal{H}_p$.

## A.4 Provable Guarantees of Idealized D3M

We present theoretical guarantees for Idealized D3M (Algorithms 1 and 2), which we refer to as D3M for the remainder of this section. Recall that D3M is tracking a sufficient condition of D-PDD. We show that with enough samples, when there is non-deteriorating shift, the algorithm achieves low false positive rates with high probability. Then, we show that with enough samples, when there is deteriorating shift, the algorithm provably succeeds. Finally, we discuss pathological cases where monitoring fails irrespective of sample size.

### A.4.1 Preliminary quantities

**Definition 3** (Deployed classifier error). *This quantifies the generalization error of the deployed base classifier $f$. This is measured on the distribution seen during training $\boldsymbol{P}_g$,*

$$\epsilon_f := \mathrm{err}(f; \boldsymbol{P}_g) \tag{3}$$

Indeed in Def. 2, we want the population error to be at most $\epsilon_f$, which results in the constraint for the empirical error in the optimization problems of Algorithm 1 at most $\epsilon = \epsilon_f - \epsilon_0$, where $\epsilon_0$ is a hyper-parameter to measure the gap between the empirical and population error.

We also define the VC dimensions of the hypothesis space $\mathcal{H}$ and the subset of interest $\mathcal{H}_p$ as:

$$\mathcal{H}_p := \{ h \in \mathcal{H} : \mathrm{err}(h; \boldsymbol{P}_g) \leq \epsilon_f \}, \quad d_p := \mathrm{VC}(\mathcal{H}_p), \quad d := \mathrm{VC}(\mathcal{H})$$

Note that $d_p \leq d$. If the base classifier $f$ is well-trained ($\epsilon_f$ is low), then $d_p$ can be much smaller than $d$ i.e., $d_p \ll d$.

**Definition 4** ($\epsilon_p, \epsilon_q$ maximum error in $\mathcal{H}_p$). *The maximum error in $\mathcal{H}_p$ for both $\boldsymbol{P}$ and $\boldsymbol{Q}$ using pseudo-labels from $f$ is defined as:*

$$\epsilon_p = \max_{h \in \mathcal{H}_p} \mathrm{err}(h; \boldsymbol{P}_f), \quad \epsilon_q = \max_{h \in \mathcal{H}_p} \mathrm{err}(h; \boldsymbol{Q}_f) \tag{4}$$

*Note that empirical quantities of these are also the maximum empirical disagreement rates used in Algo. 1 and Algo. 2. Effectively, the algorithm detects $\epsilon_q - \epsilon_p > 0$ with finite samples.*

**Definition 5** ($\xi$ quantifies D-PDD). *We define $\xi$ to quantify the degree of D-PDD. We adopt Def. 2 and define $\xi$ as*

$$\xi := \max_{h \in \mathcal{H}_p} \{ \mathrm{err}(h; \boldsymbol{Q}_f) - \mathrm{err}(h; \boldsymbol{P}_f) \} \tag{5}$$

*Therefore, D3M detects whether $\xi > 0$. Furthermore, $\xi$ is non-negative since $f \in \mathcal{H}_p$. Hence, in case of non-deteriorating shift, $\xi = 0$.*

Note that $\xi \geq \epsilon_q - \epsilon_p$. It follows that $\epsilon_q - \epsilon_p > 0 \implies \xi > 0$, though the reverse implication is not necessarily true. Therefore Algo. 2, ($\epsilon_q - \epsilon_p > 0$) is detecting a sufficient condition of D-PDD ($\xi > 0$).

Next, we relate the amount of D-PDD, $\xi$, with the amount of distribution shift in the form of TV-distance between $\boldsymbol{P}_x$ and $\boldsymbol{Q}_x$. As seen in the Eq. 5, deterioration depends on the complexity of the function class and $\epsilon_f$ which affects the size of $\mathcal{H}_p$. We capture these factors by introducing a mixture distribution $\boldsymbol{U}$:

$$\boldsymbol{U} = \frac{1}{2} \left( \boldsymbol{P}_f + \boldsymbol{Q}_{1-f} \right) \tag{6}$$

**Definition 6** ($\eta$ error gap between $\mathcal{H}_p$ and Bayes optimal). *For the distribution $\boldsymbol{U}$, the gap in error between the best classifier $h \in \mathcal{H}_p$ in the function class and the Bayes optimal classifier is $\eta$:*

$$\eta := \min_{h \in \mathcal{H}_p} \mathrm{err}(h; \boldsymbol{U}) - \mathrm{err}(f_{bayes}; \boldsymbol{U}) \tag{7}$$

Note that $\eta$ depends on the shift and complexity of the function class. We relate various definitions introduced in this section as follows.

**Proposition A.2** (D-PDD and TV distance). *The relations between $\xi$ (in Def. 5), $\eta$ (in Def. 6), and $\epsilon_p$, $\epsilon_q$ (in Def. 3 and 4) are as follows:*

$$\xi = \text{TV} - 2\eta \geq 0 \tag{8}$$

$$\xi \geq \epsilon_q - \epsilon_p \geq \xi - 2\epsilon_f \tag{9}$$

We denote the total variation distance between $\boldsymbol{P}_x$ and $\boldsymbol{Q}_x$ as TV. Intuitively, D-PDD is defined in such a way that after deployment, if we are uncertain of the performance of $f$, then the shift is deteriorating. In general, for simpler function classes such as linear models, by looking at one region of the domain ($\boldsymbol{P}_x$) it may be possible to be certain about the performance of another region ($\boldsymbol{Q}_x$), this is captured in Eq. 8. For very complex function classes, $\eta$ can be low, hence $\xi > 0$ for most shifts. For simple function classes, $\eta$ can be high, in which case $\xi$ may not be positive and hence a non-deteriorating shift. This thus highlights a trade-off with selecting expressive functions to capture complex patterns in the data.

### A.4.2 D3M algorithm in non-deteriorating shift

D3M aims to monitor and detect D-PDD in finite samples, which can inherently lead to false positives (FPR) when the shift is non-deteriorating. Therefore we set a tolerance factor $\alpha$ in Alg. 2 to account for the test's robustness. We show that for D3M, the FPR of the detection can be close to $\alpha$ for *any* shift in the data distribution. Furthermore, we show that the FPR can also be less than $\alpha$ in some cases. In the case of non deteriorating shift, by Def. 2, the following holds:

$$\forall h \in \mathcal{H}_p: \ \text{err}(h; \boldsymbol{Q}_f) \leq \text{err}(h; \boldsymbol{P}_f) \implies \epsilon_q \leq \epsilon_p \tag{10}$$

Note that D3M intuitively detects whether $\epsilon_q > \epsilon_p$. Since the above equation shows that $\epsilon_q \leq \epsilon_p$, given enough samples, the test will succeed. Recall that $n$ is the number of samples given from $\boldsymbol{P}_g$ and $m$ is the number of samples required from $\boldsymbol{Q}_x$. In the theorem below, the significance level $\alpha$ refers to the desired FPR.

**Theorem A.3.** *For $\gamma \leq \alpha$, when there is no deteriorating shift (no D-PDD) in Eq. 10, for a chosen significance level of $\alpha$, the FPR of D3M is at most $\gamma + (1 - \gamma)\, \mathcal{O}\left(\exp\left(-n\epsilon_0^2 + d\right)\right)$ if*

$$m \in \mathcal{O}\left(\left(\frac{1 - \sqrt{\delta}}{\epsilon_p - \epsilon_q}\right)^2 \left(d_p + \ln\frac{1}{\gamma}\right)\right) \tag{11}$$

*and $\epsilon_p - \epsilon_q > 0$, where $\delta = (d_p + \ln\frac{1}{\alpha})/(d_p + \ln\frac{1}{\gamma})$ and we define $\epsilon_0 \leq \epsilon_f - \widehat{\text{err}}(h; \boldsymbol{P}_f)$.*

In the case of non deteriorating shifts (specifically $\epsilon_p > \epsilon_q$) the FPR may be even less than $\alpha$ given that $m$ and $n$ are sufficiently large. The more samples from $\boldsymbol{Q}_x$ we have, the lesser the FPR in these cases. For any general case, by setting $\gamma = \alpha$ (i.e., $\delta = 1$) in the above theorem, we immediately have:

**Corollary A.4.** *For a chosen significance level $\alpha$, the FPR of D3M (Alg. 2) is no more than $\alpha + (1 - \alpha)\, \mathcal{O}\left(\exp\left(-n\epsilon_0^2 + d\right)\right)$.*

**Practical insights.** The corollary asserts the robustness of D3M against unnecessarily flagging non-deteriorating shifts. Independent of the number of deployment samples $m$, for any given significance level $\alpha$, the FPR is only slightly worse, with the additive term decaying exponentially in the number of training samples. For many practical ML pipelines that by-and-large employ linear and forest models among others of manageable VC-dimension, having these guarantees means that a D3M audit likely won't negatively impact the continuity and quality of the model usage.

### A.4.3 D3M algorithm in deteriorating shift

When deteriorating shift occurs:

$$\exists h \in \mathcal{H}_p: \text{err}(h; \boldsymbol{Q}_f) > \text{err}(h; \boldsymbol{P}_f) \tag{12}$$

However, this does not necessarily imply that $\epsilon_q > \epsilon_p$ which is ultimately the condition monitored by D3M. In the following, we break down the possible scenarios.

**Regime 1. Deteriorating shift and $\epsilon_q > \epsilon_p$.** In this case, Theorem A.5 demonstrates that the D3M algorithm detects deteriorating shift with provable high TPR. Here, the significance level $\alpha$ is understood to be 1 minus the desired TPR.

**Theorem A.5.** *For $\beta > 0$, when deteriorating shift occurs, for a chosen significance level of $\alpha$, the TPR of D3M (Alg. 2) is at least $(1 - \beta)\left(1 - \mathcal{O}\left(\exp\left(-n\epsilon_0^2 + d\right)\right)\right)$ if*

$$m \in \mathcal{O}\left(\left(\frac{1 + \sqrt{\delta}}{\xi - 2\epsilon_f}\right)^2 \left(d_p + \ln\frac{1}{\beta}\right)\right) \tag{13}$$

*and $\epsilon_q - \epsilon_p > 0$, where $\delta = (d_p + \ln\frac{1}{\alpha})/(d_p + \ln\frac{1}{\beta})$, and $\epsilon_0 \leq \epsilon_f - \widehat{\mathrm{err}}(h; \boldsymbol{P}_f)$.*

Notably, $\xi$ in the denominator indicates that as the shift becomes more deteriorating, D3M requires fewer samples $m$ to detect, evidencing its effectiveness. Further, having a high-quality base classifier $f$ with low $\epsilon_f$ is better for detection: this is seen through in Eq. 9 where low $\epsilon_f$ makes the monitoring more faithful at a lower sample complexity $m$. Another remark is that $m$ depends on $d_p$ which can be much less than $d$ with $n$ being dependent on the latter. The test, thus, can work for a $m$ significantly smaller than $n$. The dependency on $n$ is due to the requirement of satisfaction of the first condition in Def. 2. In the constrained optimization problems in Algo. 1, the constraint is satisfied but the population constraint will be satisfied either for a larger $\epsilon_0$ or for sufficiently large $n$ as seen in the above theorem.

**Regime 2. (Possible tradeoff)** -Deteriorating shift but $\epsilon_q \leq \epsilon_p$. In this case, Theorem A.6 demonstrates that either the false negative or false positive rates (FNR, FPR) should be high. The illustration in Fig. 4 exemplifies this failure mode, and how a low $\epsilon_f$ can help alleviate it.

**Theorem A.6.** *When deteriorating shift occurs and $\epsilon_q \leq \epsilon_p$, for a chosen significance level of $\alpha$, the TPR of Alg. 2 is $\mathcal{O}(\alpha)$.*

If $\alpha$ is low, then by Theorem A.6 we have that the FNR is high. On the other hand, if $\alpha$ is high, then by Corollary A.4 we have that the FPR can be very high, thereby trading off the significance level of D3M to reduce FNR but loosening guarantees on FPRs.

### A.4.4 Solutions for FNR/FPR tradeoff

This part provides a possible failure scenario illustrated in Fig. 4. Notably, we will highlight a badly trained base classifier $f$ in the possible failure scenario (Fig. 4 (a)), if $f$ is trained with lower $\epsilon_f$, can move to the scenarios (Fig. 4 (b) and (c)) where the D3M algorithm can succeed.

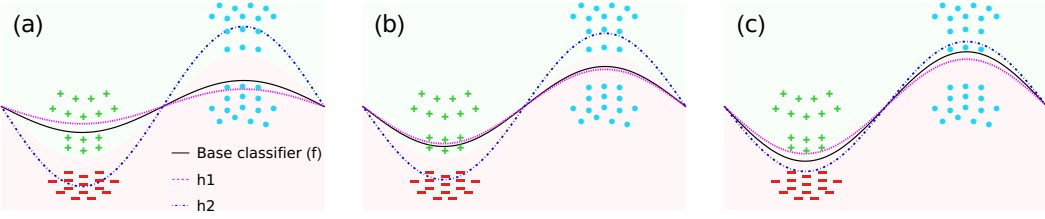

**Figure 4: Illustration of the FNR/FPR tradeoff and its remedy.** The background color indicates the fixed ground truth. Positive and Negative points are from $\boldsymbol{P}_g$ (labeled) and the unlabeled points are from $\boldsymbol{Q}_x$. The solid black curve represents the deployed base classifier $f$. The dotted Pink ($h_1$) and Blue ($h_2$) curves represent the envelope boundary for $\mathcal{H}_p$ i.e., all the functions passing between these two curves are contained in $\mathcal{H}_p$. (a) Failure scenario (i.e, Regime 2) where D3M algorithm fails. (b) No deteriorating shift scenario. (c) Deteriorating shift and the D3M algorithm succeeds. In summery, a decreasing on $\epsilon_f$ could move the failure scenario to the solvable scenarios (a) or (b).

In Fig. 4 (a), if $f$ is not well-trained, we will encounter a failure scenario. The disagreement of $h_1$ with $f$ on $\boldsymbol{Q}_x$ is larger than that of $\boldsymbol{P}_x$, evidencing D-PDD. However, note that $h_2$ can maximize $\epsilon_p$ more than any function (in $\mathcal{H}_p$) can maximize $\epsilon_q$, which implies $\epsilon_p > \epsilon_q$. If $f$ is better trained in Fig. 4 (b), for all functions in $\mathcal{H}_p$ (curves between $h_1$ and $h_2$) disagreement with $f$ on $\boldsymbol{P}_x$ is not less than that of $\boldsymbol{Q}_x$. Hence there is no deteriorating shift and D3M algorithm could provably address this. Alternatively, if $f$ is trained well and is closest to the ground truth Fig. 4 (c) the disagreement of $h_2$ with $f$ on $\boldsymbol{Q}_x$ is more than that of $\boldsymbol{P}_x$. Also, note that $\epsilon_p = 0$ since there is no function that can have any error on $\boldsymbol{P}_f$. However, $h_2$ can be the classifier to get non-zero $\epsilon_q$ which gives $\epsilon_q > \epsilon_p$. Hence (c) recovers the Regime 1 and is solvable.

**Practical implications.** Training base classifiers with strong in-distribution generalization performance helps in reducing the likelihood of falling into **Regime 2**. Then, Theorem A.6 guarantees that with high probability, the desired TPR of D3M can be achieved modulo an exponentially decaying factor in the number of training samples. In this way, D3M is robust in monitoring deteriorating shifts with provable TPR guarantees, satisfying the robustness desiderata for PDD monitoring.

## A.5 Proofs

**Lemma A.7.** *For any $\gamma > 0$, $\mu > \epsilon_q$, we have $\widehat{\mathrm{err}}(h; \boldsymbol{Q}_f) \leq \mu$ for all $h \in \mathcal{H}_p$ with probability at least $1 - \gamma$ if*

$$m \in \mathcal{O}\left(\frac{d_p + \ln\frac{1}{\gamma}}{(\mu - \epsilon_q)^2}\right) \tag{14}$$

*Proof.* We use the generalization bound for agnostic learning in [63].

$$ce^{d_p}e^{-\epsilon^2 m} \geq \Pr_{X,Y \sim \boldsymbol{Q}_{1-f}^m}[\exists h \in \mathcal{H}_p : \mathrm{err}(h; \boldsymbol{Q}_{1-f}) - \widehat{\mathrm{err}}(h; \boldsymbol{Q}_{1-f}) \geq \epsilon] \tag{15}$$

$$= \Pr_{X,Y \sim \boldsymbol{Q}_{1-f}^m}[\exists h \in \mathcal{H}_p : \widehat{\mathrm{err}}(h; \boldsymbol{Q}_f) \geq \mathrm{err}(h; \boldsymbol{Q}_f) + \epsilon] \tag{16}$$

$$\geq \Pr_{X,Y \sim \boldsymbol{Q}_{1-f}^m}[\exists h \in \mathcal{H}_p : \widehat{\mathrm{err}}(h; \boldsymbol{Q}_f) \geq \epsilon_q + \epsilon] \tag{17}$$

Choose $\epsilon = \mu - \epsilon_q$ for any $\mu > \epsilon_q$. Now,

$$ce^{d_p}e^{-\epsilon^2 m} \leq \gamma \tag{18}$$

$$m \in \mathcal{O}\left(\frac{d_p + \ln\frac{1}{\gamma}}{(\mu - \epsilon_q)^2}\right) \tag{19}$$

$\square$

**Lemma A.8.** *For any $h \in \mathcal{H}$, if the $\widehat{\mathrm{err}}(h; \boldsymbol{P}_f) \leq \epsilon_f - \epsilon_0$ then with probability at least $1 - \mathcal{O}\left(\exp\left(-n\epsilon_0^2 + d\right)\right)$, $h$ will be in $\mathcal{H}_p$*

*Proof.* We use the generalization bound for agnostic learning in [63].

$$ce^{d}e^{-\epsilon^2 n} \geq \Pr_{X,Y \sim \boldsymbol{P}_g^n}[\exists h \in \mathcal{H} : \mathrm{err}(h; \boldsymbol{P}_g) - \widehat{\mathrm{err}}(h; \boldsymbol{P}_g) \geq \epsilon] \tag{20}$$

$$= \Pr_{X,Y \sim \boldsymbol{P}_g^n}[\exists h \in \mathcal{H} : \mathrm{err}(h; \boldsymbol{P}_g) \geq \widehat{\mathrm{err}}(h; \boldsymbol{P}_g) + \epsilon] \tag{21}$$

$$\geq \Pr_{X,Y \sim \boldsymbol{P}_g^n}[\exists h \in \mathcal{H} : \mathrm{err}(h; \boldsymbol{P}_g) \geq \epsilon_f - \epsilon_0 + \epsilon] \tag{22}$$

Choose $\epsilon = \epsilon_0$ to get

$$\Pr_{X,Y \sim \boldsymbol{P}_g^n}[\exists h \in \mathcal{H} : \mathrm{err}(h; \boldsymbol{P}_g) \geq \epsilon_f] \leq ce^{d}e^{-\epsilon^2 n} \tag{23}$$

$\square$

**Theorem A.9.** *For $\gamma \leq \alpha$, when there is no deteriorating shift, for a chosen significance level of $\alpha$, the FPR of Algo. 2 is at most $\gamma + (1 - \gamma)\,\mathcal{O}\left(\exp\left(-n\epsilon_0^2 + d\right)\right)$ if*

$$m \in \mathcal{O}\left(\left(\frac{1 - \sqrt{\delta}}{\epsilon_p - \epsilon_q}\right)^2 \left(d_p + \ln\frac{1}{\gamma}\right)\right) \tag{24}$$

*and $\epsilon_p - \epsilon_q > 0$, where $\delta = \frac{d_p + \ln\frac{1}{\alpha}}{d_p + \ln\frac{1}{\gamma}}$*

*Proof.* We show that in the case of no deteriorating shift (which implies $\epsilon_p \geq \epsilon_q$) the false positive rate cannot be more than $\alpha$ and also having more samples from $\boldsymbol{Q}_x$ will decrease the false positive rate if $\epsilon_p > \epsilon_q$.

We assume that during pre-training phase, while populating $\Phi$ we discard disagreement from $h \notin \mathcal{H}_p$ i.e., not satisfying the constraint $\text{err}(h; \boldsymbol{P}_f) \leq \epsilon_f$. We cannot do the same during the detection phase since the detection phase is time-sensitive. Due to this, we have to account for $h \notin \mathcal{H}_p$ in the FPR calculation.

Now, FPR can be written and bounded as follows. Let $\mu$ be the disagreement at $1 - \alpha$ percentile of $\Phi$

$$\text{FPR} = \Pr\left[\widehat{\text{err}}(h; \boldsymbol{Q}_f) \geq \mu\right] \tag{25}$$

$$= \Pr\left[\{\{h \notin \mathcal{H}_p\} \wedge \{\widehat{\text{err}}(h; \boldsymbol{Q}_f) \geq \mu\}\} \vee \{\{h \in \mathcal{H}_p\} \wedge \{\widehat{\text{err}}(h; \boldsymbol{Q}_f) \geq \mu\}\}\right] \tag{26}$$

$$\leq \Pr\left[\{h \notin \mathcal{H}_p\} \vee \{\{h \in \mathcal{H}_p\} \wedge \{\widehat{\text{err}}(h; \boldsymbol{Q}_f) \geq \mu\}\}\right] \tag{27}$$

$$\leq \Pr\left[h \notin \mathcal{H}_p\right] + \Pr\left[\{\widehat{\text{err}}(h; \boldsymbol{Q}_f) \geq \mu\} \mid \{h \in \mathcal{H}_p\}\right] \Pr\left[h \in \mathcal{H}_p\right] \tag{28}$$

$$= \gamma + (1 - \gamma) \Pr\left[h \notin \mathcal{H}_p\right] \tag{29}$$

$$\text{FPR} \leq \gamma + (1 - \gamma)\, \mathcal{O}\left(\exp\left(-n\epsilon_0^2 + d\right)\right) \tag{30}$$

where last equation comes from A.8 and $\gamma := \Pr\left[\{\widehat{\text{err}}(h; \boldsymbol{Q}_f) \geq \mu\} \mid \{h \in \mathcal{H}_p\}\right]$

Now, we derive sample complexity $m$ in terms of $\gamma$. Using A.7 on $\boldsymbol{P}$ with $1 - \alpha$ probability we get

$$m \in \mathcal{O}\left(\frac{d_p + \ln\frac{1}{\alpha}}{(\mu - \epsilon_p)^2}\right) \tag{31}$$

We use $\mu \in \Omega\left(\epsilon_p + \sqrt{\frac{d_p + \ln\frac{1}{\alpha}}{m}}\right)$ from above while using A.7 on $\boldsymbol{Q}$ with $1 - \gamma$ probability to get

$$m \in \mathcal{O}\left(\left(\frac{1 - \sqrt{\frac{d_p + \ln\frac{1}{\alpha}}{d_p + \ln\frac{1}{\gamma}}}}{(\epsilon_p - \epsilon_q)}\right)^2 \left(d_p + \ln\frac{1}{\gamma}\right)\right) \quad \text{for } \gamma < \alpha \tag{32}$$

Note that since the chosen $\mu$ was greater than $\epsilon_p$ and we are dealing with the case $\epsilon_p > \epsilon_q$, we get that the chosen $\mu$ is greater than $\epsilon_q$. Thus the requirement of $\mu$ is satisfied for A.7 while using for $\boldsymbol{Q}$.

$\square$

This theorem shows that when there are non deteriorating shifts (specifically $\epsilon_p > \epsilon_q$) FPR may be even less than $\alpha$, given $m$ and $n$ is sufficiently large. The more samples from $\boldsymbol{Q}_x$ we have the lesser the FPR in these cases. For any general case, by setting $\gamma = \alpha$ (i.e., $\delta = 1$) in the above theorem, we obtain the following:

**Corollary A.10.** *For a chosen significance level $\alpha$, the FPR of the D3M algorithm is no more than $\alpha + (1 - \alpha)\, \mathcal{O}\left(\exp\left(-n\epsilon_0^2 + d\right)\right)$.*

Note that this statement is independent of $m$ and the distribution shift. If $n$ is sufficiently large, the exponential term is small. This is often the case when the base classifier error $\epsilon_f$ is small, which is an indicator that a large number of samples ($n$) were available from $\boldsymbol{P}_g$. Ignoring non deteriorating shift (and $\boldsymbol{Q}_x \neq \boldsymbol{P}_x$) cases while calculating $\Phi$ in Algo. 2 does not adversely affect the FPR of the test.

**Lemma A.11.** *For any $\beta > 0, \mu < \epsilon_q$, there exists an $h \in \mathcal{H}_p$ such that $\widehat{\text{err}}(h; \boldsymbol{Q}_f) \geq \mu$ with probability at least $1 - \beta$ if*

$$m \geq \mathcal{O}\left(\frac{d_q + \ln\frac{1}{\beta}}{(\epsilon_q - \mu)^2}\right) \tag{33}$$

*Proof.* We use the generalization bound for agnostic learning case [63].

$$ce^{d_p}e^{-\epsilon^2 m} \geq \Pr_{X,Y \sim \boldsymbol{Q}_{1-f}^m}[\exists h \in \mathcal{H}_p : \widehat{\text{err}}(h; \boldsymbol{Q}_{1-f}) - \text{err}(h; \boldsymbol{Q}_{1-f}) \geq \epsilon] \tag{34}$$

$$= \Pr_{X,Y \sim \boldsymbol{Q}_{1-f}^m}[\exists h \in \mathcal{H}_p : \widehat{\text{err}}(h; \boldsymbol{Q}_f) \leq \text{err}(h; \boldsymbol{Q}_f) - \epsilon] \tag{35}$$

$$\overset{(a)}{=} \Pr_{X,Y \sim \boldsymbol{Q}_{1-f}^m}[\forall h \in \mathcal{H}_p : \widehat{\text{err}}(h; \boldsymbol{Q}_f) \leq \epsilon_q - \epsilon] \tag{36}$$

where (a) follows from Def. 4
Choose $\epsilon = \epsilon_q - \mu$ for any $\mu < \epsilon_q$

$$ce^{d_q}e^{-\epsilon^2 m} \leq \beta \tag{37}$$

$$m \geq \mathcal{O}\left(\frac{d_q + \ln\frac{1}{\beta}}{(\epsilon_q - \mu)^2}\right) \tag{38}$$

$\square$

**Proposition A.12** (D-PDD and TV distance). *The relations between $\xi$ (in Def. 5), $\eta$ (in Def. 6), and $\epsilon_p$, $\epsilon_q$ (in Def. 3 and 4) are as follows:*

$$\xi = \text{TV} - 2\eta \geq 0 \tag{39}$$

$$\xi \geq \epsilon_q - \epsilon_p \geq \xi - 2\epsilon_f \tag{40}$$

*Proof.* Recall the definition of $U$ from 6. We first derive the Bayes error in terms of TV distance. Let

$$A = \{x \in \mathcal{X} \mid \boldsymbol{Q}_x(x) \leq \boldsymbol{P}_x(x)\} \tag{41}$$

$$A' = \{x \in \mathcal{X} \mid \boldsymbol{Q}_x(x) > \boldsymbol{P}_x(x)\} \tag{42}$$

The TV distance is equal to half of the $L_1$ distance. Note that $\boldsymbol{P}_x(A) + \boldsymbol{P}_x(A') = 1$ and similarly for $\boldsymbol{Q}_x$. [2]

$$\text{TV}(\boldsymbol{P}_x, \boldsymbol{Q}_x) = \frac{1}{2}\left(\boldsymbol{P}_x(A) - \boldsymbol{Q}_x(A) + \boldsymbol{Q}_x(A') - \boldsymbol{P}_x(A')\right) \tag{43}$$

$$= 1 - \boldsymbol{P}_x(A') - \boldsymbol{Q}_x(A) \tag{44}$$

Now, we use the definition of $U$ and the above TV relation to get the following

$$\text{err}(f_{\text{bayes}}; \boldsymbol{U}) = \frac{1}{2}\left(\text{err}(f_{\text{bayes}}; \boldsymbol{P}_f) + \text{err}(f_{\text{bayes}}; \boldsymbol{Q}_{1-f})\right) = \frac{1}{2}\left(\boldsymbol{Q}_x(A) + \boldsymbol{P}_x(A')\right) \tag{45}$$

$$= \frac{1}{2}\left(1 - \text{TV}(\boldsymbol{P}_x, \boldsymbol{Q}_x)\right) \tag{46}$$

Next, with the above result and $\eta$ in Eq. 6 we derive Eq. 8

$$\eta + \text{err}(f_{\text{bayes}}; \boldsymbol{U}) = \min_{h \in \mathcal{H}_p} \text{err}(h; \boldsymbol{U}) = \frac{1}{2} \min_{h \in \mathcal{H}_p} \left(\text{err}(h; \boldsymbol{P}_f) + \text{err}(h; \boldsymbol{Q}_{1-f})\right) \tag{47}$$

$$2\eta + 1 - \text{TV} = \min_{h \in \mathcal{H}_p} \left(\text{err}(h; \boldsymbol{P}_f) + \text{err}(h; \boldsymbol{Q}_{1-f})\right) \geq \min_{h \in \mathcal{H}_p} \text{err}(h; \boldsymbol{Q}_{1-f}) \tag{48}$$

$$2\eta + 1 - \text{TV} = \min_{h \in \mathcal{H}_p} \left(\text{err}(h; \boldsymbol{P}_f) - \text{err}(h; \boldsymbol{Q}_f)\right) + 1 \tag{49}$$

$$2\eta - \text{TV} = \min_{h \in \mathcal{H}_p} -\left(\text{err}(h; \boldsymbol{Q}_f) - \text{err}(h; \boldsymbol{P}_f)\right) \tag{50}$$

$$\text{TV} - 2\eta = \max_{h \in \mathcal{H}_p} \left(\text{err}(h; \boldsymbol{Q}_f) - \text{err}(h; \boldsymbol{P}_f)\right) = \xi \tag{51}$$

For Eq. 9, we use Eq. 48 and the above result to get the following

$$\epsilon_q = \max_{h \in \mathcal{H}_p} \text{err}(h; \boldsymbol{Q}_f) = 1 - \min_{h \in \mathcal{H}_p} \text{err}(h; \boldsymbol{Q}_{1-f}) \geq \text{TV} - 2\eta = \xi \tag{52}$$

---

[2] With some abuse of notation, we use the same notation for both pdf and probability measure.

We can write an inequality for errors similar to triangle inequality as follows

$$\text{err}(h; \boldsymbol{P}_f) \leq \text{err}(h; \boldsymbol{P}_g) + \text{err}(g; \boldsymbol{P}_f) \tag{53}$$

$$= \text{err}(h; \boldsymbol{P}_g) + \text{err}(f; \boldsymbol{P}_g) = \text{err}(h; \boldsymbol{P}_g) + \epsilon_f \tag{54}$$

$$\epsilon_p = \max_{h \in \mathcal{H}_p} \text{err}(h; \boldsymbol{P}_f) \leq \max_{h \in \mathcal{H}_p} \text{err}(h; \boldsymbol{P}_g) + \epsilon_f = 2\epsilon_f \tag{55}$$

The last equality follows from the definition of $\mathcal{H}_p$. Thus we get

$$\epsilon_q - \epsilon_p \geq \xi - 2\epsilon_f \tag{56}$$

By definition it follows that $\xi \geq \epsilon_q - \epsilon_p$ $\qquad\square$

**Proposition A.13.** *For $\beta > 0$, when the deteriorating shift occurs, for a chosen significance level of $\alpha$, the TPR of Algo. 2 is at least $(1 - \beta)\left(1 - \mathcal{O}\left(\exp\left(-n\epsilon_0^2 + d\right)\right)\right)$ if*

$$m \in \mathcal{O}\left(\left(\frac{1 + \sqrt{\delta}}{\xi - 2\epsilon_f}\right)^2 \left(d_p + \ln\frac{1}{\beta}\right)\right) \tag{57}$$

*and $\epsilon_q - \epsilon_p > 0$, where $\delta = \frac{d_p + \ln\frac{1}{\alpha}}{d_p + \ln\frac{1}{\beta}}$*

*Proof.* Similar to the proof of Theorem. A.3, we derive the statistical power (TPR) of the test as follows. Let $\mu$ be the disagreement at $1 - \alpha$ percentile of $\Phi$

$$\text{TPR} = 1 - \Pr\left[\widehat{\text{err}}(h; \boldsymbol{Q}_f) \leq \mu\right] \tag{58}$$

$$= 1 - \Pr\left[\{\{h \notin \mathcal{H}_p\} \wedge \{\widehat{\text{err}}(h; \boldsymbol{Q}_f) \leq \mu\}\} \vee \{\{h \in \mathcal{H}_p\} \wedge \{\widehat{\text{err}}(h; \boldsymbol{Q}_f) \leq \mu\}\}\right] \tag{59}$$

$$\geq 1 - \Pr\left[\{h \notin \mathcal{H}_p\} \vee \{\{h \in \mathcal{H}_p\} \wedge \{\widehat{\text{err}}(h; \boldsymbol{Q}_f) \leq \mu\}\}\right] \tag{60}$$

$$\geq 1 - \Pr\left[h \notin \mathcal{H}_p\right] - \Pr\left[\{\widehat{\text{err}}(h; \boldsymbol{Q}_f) \leq \mu\} \mid \{h \in \mathcal{H}_p\}\right]\Pr\left[h \in \mathcal{H}_p\right] \tag{61}$$

$$= (1 - \beta)\Pr\left[h \in \mathcal{H}_p\right] \tag{62}$$

$$\text{TPR} \in (1 - \beta)\left(1 - \mathcal{O}\left(\exp\left(-n\epsilon_0^2 + d\right)\right)\right) \tag{63}$$

where last equation comes from A.8 and $\beta := \Pr\left[\{\widehat{\text{err}}(h; \boldsymbol{Q}_f) \leq \mu\} \mid \{h \in \mathcal{H}_p\}\right]$

Next, we derive the sample complexity $m$ in terms of $\beta$. We show that there exists a $\mu^*$ such that both A.7 (for $\boldsymbol{P}$ and $\alpha$) and A.11 (for $\boldsymbol{Q}$ and $\beta$) hold.

$$\epsilon_p < \mu < \epsilon_q \tag{64}$$

This implies some $\mu$ exists if $\epsilon_q - \epsilon_p > 0$

We find optimal $\mu^*$ such that the maximum of $m$ in Eq. 14 and Eq. 33 is minimized.

$$\left(\frac{\mu - \epsilon_p}{\epsilon_q - \mu}\right)^2 = \frac{d_p + \ln\frac{1}{\alpha}}{d_p + \ln\frac{1}{\beta}} := \delta \tag{65}$$

$$\mu^* = \frac{\epsilon_p + \sqrt{\delta}\epsilon_q}{1 + \sqrt{\delta}} \tag{66}$$

Plugging this $\mu^*$ in Eq. 33 gives

$$m \in \mathcal{O}\left(\frac{d + \ln\frac{1}{\beta}}{(\epsilon_q - \epsilon_p)^2}\left(1 + \sqrt{\frac{d + \ln\frac{1}{\alpha}}{d + \ln\frac{1}{\beta}}}\right)^2\right) \tag{67}$$

Use Eq. 9 to get the result. $\qquad\square$

$\xi$ in the denominator indicates that as the shift becomes more deteriorating, it is easier (fewer samples $m$) to monitor, indicating the effectiveness of the D3M algorithm. Also, having a high-quality base classifier (low $\epsilon_f$) is better for D3M which was also seen in Eq. 9 where low $\epsilon_f$ makes the algorithm more faithful. Note that $m$ depends on $d_p$ which can be much less than $d$ which $n$ depends on, suggesting that monitoring may be effective in few-shot settings. The dependence on $n$ is due to the requirement of satisfaction of condition 1 in Def. 2. In the optimization problems in Algo. 1, the empirical constraint is satisfied but the population constraint will be satisfied either for larger $\epsilon_0$ or for sufficiently large $n$ as seen in the theorem.

Next, we move to the regime where deteriorating shift occurs but $\epsilon_q - \epsilon_q \leq 0$. As a negative result, the following theorem states that in such cases the statistical power of the test is low.

**Theorem A.14.** *When deteriorating shift occurs and $\epsilon_q \leq \epsilon_p$, for a chosen significance level of $\alpha$, the statistical power of the test in Alg. 2 is $\mathcal{O}(\alpha)$.*

*Proof.* From the proof of Theorem. A.5 we have

$$\text{TPR} \geq (1 - \beta)\left(1 - \mathcal{O}\left(\exp\left(-n\epsilon_0^2 + d\right)\right)\right) \tag{68}$$

$$\beta := \Pr\left[\{\widehat{\text{err}}(h; \boldsymbol{Q}_f) \leq \mu\} \,|\, \{h \in \mathcal{H}_p\}\right] \tag{69}$$

Using A.7 on $\boldsymbol{P}$ and $\alpha$ we get

$$\alpha \in \mathcal{O}(\exp\left(-n(\mu - \epsilon_p)^2 + d_p\right)) \tag{70}$$

Using A.7 on $\boldsymbol{Q}$ we get

$$\Pr\left(\{\widehat{\text{err}}(h; \boldsymbol{Q}_f) \geq \mu\} \,|\, \{h \in \mathcal{H}_p\}\right) \in \mathcal{O}\left(\exp\left(-n(\mu - \epsilon_q)^2 + d_p\right)\right) \tag{71}$$

$$1 - \beta \in \mathcal{O}\left(\exp\left(-n(\mu - \epsilon_q)^2 + d_p\right)\right) \tag{72}$$

$$1 - \beta \in \mathcal{O}\left(\alpha\right) \tag{73}$$

The last equation follows since we are dealing with the case where $\epsilon_p \geq \epsilon_q$. Thus, the TPR is $\mathcal{O}(\alpha)$ irrespective of the magnitude of $n$, as desired. $\qquad\square$

# B   Experimental Setup and Additional Details

## B.1   Baselines Details

We compare D3M against several other methods from the literature that either detect distribution changes or **can be converted into a PDD monitoring protocol**. Let $X = \{\boldsymbol{x}^{(i)}\}_{i=1}^{n}$ from $\boldsymbol{P}_x$ and $Y = \{\boldsymbol{y}^{(i)}\}_{i=1}^{m}$ from $\boldsymbol{Q}_x$ be given. These algorithms seek to accept or reject the hypothesis that $\boldsymbol{P}_x = \boldsymbol{Q}_x$ in distribution.

1. Deep Kernel MMD (MMD-D, [13]) The algorithm first learns a deep kernel by optimizing a criterion which yields the most powerful hypothesis test. With this learned kernel, permutation tests are run multiple times in order to determine a true positive rate for the algorithm. We interface the authors' original source code with our repository and recycle their training procedures. Theirs can be found at `https://github.com/fengliu90/DK-for-TST`.

2. H-Divergence (H-Div, [14]) The algorithm fits Gaussian kernel density estimates for $\boldsymbol{P}_x$, $\boldsymbol{Q}_x$, and their uniform mixture $(\boldsymbol{P}_x + \boldsymbol{Q}_x)/2$. Then, permutation tests are performed using the test statistic $H_\ell((\boldsymbol{P}_x + \boldsymbol{Q}_x)/2) - \min\{H_\ell(\boldsymbol{Q}_x), H_\ell(\boldsymbol{P}_x)\}$ where $H_\ell$ is the H-entropy with $\ell(x, a)$ the negative log likelihood of $x$ under distribution $a$ estimated by the Gaussian kernel density, in order to determine a true positive rate for the algorithm. This test statistic is an empirical estimate of the H-Min divergence. The choice of the particular H-divergence is a hyperparameter and is problem dependent, as well as the choice for how to generatively model the data distributions. The original paper further experimented with fitting Gaussian distributions as well as variational autoencoders (VAEs), both of which are not explored here. We interface the authors' original source code with our repository. Theirs can be found at `https://github.com/a7b23/H-Divergence/tree/main`.

3. $f$-Divergences (KL-Div for KL-Divergence, JS-Div for Jensen-Shannon Divergence, [61]) $f$-divergence generalizes several notions of distances between probability distributions commonly used in machine learning. In this paper, we convert the Kullback-Leibler (KL) and the Jensen-Shannon (JS) divergences, particular cases of $f$-divergences, into permutation tests. More specifically, we first fit Gaussians on samples coming from $\boldsymbol{P}_x$ and $\boldsymbol{Q}_x$ using maximum likelihood. In the case of KL-divergence, the empirical KL-divergence is computed between the fitted Gaussians whereas for the JS-divergence, we fit an additional Gaussian on the mixture distribution $\boldsymbol{M}$ and leverage the identity:

$$\mathrm{JS}(\boldsymbol{P}_x || \boldsymbol{Q}_x) = \frac{1}{2}(\mathrm{KL}(\boldsymbol{P}_x || \boldsymbol{M}) + \mathrm{KL}(\boldsymbol{Q}_x || \boldsymbol{M}))$$

We run permutation tests by permuting the samples in the union $(X \sim \boldsymbol{P}_x^n) \cup (Y \sim \boldsymbol{Q}_x^m)$. It is worth noting that as with H-divergence, more elaborate generative models could be fitted onto samples $X$ and $Y$, which we do not explore in this work.

4. Black Box Shift Detection (BBSD, [38]) involves estimating the changes in the distribution of target labels $p(y)$ between training and test data while assuming that the conditional distribution of features given labels $p(x|y)$ remains constant. This is achieved by using a black box model's confusion matrix to identify discrepancies in the marginal label probabilities between the training and test distributions, allowing detection and correction of the shift. We borrow the experimental setup directly from [1].

5. Relative Mahalanobis Distance (RMD, Rel-MH, Rel. Mahalanobis, [39]) RMD modifies the traditional Mahalanobis Distance (MD) for out-of-distribution (OOD) detection by accounting for the influence of non-discriminative features. It subtracts the MD of a test sample to a background class-independent Gaussian from the MD to each class-specific Gaussian, effectively isolating discriminative features and improving OOD detection, especially for near-OOD tasks. We test for shift by performing a KS test directly on the distribution of the RMD confidence scored computed on $\boldsymbol{Q}_x$ and $\boldsymbol{P}_x$.

6. Classifier two-sample test (CTST, C2ST, a.k.a. Domain Classifier, DC, [41]) Following the procedure presribed by the original work, a binary domain classifier is trained to predict whether a sample came from $\boldsymbol{P}_x$ or $\boldsymbol{Q}_x$. Then, on a held-out mixture of data from $\boldsymbol{P}_x$ and $\boldsymbol{Q}_x$, we compute the domain classifier's accuracy and compare its performance to random chance using a binomial test.

7. Deep Ensemble (Deep Ensemble, [40]) An ensemble of neural networks are trained independently on the entire training dataset using random initialization. A KS test is then performed on the distribution of entropy values computed from each sample in $P$ and $Q$.

## B.2 Standard Benchmark Datasets

**UCI Heart Disease.**    The UCI Heart Disease dataset (UCI) [21, 64] includes 76 variables gathered from four distinct patient cohorts located in Cleveland, Hungary, Switzerland, and the VA Long Beach. To reduce the impact of missing data, we focus on nine out of the 14 most frequently used features: age, sex, chest pain type, resting blood pressure, serum cholesterol, fasting blood sugar, resting ECG results, maximum heart rate achieved, and exercise-induced angina. The objective is to predict heart disease diagnosis, measured on a scale from 0 to 4—where 0 denotes no disease and values 1 through 4 indicate increasing severity related to arterial narrowing. Using the proposed setup in [19], the task is binarized, distinguishing between patients with a normal angiographic diagnosis (label 0) and those with any abnormal diagnosis (label greater than 0). The Cleveland and Hungary datasets serve as the ID domain, while Switzerland and VA Long Beach datasets form the deteriorating OOD domain.

**CIFAR-10/10.1.**    The CIFAR-10 image dataset [32] consists of 60,000 color images, each sized 32x32 pixels, divided into 10 different classes such as airplanes, cars, birds, and dogs. The dataset is split into 50,000 training images and 10,000 test images, with each class represented equally. Due to its manageable size and diversity of categories, CIFAR-10 is commonly used for testing and comparing the performance of image recognition models. The CIFAR-10.1 dataset [2] is a test set designed to evaluate how well models trained on CIFAR-10 generalize to new, but similar, data. It consists of 2,000 images collected in a way that closely matches the original CIFAR-10 distribution, but from a separate data source to reduce the risk of overlap or memorization. CIFAR-10.1 was introduced to assess model robustness and identify potential overfitting to the original test set. Despite its similarity to CIFAR-10, many models perform slightly worse on CIFAR-10.1, highlighting the challenge of generalization in machine learning.

The CIFAR-10 dataset is used as the ID dataset with which the mean model of D3M is trained, while the CIFAR-10.1 dataset is considered as a deteriorating shift from CIFAR-10. Thus, we gauge the ability to recognize CIFAR-10.1 as deterioratingly OOD by D3M and its competing baselines.

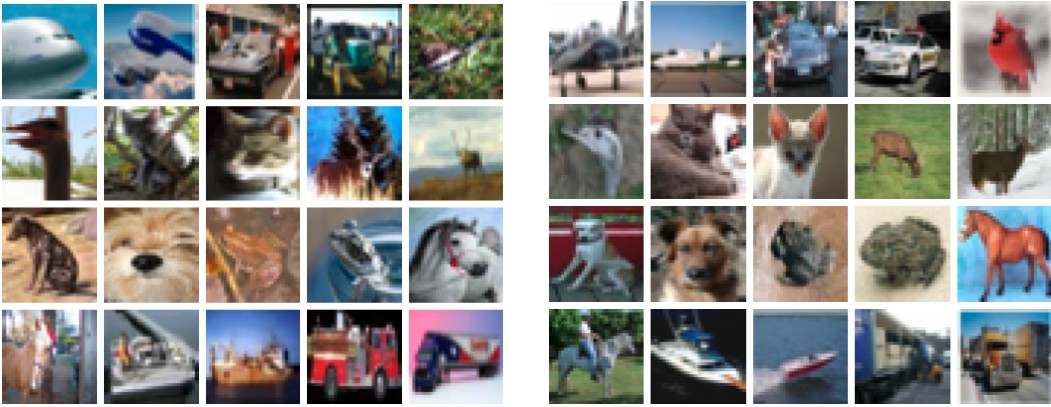

**Figure 5:** (**Left**) Random samples from CIFAR-10.1. (**Right**) Random samples from CIFAR-10's test set. The images above are borrowed from Recht et. al. "Do CIFAR-10 Classifiers Generalize to CIFAR-10?" [2].

**Camelyon17.**    The Camelyon17 dataset is a challenging histopathology image classification dataset originally described by [65]. It comprises 327,680 color images (96×96) extracted from lymph node tissue slides, with binary labels indicating the presence of metastatic cancer in the central 32×32 region. Following the WILDS setup, we treat the hospitals from which data was collected as domains: hospitals 1, 2, and 3 are used as source domains for training, while hospital 5 serves as the target test

domain. We use the WILDS framework [4] to handle dataset download, preprocessing, and domain partitioning.

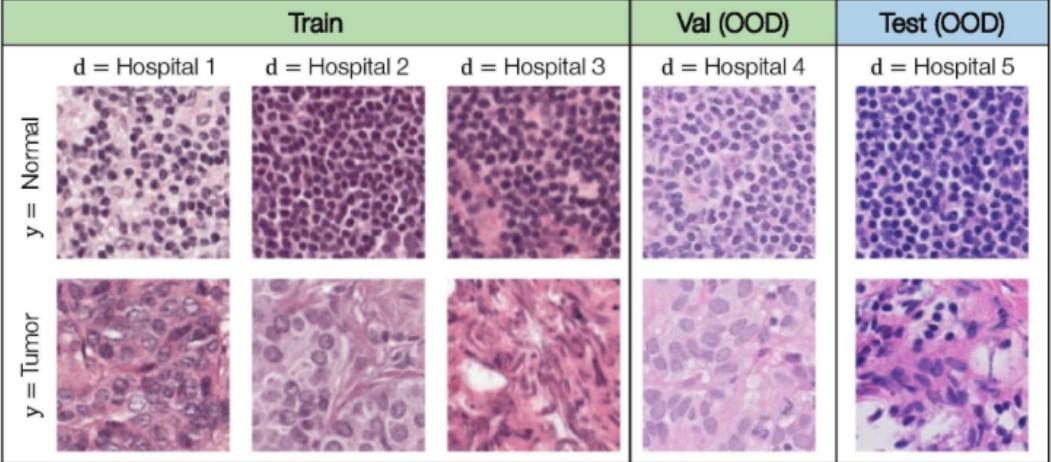

**Figure 6:** Samples from the Camelyon17 dataset, where hospitals 1, 2, 3 are treated as ID, while hospitals 4 and 5 are deterioratingly OOD. The image above is borrowed from Koh and Sagawa et. al. "WILDS: A benchmark of in-the-wild distribution shifts" [4].

## B.3 The GEneral Medicine INpatient Initiative (GEMINI) Dataset

| Year | Patient Count | Label Ratio |
|---|---|---|
| Pre-2017 | 72316 | 3.99% |
| 2017H2 | 17208 | 3.60 % |
| 2018H1 | 18233 | 4.15% |
| 2018H2 | 18469 | 3.83% |
| 2019H1 | 19041 | 3.50% |
| 2019H2 | 18601 | 3.49% |
| 2020H1 | 15575 | 4.50% |
| 2020H2 | 11155 | 3.48% |
| 2021H1 | 10625 | 3.46% |
| 2021H2 | 7396 | 2.95% |

**Table 3:** Temporal Split Data Summary

| Age | Patient Count | Label Ratio |
|---|---|---|
| 18-52 | 33220 | 0.82 % |
| 52-66 | 33146 | 2.36 % |
| 66-72 | 31048 | 3.36 % |
| 76 - 85 | 34055 | 4.77 % |
| 85+ | 32399 | 7.82 % |

**Table 4:** Age Split Data Summary

**GEMINI Dataset: Study and Preprocessing.** The GEMINI study is a retrospective cohort study of adult patients and their clinical and administrative data[22, 23]. This analysis used data from over 200,000 patients from the GEMINI database, spanning 7 different hospitals that participated in the GEMINI Study. Each patients information is processed into 900 features including but not limited to: (i) laboratory results and vital results collected up to 48-hours after admission, split into 6 hour intervals, (ii) patient demographic information: age, sex etc, (iii) Patient diagnosis using ICD-10-CA codes. Missing feature values are imputed based on simple averaging. The predictive task related to this data is 14-day mortality for patients based on these collected features. This dataset is temporal by nature as data splits are arranged per half-years starting from 2017, where shifts in patient features become even more pronounced during the COVID period of 2020-2021. Since our D3M base models trained on data before 2018 surprisingly do not underperform on later splits (Figure 2), we make the assessment that this shift is non-deteriorating.

Furthermore, an important feature of this dataset is its inherent temporal covariate shift, the evidence of which is in the elevated FPR of divergence/distance-based detectors of Deep Kernel MMD (MMD-D) and Relative Mahalanobis Distance (Rel-MH). Functionally, this means that patients' static covariates as well as their labs, medications, and medical interventions have drifted over time, especially into the COVID years.

**Data Splitting and Shift.** Based on this pre-processed data, 2 shifts are analyzed: (i) temporal shift, and (ii) age-group shift. The temporal shift analysis splits data into half-years - 2018H1, 2019H2, etc. The baseline model uses 2017H1 and prior data for training, and 2017H2 for validation; Tab. 3 shows patient statistics for this split. It is subsequently tested on unseen in distribution data and later splits. The different age groups are created by splitting the data into 5 equally sized groups based on ages of patients: (1) 18-52, (2) 52-66, (3) 66-72, (4) 76 - 85, (5) 85+; Tab. 4 shows patient statistics for this split. The reported analysis trains a baseline model on group 1 (18-52) and then tests on test-sets that contain some portion of data from the 5th group (85+) and the remaining as unseen in distribution data. The portions [0.0, 0.2, 0.4, 0.6, 0.8, 1.0] represent what percentage of the test set is OOD (from group 5), whilst the remaining amount is ID (from group 1). For example a ratio of 0.2 means 20% of the test data is from group 5(OOD) whilst 80% is from group 1(ID). We chose to experiment on such portions instead of just subsequent age groups as this process better displays the TPR of D3M as well as baselines with respect to the degree of shift / performance deterioration.

## B.4   Configurations for $\mathrm{FE}_\theta$ and $\mathrm{VBLL}_\theta$

**Tabular features.**   In the tabular setting, we chain affine layers of similar hidden dimension coupled with exponential linear unit (ELU) non-linearities [66]. We use a standard dropout rate of 0.2 across all experiments and employ skip connections between hidden layers. This architecture is used in all experiments with the UCI Heart Disease dataset as well as the GEMINI dataset.

**Convolutional features.**   For CIFAR-10/10.1, we pass the input through an initial convolution and max-pooling, then through a sequence of same-dimension convolutions with batch normalization [67] and skip-connections. Another max-pool is applied, before the representation is flattened and forwarded through an affine layer to obtain a representation.

For Camelyon17 experiments, we employ pre-trained ResNets [37] and either train from scratch, finetune them during the training of the $\mathrm{VBLL}_\theta$ layer, or freeze them while only training the VBLL layer.

**Variational Bayesian Last Layers (VBLL).**   We borrow the implementation from [25] which can be readily coupled with the above neural feature extractors for end-to-end ELBO maximization. In our experiments, we employ the VBLL variant for discriminative classification `vbll.DiscClassification`, and parametrize the covariance matrix of the normal distribution of logits as a diagonal. As for VBLL's hyperparameters, the prior scale and wishart scale hyperparameters are as described in the original manuscript, while we use the **regularization factor** to VBLL to be a factor of $n^{-1}$ where $n$ is the size of the training set. This factor controls the weight of the KL estimate during ELBO maximization at training.

## B.5   Sweeping and Hyperparameters

In each experiment, for each test sample size $m$, we run a hyperparameter sweep in order to identify hyperparameters that (1) achieves the most consistent low FPR when tested in-distribution, and (2) achieves the highest deterioration monitoring TPR. As mentioned previously, the distribution of disagreement rates can be overfitted so that D3M flags **any** sample, regardless of whether they are ID or not, thus justifying the choice of selecting hyperparameters jointly satisfying (1) and (2).

Once the best sets are identified, we run 10 independent seeded runs for each test sample size $m = 10, 20, 50$. We report used hyperparameters in Tables 5 and 6 for transparency and reproducibility. In CIFAR-10/10.1, "Hidden dimension" refers to the dimensionality of the final output of $\mathrm{FE}_\theta$, and test size $m$ is identical for D3M and other baseline algorithms for each set of experiments.

**Common to all setups.**   For all experiments, the number of posterior samples $K$ is set to 5000, the size of the empirical distribution of maximum disagreement rates $|\Phi|$ is set to 1000. For each experiment, once **Train** and **Calibrate** is completed, we deploy the model on 100 independent samplings of the questionable test data. If model deterioration occurs (this is the case for the standard benchmark experiments as well as the GEMINI dataset *temporal* shift experiment but not the GEMINI dataset *age* shift experiment), we report the number of times D3M flagged these deteriorating samples out of 100 as TPR. Finally, confidence statistics are aggregated and computed on TPRs reported from seeded, independent **Train**-**Calibrate**-**Deploy** D3M cycles. All optimization is done using AdamW

**Table 5:** Hyperparameters for UCI Heart Disease and GEMINI datasets

| Group | Hyperparameter | UCI | GEMINI |
|---|---|---|---|
| **Train** | Learning rate | $1 \times 10^{-3}$ | $1 \times 10^{-3}$ |
| | Batch size | 64 | 64 |
| | Epochs | 50 | 50 |
| | Weight decay | $1 \times 10^{-4}$ | $1 \times 10^{-4}$ |
| **Model** | Hidden dimension | 16 | 128 |
| | Num. hidden layers | 4 | 4 |
| | Dropout | 0.2 | 0.2 |
| **VBLL** | Regularization factor | 100.0 | 100.0 |
| | Prior scale | 1.0 | 1.0 |
| | Wishart scale | 1.0 | 1.0 |
| **D3M** | Sampling temperature | 1.0 | 1.0 |
| | Test size $m$ | $10, 20, 50$ | 200 |

**Table 6:** Hyperparameters for CIFAR-10/10.1 and Camelyon17 datasets

**CIFAR-10/10.1**

| Group | Hyperparameter | Value |
|---|---|---|
| **Train** | Learning rate | $1 \times 10^{-3}$ |
| | Batch size | 64 |
| | Epochs | 10 |
| | Weight decay | $1 \times 10^{-4}$ |
| **Model** | Hidden dimension | 256 |
| | Num. mid layers | 3 |
| | Initial kernel size | 9 |
| | Kernel size | 7 |
| | Num. mid channels | 128 |
| **VBLL** | Regularization factor | 10.0 |
| | Prior scale | 1.0 |
| | Wishart scale | 1.0 |
| **D3M** | Sampling temperature | 1.0 |
| | Test size $m$ | $10, 20, 50$ |

**Camelyon17**

| Group | Hyperparameter | Value |
|---|---|---|
| **Train** | Learning rate | $1 \times 10^{-5}$ |
| | Batch size | 256 |
| | Epochs | 2 |
| | Weight decay | $1 \times 10^{-4}$ |
| **Model** | ResNet type | ResNet34 |
| | From pretrained | True |
| | Freeze features | True |
| **VBLL** | Regularization factor | 100.0 |
| | Prior scale | 5.0 |
| | Wishart scale | 2.0 |
| **D3M** | Sampling temperature | 2.0 |
| | Test size $m$ | $10, 20, 50$ |

[68]. Finally, we start seeded, independent, identical runs beginning at seed $= 57$ since it is our favorite prime number, and increase seed by $+1$ for each run.

**Only reporting TPRs of runs achieving low FPR.** Importantly, we aggregate only TPRs achieved when a FPR below 0.10 in-distribution is achieved. This FPR is calculated on a held-out ID validation set that D3M has not yet seen during **Train** nor **Calibrate**. Therefore, for all experiments we run seeded runs until 10 runs recording ID FPR below 0.10 are found, and their TPR statistics are computed, as certain runs do not achieve the ID FPR tolerance desired.

When deploying D3M in a real-world healthcare setting, for instance, upon the completion of the **Calibrate** step, one could imagine validating this calibration step by computing an ID FPR score, independent of deployment. IF this ID FPR score is higher than a tolerated threshold $\alpha$, the practitioner could either increase the size of $\Phi$ to eliminate noise from sampling, or consider finding another set of hyperparameters that would lead to more stable calibration.

### B.6 Additional D3M Results on Standard Benchmark

We further tested D3M on 100 and 200 calibration/test samples on the same set of hyperparameters above. Table 7 summarizes our findings.

|  | UCI Heart Disease | | CIFAR 10.1 | | Camelyon 17 | |
|---|---|---|---|---|---|---|
|  | 100 | 200 | 100 | 200 | 100 | 200 |
| **D3M (Ours)** | .93±.10 | .99±.01 | .91±.11 | .99±.01 | 1.0±.00 | 1.0±.00 |

**Table 7:** True positive rates (TPR) for D3M across datasets and test sizes $100, 200$.

**At higher test sizes, D3M achieves near-perfect TPR.** This is consistent across all experiments. This suggests that D3M could be an effective tool to monitor model deterioration. However, of note is that for test size 100, in the UCI Heart Disease and CIFAR-10/10.1 experiments, D3M is still logging unusually high standard deviation, where we suspect the noise to be coming from the sampling procedure from the $\mathrm{VBLL}_\theta$ distribution of logits.

**Trading off deployment-time computations with sweeping** D3M is efficient when comparing to other disagreement-based detection and monitoring algorithms. However, the method is inherently noisier due to several levels of approximation to its idealized version (Algorithms 1 and 2). In particular, we found that D3M is **sensitive to the choice of hyperparameters** in order to achieve great performance. Therefore, the payment of computational cost is carried through prior to deployment in the sweeping itself, rather than during deployment as is done in [19]. We argue, however, that this is **preferable in edge deployment scenarios**, such as in hospitals or embedded systems, where real-time responsiveness and resource constraints are critical by front-loading the computational burden during the sweeping phase. One can imagine that for an AI terminal as part of a hospital computational infrastructure for instance, the ability to perform robust detection and monitoring with minimal overhead at deployment time significantly enhances reliability and usability in practice.

### B.7  Statement on the Usage of Computing Resources

All experiments were run on High Performance Computing (HPC) clusters.

**UCI Heart Disease.** UCI Heart Disease experiments were run on GPU nodes with at minimum 8GB of GPU memory, 6 CPU cores, and 8GB RAM. The total runtime of D3M prior to deployment is less than 5 minutes.

**CIFAR-10/10.1.** CIFAR-10/10.1 experiments were run on GPU nodes with at minimum 24GB of GPU memory to accomodate the largest configurations of convolutions, 12 CPU cores, and 12GB RAM. The total runtime of D3M prior to deployment is less than 10 minutes.

**Camelyon17.** Camelyon17 experiments were run on GPU nodes with at minimum 80GB of GPU memory to accomodate the largest ResNets during sweeping, 12 CPU cores, and 12GB RAM. The total runtime of D3M prior to deployment is less than 1 hour.

**GEMINI Dataset.** Experiments on the GEMINI datset were run on GPU and CPU nodes. We request at minimum 16GB of GPU memory (when applicable), 9 CPU cores, and 32GB of RAM. The total runtime of D3M prior to deployment is less than 1 hour. Although models for GEMINI were significantly smaller than vision models for Camelyon17 and CIFAR-10/10.1, due to the number of samples available as well as older CPU hardware, the runtime was significantly extended.

# C Ablations

To complement the theoretical analysis and main results, we provide additional ablation studies that clarify the empirical behavior of D3M. Each ablation isolates a specific modeling or design choice, allowing us to assess its contribution to performance, stability, and efficiency. These results also serve to validate our claims about scalability and robustness, while highlighting trade-offs between alternative design choices (e.g., linear vs. VBLL heads). We organize this section as follows: (i) VBLL head vs. Linear head, (ii) computational costs, especially when compared with Detectron, (iii) sensitivity to hyperparameters such as temperature and ID thresholds, and (iv) the relationship between empirical oversampling and the theoretical D3M framework. Importantly, where applicable, all ablation experiments are run with the same best hyperparameter sets which we isolated during our sweeps and reported in Appendix B.5.

## C.1 VBLL Head vs. Linear Head

Implementing a VBLL head is what allows D3M to sample alternate hypotheses (re: Idealized D3M formulation, i.e. Algorithms 1 and 2). It is important that transitioning to VBLL from a classical linear head does not amount to significant decreases in base model performance on the ID task. The following table reports model accuracy, F1 score, AUROC, and Mathew's Correlation Coefficient (MCC), all computed on an ID validation set, along with a deployment accuracy drop.

**Table 8:** Comparison of D3M vs. Linear head (Lin) across datasets.

| Dataset | Model | Accuracy | F1 Score | AUROC | MCC | Acc. Drop OOD |
|---------|-------|----------|----------|-------|-----|---------------|
| **UCI** | D3M | $0.76 \pm 0.02$ | $0.75 \pm 0.02$ | $0.86 \pm 0.01$ | $0.51 \pm 0.04$ | $-0.11 \pm 0.02$ |
| | Lin | $0.77 \pm 0.01$ | $0.76 \pm 0.01$ | $0.87 \pm 0.01$ | $0.52 \pm 0.02$ | — |
| **CIFAR-10/10.1** | D3M | $0.70 \pm 0.02$ | $0.70 \pm 0.02$ | $0.96 \pm 0.00$ | $0.67 \pm 0.02$ | $-0.13 \pm 0.02$ |
| | Lin | $0.72 \pm 0.01$ | $0.72 \pm 0.01$ | $0.96 \pm 0.00$ | $0.69 \pm 0.01$ | — |
| **Camelyon17** | D3M | $0.94 \pm 0.01$ | $0.94 \pm 0.01$ | $0.98 \pm 0.00$ | $0.88 \pm 0.01$ | $-0.05 \pm 0.00$ |
| | Lin | $0.94 \pm 0.00$ | $0.94 \pm 0.00$ | $0.98 \pm 0.00$ | $0.88 \pm 0.01$ | — |

In the above, for multi-class classification datasets, F1, AUROC, and MCC are averaged across all pairs of classes. All hyperparameters of the feature extractors in the above setup are kept the same. We confirm that consistently across all metrics, the performance of D3M's base model with a VBLL head nearly matches that of a base linear head. We argue that this is a worthwhile tradeoff in order to leverage hypothesis sampling with VBLL. We observe that there is virtually no performance drop for Camelyon17. We suspect this is due to the base model being finetuned from a pretrained ResNet-34 on ImageNet-1K, having learned more general representations, as opposed to our ConvNet feature extractors implemented for the CIFAR-10/10.1 experiments that may have overfitted to the ID training dataset.

## C.2 Computational Costs Analysis

To better understand the scalability of D3M relative to existing approaches, we report the floating point operation counts (FLOPs) required by each algorithm across their main computational stages. Table 9 presents the breakdown of cost into (1) base model training and (2) calibration, along with the total compute required. We assume a query size of 100, and all hyperparameters are identical to those reported in the experimental setup.

We observe several consistent patterns across datasets and architectures. First, in D3M, the bulk of the computational burden lies in the one-time base training of the feature extractor, after which calibration incurs only a marginal cost. This scaling behavior arises because calibration in D3M consists of freezing the backbone in evaluation mode and drawing samples from the VBLL layers. As a result, no backpropagation or fine-tuning steps are required once the base model has been trained.

In contrast, Detectron's design requires fine-tuning on held-out subsets during calibration. This entails repeating gradient computations and updating parameters at deployment time. Consequently, the calibration phase is nearly as expensive as the initial training phase, and in some cases exceeds

**Table 9:** Comparison of FLOPs between D3M and Detectron across computational stages.

| Algorithm | Dataset | Base Model Training | Calibration | **Total** |
|---|---|---|---|---|
| D3M | UCI | 0.23 GFLOPs | 9.00 GFLOPs | **9.23** GFLOPs |
| Detectron | UCI | 0.15 GFLOPs | 18.69 GFLOPs | **18.84** GFLOPs |
| D3M | CIFAR-10/10.1 | 1.56 PFLOPs | 0.13 PFLOPs | **1.69** PFLOPs |
| Detectron | CIFAR-10/10.1 | 1.56 PFLOPs | > 1.95 PFLOPs | > **3.51** PFLOPs |
| D3M | Camelyon17 | 12.02 PFLOPs | 0.74 PFLOPs | **12.76** PFLOPs |
| Detectron | Camelyon17 | 12.02 PFLOPs | > 11.04 PFLOPs | > **23.06** PFLOPs |

it. For example, on Camelyon17, calibration in Detectron adds over 11 PFLOPs on top of the 12 PFLOPs needed for training, while D3M incurs less than 1 PFLOP in calibration overhead.

These results highlight that as feature extractors scale in size, D3M's compute is dominated by the unavoidable cost of initial training, whereas Detectron continues to accrue significant additional cost in deployment. The difference grows more pronounced for large-scale datasets and larger-scale models, where Detectron effectively doubles its compute requirements, while D3M remains lightweight in its post-training stages. This distinction is especially important for real-world deployment settings, where calibration often needs to be performed repeatedly under strict resource constraints. In sum, FLOP comparisons show that D3M achieves up to 10x training reduction and 2x overall reduction versus Detectron, requiring no training data once the base model completes training.

### C.3 Temperature Sensitivity and ID Thresholds

We conduct an ablation study to assess the effect of the temperature parameter used in sampling from the VBLL layers. Figure 7 report results across UCI, CIFAR-10/10.1, and Camelyon17. Metrics include the false positive rate under ID data (FPR ID), true positive rate (TPR) for deteriorating OOD detection, and the mean disagreement rates on both ID and OOD samples.

Across datasets, we observe a consistent trend: increasing the temperature broadens the effective sampling distribution, which amplifies disagreement rates for both ID and OOD samples. While this increases diversity among hypotheses, it also skews the base ID disagreement rate distribution toward higher values. As a consequence, the separation between ID and OOD disagreement rates diminishes, leading to elevated FPR ID and reduced TPR for deteriorating OOD detection. For instance, on CIFAR-10/10.1, raising the temperature from 2 to 10 causes TPR to collapse from nearly perfect detection to just $0.34$, despite higher disagreement values overall. Similar trends are evident in UCI and Camelyon17.

We further plot the gap between the mean ID and OOD disagreement rates across our datasets and report in Figure 8. We find that as the temperature increases, the mean disagreement rates in- and out-of-distribution tend closer to each other. Since D3M's core functionality is to track this difference, as the temperature increases, ID and OOD become harder to distinguish on known deteriorating tasks, which explains why the TPR of D3M drops as temperature $\tau$ grows.

These findings highlight the critical role of temperature as an important tradeoff parameter where high temperatures blur the ID–OOD boundary and degrade detection reliability. In practice, we find that values in the range 1 to 5 offer the most stable trade-off across datasets, providing both sufficient hypothesis diversity and reliable threshold calibration.

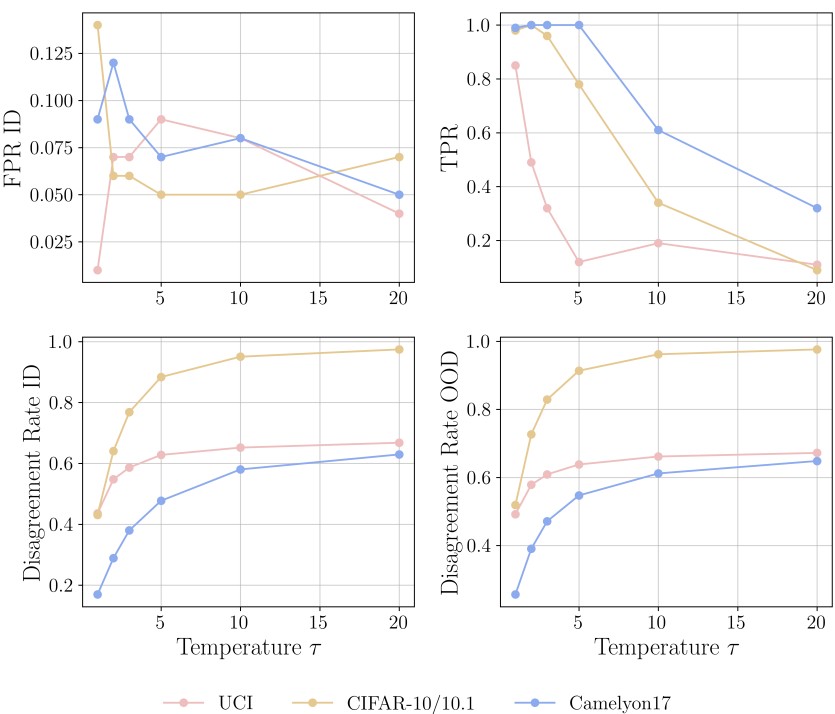

**Figure 7:** Ablation on the temperature parameter $\tau$ for D3M. We plot the ID FPR, TPR, ID mean disagreement rate, and deteriorating OOD mean disagreement rate across temperatures $\{1, 2, 3, 5, 10, 20\}$. We remark that as the temperature increases, softening the sampling logits distributions of D3M's Calibration and Deployment, while the ID FPR stays roughly within the same range, the TPR tanks significantly across all datasets.

Finally, we study the variation of the ID threshold during the Calibration step itself as ID disagreement rates measured on bootstrapped ID validation subsets get appended to $\Phi$. The ID threshold here refers to the $(1-\alpha)$ Quantile of $\Phi$, where disagreement rates beyond the ID threshold is labeled as deteriorating OOD by D3M.

**Table 10:** ID thresholds for $\alpha = 0.1$ across 10 independent runs. Columns are the rounds $t \leq T = 1000$ from which the $1 - \alpha$ quantile is computed for $\Phi$.

| Dataset | $t = 5$ | $t = 50$ | $t = 100$ | $t = 500$ | $t = T = 1000$ |
|---|---|---|---|---|---|
| UCI Heart Disease | $0.457 \pm 0.011$ | $0.468 \pm 0.007$ | $0.467 \pm 0.004$ | $0.468 \pm 0.004$ | $0.470 \pm 0.000$ |
| CIFAR-10/10.1 | $0.452 \pm 0.013$ | $0.464 \pm 0.009$ | $0.463 \pm 0.008$ | $0.465 \pm 0.005$ | $0.463 \pm 0.005$ |
| Camelyon17 | $0.306 \pm 0.011$ | $0.312 \pm 0.004$ | $0.312 \pm 0.004$ | $0.310 \pm 0.000$ | $0.310 \pm 0.000$ |

Table 10 shows that although some variability is exhibited for early $t \leq T$, the threshold stabilizes with more independent rounds, as evidenced by the decrease in standard deviation. Across our datasets, we observe that the ID threshold seems to increase with more independent realizations of Calibration rounds, as observed by the pronounced increment going from $t = 5$ to $t = 50$, but stabilizes beyond this point. A rough recommendation is to choose $T = |\Phi| \geq 50$, but we note that each round $t \leq T$ has the same wall-clock time as they all involve the same 1. bootstrapping a random ID validation subset with replacement from a ID validation set, 2. sampling $K$ candidate hypotheses, and 3. computing maximum disagreements across all candidate hypotheses.

## C.4 Oversampling of the ideal restricted hypothesis space $\mathcal{H}_p$

We investigate the gap between the empirical version of D3M and Idealized D3M (Algorithms 1 and 2). In particular, we investigate whether hypotheses sampled during the Calibration and Deployment phases of D3M belong in $\mathcal{H}_p = \{h \in \mathcal{H} : \mathrm{err}(h, \boldsymbol{P}_g) \leq \epsilon_f\}$, or that D3M oversamples $\mathcal{H}_p$ and

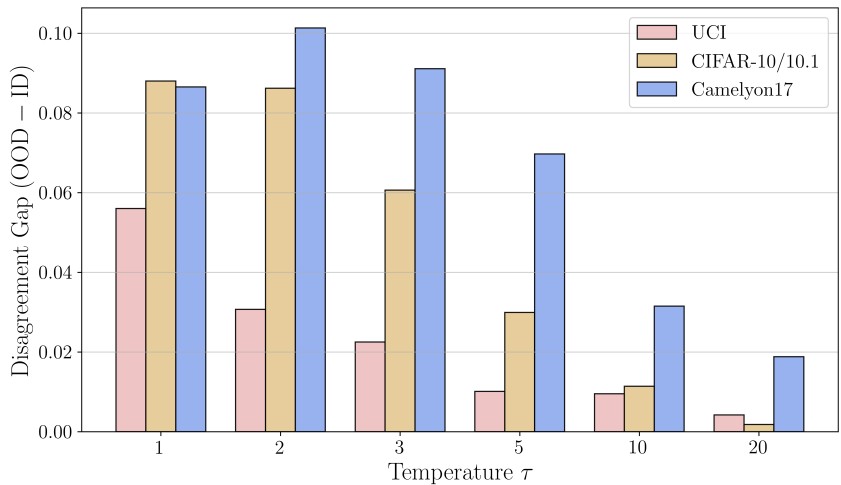

**Figure 8:** Evolution of ID–OOD disagreement gap over temperatures $\tau$. We find that the gap decreases monotonically (with the exception of a spike at $\tau = 2$ for Camelyon17) across datasets as $\tau$ increases. Since D3M's core functionality relies on distinguishing deteriorating OOD disagreement rates from ID disagreement rates, too high temperatures result in worse disagreement gaps, which lead to significantly worse TPRs.

obtains hypotheses beyond it. Our experiment setup is as follows. During Calibration, instead of sampling labels and computing maximum disagreement rates for bootstrapped unsupervised ID datasets, we sample labels for this bootstrapped set concatenated with the original training set along with the true training labels. For the Monte-Carlo sample resulting in the maximum disagreement rate (only computed on the bootstrapped set), we compute the accuracy against the true labels of the training set. For $|\Phi| = 1000$ independent runs on the UCI dataset and the CIFAR-10/10.1 datasets, we report the mean and standard deviation of training accuracy scores, the mean model's validation score, and the percentage drop.

**Table 11:** Comparison of accuracy of MaxDisRate hypothesis against the ID validation accuracy across datasets

| Dataset | MaxDisRate Acc. (Train) | Validation Acc. (ID Valid) | % Difference |
|---------|------------------------|----------------------------|--------------|
| UCI | $0.65 \pm 0.02$ | $0.76 \pm 0.02$ | $-11\%$ |
| CIFAR-10/10.1 | $0.55 \pm 0.00$ | $0.70 \pm 0.02$ | $-15\%$ |
| Camelyon17 | — | $0.94 \pm 0.01$ | — |

We remark that for no oversampling to occur, the sampled labels' training accuracy should be at least as good as the ID validation accuracy. Since we have significant drops on these above datasets in Table 11, we conclude that the sampled decision boundaries of D3M most likely lie beyond the desired set $\mathcal{H}_p$.

We further remark that a single sample of labels from a Bayesian model is not representative of the model's predictive capabilities as the latter relies on averaging over large samples of labels. We therefore make the preliminary assessment that models achieving the maximum disagreement rate are "oversampling", i.e. they do not fall in $\mathcal{H}_p$. In addition, the second sampling from the categorical distribution given by logits further derails our empirical D3M from the idealized D3M as with respect to the training data, this can be seen as a noisy assignment of labels for a training accuracy computation.

In contrast, Detectron [19] optimizes a joint loss term trading off between a training error minimization and a target unsupervised dataset's error maximization (with respect to its pseudolabels). As this tradeoff is tunable, it's possible to enforce stronger adherence to hypotheses within $\mathcal{H}_p$ at the heavy price of requiring the training set to be at arm's length.

Given this significant deviation from the idealized setup where candidate hypotheses are obtained following an optimization oracle returning hypotheses only within $\mathcal{H}_p$, to what extent can one trust the verdicts outputted by D3M? Intuitively, D3M is ID-oversampling "by the same amount" during calibration and deployment, akin to a *miscalibrated balance scale*. Oversampling on an ID deployment sample would result in disagreement rates resembling those in $\Phi$, thus incurring no more false positives. For a deteriorating OOD deployment sample, the effect of oversampling exacerbates disagreement rates but only to the extent that it exacerbates disagreement rates for bootstrapped ID samples during the calculation of $\Phi$. In contrast, oversampling might result in a wider dispersion of $\Phi$, meaning the $1 - \alpha$ quantile might be pushed closer to 1, resulting in more false negatives.

Our analysis reveals that D3M consistently samples hypotheses beyond the ideal restricted hypothesis space $\mathcal{H}_p$, with training accuracies falling 11-15% below validation performance across datasets. Despite this departure from the idealized framework, D3M operates as a "miscalibrated balance scale" that maintains relative consistency between calibration and deployment phases. The systematic nature of this oversampling means both ID and OOD samples are subject to the same bias, potentially preserving discriminative power while increasing variance in $\Phi$ and risking higher false negative rates.

### C.5 Comparisons with a fully Bayesian network

We hereby provide an additional ablation comparing D3M's performance against a fully Bayesian network. Bayesian networks offer the most principled approach to approximating the idealized D3M (Algorithms 1 and 2) when viewing optimization in $\mathcal{H}_p$ as sampling from a distribution over hypotheses modeled by said network. Given that D3M's sampling is only restricted to the last layer, it follows that D3M's sampled hypotheses are less diverse than their fully Bayesian counterparts.

Without (double) sampling once again from the categorical distribution generated by the sampled logits, the maximum disagreement rates achieved across 10000 runs hover around 5% for both ID and deteriorating OOD samples, making them nearly indistinguishable by D3M. Instead, sampling candidate disagreeing predictions from softmaxed and temperature-scaled logits rather than argmaxing allows us to bring the ID disagreement rates in $\Phi$ to around 30%-50%, where deteriorating OOD samples often score 5%-10% higher, making them distinguishable. How would a fully Bayesian model fare? The following table compares the mean ID disagreement rates of a fully Bayesian (FB) model and a VBLL model (D3M), all using the same set of best hyperparameters for each dataset.

**Table 12:** Comparison of mean ID disagreement rates in $\Phi$ of fully Bayesian and VBLL models.

| Dataset | Mean ID DisRate (FB) | Mean ID DisRate (VBLL) | FB Accuracy |
|---|---|---|---|
| UCI | $0.30 \pm 0.04$ | $0.44 \pm 0.01$ | $0.79 \pm 0.02$ |
| CIFAR-10/10.1 | $0.36 \pm 0.01$ | $0.42 \pm 0.03$ | $0.70 \pm 0.02$ |
| Camelyon17 | — | — | — |

We remark that using the mean model of a FB network as the predictive function does not trade off performance as shown in the last column. Further, we observe that even without the double sampling scheme used for VBLL, the mean ID disagreement rate of FB does not collapse into $0.05$ as is usually the case with VBLL without any adjustments.

In Figure 9, we compare the fully Bayesian neural network's decision against various instances of VBLL neural networks. Indeed, we find that when no double sampling is used, decision boundaries tend to collapse to the mean function. This is evidenced in very low disagreement rates for VBLL compared to fully Bayesian models. With double sampling, however, we observe that predictions are more stochastic, leading to increasingly more likely disagreement with respect to the mean model. This illustrates how we may approximate the boundary/decision diversity of a fully Bayesian network with VBLL and additional double sampling.

Notably, on UCI, while VBLL trains each element $\Phi$ in approximately 1 second, a FB base model in D3M takes approximately 17 minutes. The gap worsens for CIFAR-10/10.1 as it takes FB and hours for CIFAR-10/10.1 while only 2 minutes for VBLL. This example further showcases the computational overhead incurred when sampling iteratively rather than in parallel. One either reparametrizes layers, making each forward pass's output essentially a sample of logits, requiring

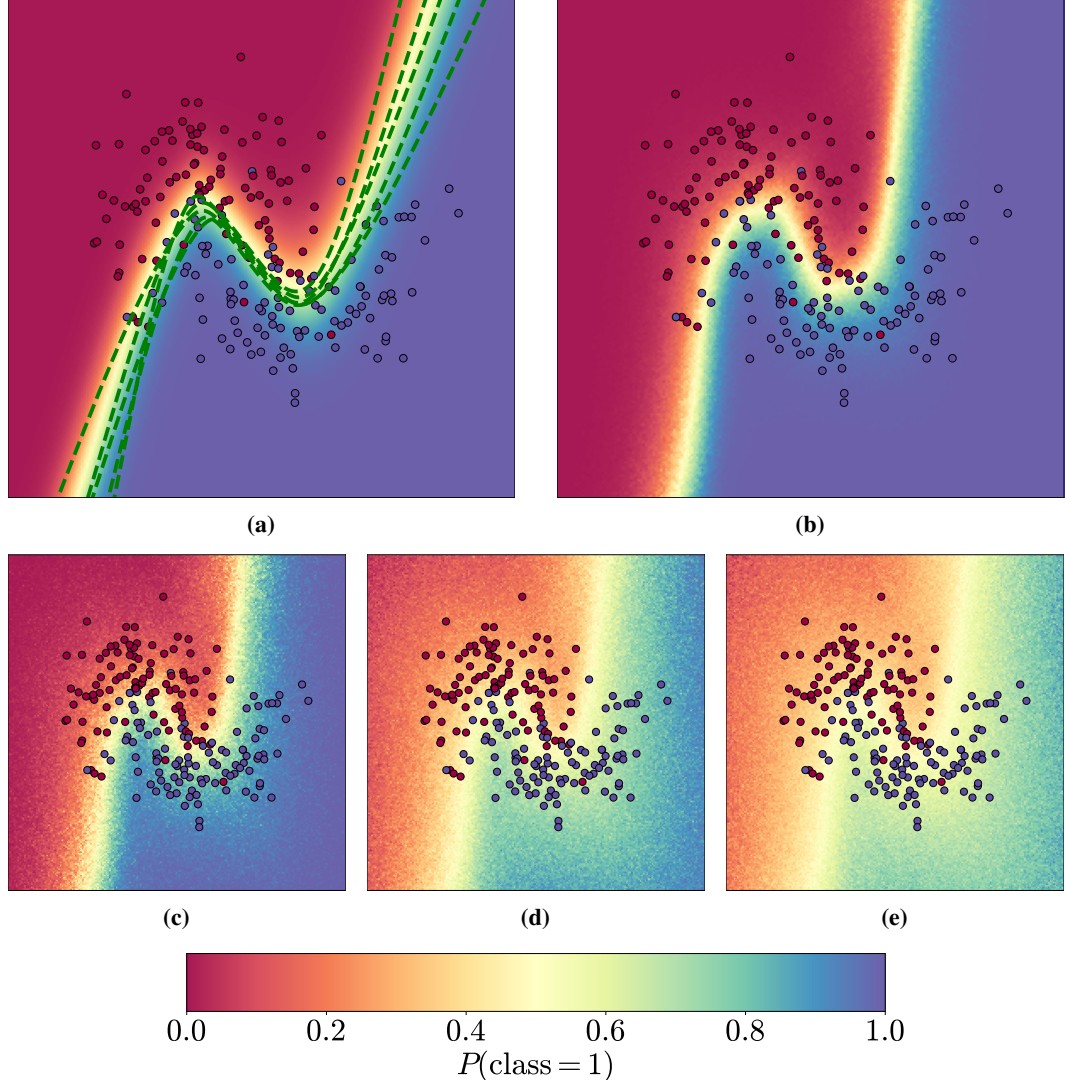

**Figure 9:** Toy experiment on the Two Moons dataset generator [69]. 500 points are noisily sampled from each class at noise level 0.3. We visualize 200 total samples here for clarity. (**Figure 9a**) A fully Bayesian neural network is trained on this dataset. The decision boundaries correspond to the level set of the probabilities of predicting either class at level 0.5. Forward passing the full grid through this net, we find different decision boundaries pictured in green. (**Figure 9b**) A comparable VBLL model is trained. Without our double sampling scheme, most decision boundaries samples collapse onto the mean model's boundary, highlighted by the yellow region. **Figures 9c**, **9d**, and **9e** depict probabilities under double sampling with temperatures $\tau = 2, 5, 10$ respectively. We observe that sampling predictions from gradually softer probabilities give noisier estimates across the space. Although this does not provide a concrete continuous level set at 0.5, the colors depict the distribution of predictions from which D3M samples, which "simulates" the diversity of decision boundaries.

sequential forward passes to collect MC logits, or one must lift all weight tensors to include a sample dimension and ensure every downstream operation — including residual connections, batch norm, and other architectural components — is broadcast-compatible.

While this is realizable, it offers no benefits to our two-stage sampling strategy. We find that when temperatures are tuned to yield ID disagreement rates in the 30% to 50% range, deteriorating OOD disagreement rates are easily discernable as on those inputs, D3M tends to disagree 5%-10% better, thus easily achieving high TPR.

