# OpenReview forum: "Reliably detecting model failures in deployment without labels"
_NeurIPS.cc/2025/Conference — NeurIPS 2025 poster_

### Official Review · Reviewer_88gc · 2025-06-22

**Clarity:** 1
**Significance:** 3
**Originality:** 3
**Rating:** 4
**Confidence:** 3

**Summary:**

This paper proposes a method to determine whether a model has suffered from "post-deployment deterioration" (PDD). That is, the joint distribution of the covariates $x$ and labels $y$ have shifted such that the model's performance is worse. The paper distinguishes this from out of distribution (OOD) detection in that OOD detection does not ask whether or not the performance of the model has become worse. The method takes inspiration from the model disagreement literature, which checks if multiple models make different predictions on a test point. Here, the "multiple models" come from adding a "Variational Bayesian Last Layers" to an existing model's output layer (i.e., the model now outputs a distribution over logits). This distribution over logits is sampled from, passed through a softmax, and then the resulting probability distribution is sampled from with an increased temperature to get a prediction. The number of these sampled predictions that disagree with the base-model's argmax prediction is the "model disagreement". This process is repeated for a number of training datapoints to get a sense for typical model disagreement for in-distribution data. Model disagreement is computed the same way for input test data, and if the disagreement is higher than typical for in-distribution data, the test data are flagged. In experiments, the paper shows the proposed method to have often higher true positive rate (TPR) than similar methods with the same false positive rate (FPR); however, the TPR is fairly noisy across different random seeds. In the appendix, an "idealized" version of the algorithm is proposed and its theoretical properties (e.g., FPR and TPR related to its input parameters) are studied.

**Questions:**

1. How does the idealized algorithm that the theory is about relate to the actual algorithm? How much should users trust the theoretical results to apply to the real algorithm?

2. When will the algorithm fail? Are there any indicators or situations that users should look out for?

3. How can users reduce the variability of the proposed method that's seen in, e.g., Table 2? What could lower this variance and why?

4. The paper seems to propose waiting to make PDD predictions until a large batch of test points has been accumulated. Why is this realistic? How would this work in time-sensitive situations?

Thank you!

**Ethical Concerns:**

["NO or VERY MINOR ethics concerns only"]

**Final Justification:**

I've increased my score from a 2 to a 4 (weak accept) after the discussion period. I think the authors have addressed essentially all of the issues I raised, and I think they've also addressed other reviewer's comments well.

I've only increased to a weak accept rather than an accept because I think the proposed experiment showing oversampling isn't air-tight. In particular, I think this only shows that the sampling method can generate points that are outside the theoretical $\epsilon$ ball that we want. But I think what we really want is to show that the method generates points completely covering the $\epsilon$ ball. To be fair, though, the proposed experiment gets at *suggesting* this coverage, and it's not like I have any great ideas on how one would show this more precisely.

But overall, I vote for accepting the paper.

Best,
88gc

**Limitations:**

I don't think the main text included much discussion of the proposed method's limitations besides pointing out the fact that its performance can be noisy -- the "Conclusions and Limitations" section actually mostly summarizes the strengths of the proposed method. As noted above, I would suggest really digging into the intuition behind the proposed method so that its failure case(s) can be better understood by readers.

Further, I think it's important to stress that the PDD task is fundamentally impossible without labels. If the distribution of $(x,y)$ changes such that $y\mid x$ is totally different but the distribution of $x$ is the same, there's no way to detect this from labels. So I think it's really important to make the assumptions this paper is making more clear and emphasize how reality might come into conflict with those assumptions.

**Quality:**

2

**Strengths And Weaknesses:**

# Weaknesses

I've broken down comments here into various sections below.

## What is the proposed method doing?

I overall think it's not clear what this algorithm is really doing and why it should work. And I think it's really important when proposing new methods to give some kind of scientific understanding of the methods (unless the empirical results are so phenomenal that the research community just needs to know about this new method without really knowing anything about it, but I don't think that's the case here). I don't think the paper is currently meeting this standard, which is the main influence on my score. A few specific comments on this below:

1. In the main text of the paper, I didn't see any discussion of why readers should think the proposed method would work. It has the flavor of existing OOD algorithms, but it's supposed to (1) detect OOD points, and (2) detect if those points are going to give bad performance from the model. I didn't really see why we would expect (2) to hold. Overall, I think there needs to be a lot more discussion of the intuition behind the algorithm in the paper.

2. The theoretical discussion in the supplement has some intuition of why the proposed algorithm will work. This development proves conditions under which an "idealized" version of the algorithm will work well. I think this is really great, but could use some expansion. First, the idealized version of the algorithm is pretty idealized, and it's not clear how connected it is to the real version of the algorithm. The idealized version assumes access to all other models that have training error within $\epsilon$ of the base model. I think the paper is arguing that this is approximated by some of the Bayesian sampling and increased temperature. But why does this approximate this model set well? The paper briefly implies (line 1010) that the Bayesian sampling and temperature will *over-approximate* the size of this model class, but it's not clear to me that this is true. I think careful exploration of the connection between the idealized and practical algorithm is needed to finish the story of this paper. Second, I really think some of the theoretical development needs to be moved to the main text, as it's what ties the entire paper together.

3. There are no experients that get at the heart of what's going on with the algorithm. I think synthetic experiments are an excellent opportunity to really probe what's going on behind the scenes of an algorithm -- when should we expect the algorithm to fail or succeed? How well does the intuition behind the algorithm hold up empirically? Carefully designed synthetic experiments give the opportunity to answer all of these questions, so I would really recommend thinking about how that could be done for the current paper.

## Theoretical results

I thought that two potentially impractical assumptions were made in the theory.

**Assumptions**
- The theoretical results hinge on assuming the model class is correctly specified, which is a major assumption that is rarely true in practice.
- Assumes that $m$ datapoints are available at once to make a decision of prediction on. Again, it's not clear to me how practical this setting is.


**Clarity**
There were a few places towards the beginning of the theory where I found things either underdefined or just a little unclear. These are mostly towards the beginning of the theoretical development, as I started skimming towards the end as these built up.

- In Lemma A.1: $h$ was not defined. Additionally, the proof of the lemma states that $err(f, Q_g) > err(f, P_g)$, where $Q$ is the deployment distribution and $P$ is the training distribution. Why must this be true given the assumptions of the theorem? What if $Q$ is a point-mass at a point that $f$ happens to get correct? And below line 970, a sup is taken over $A$ (I think sets $A$?) without saying what $A$ varies over. Finally, the proof starts by stating it is going to show that D-PDD implies PDD, but it never shows the reverse direction, despite the statement of the Lemma saying that PDD also implies D-PPD.
- Definition 4 states that the algorithm detects when $\epsilon_q - \epsilon_p > 0$, then Definition 5 states that the algorithm detects when $\xi \geq 0$, and then the line after Definition 5 states that $\xi \geq \epsilon_q - \epsilon_p$. So these quantities aren't the same, so how can it be that the algorithm is simultaneously thresholding both?
- Definition 5 states that $\xi \geq 0$. But why? Does this assume $f$ is the minimizer of $err(f; P_g)$? I don't think this was stated.
- Eq (6) uses the notation $Q_{1-f}$, but I don't think the $1-f$ notation was ever defined.
- Thm A.3 proves a bound on the FPR of the algorithm. The discussion around it implies that it bounds the FPR by $\alpha$, the "significance" level of the algorithm. But I don't think the theorem proves this. In particular, it states that the FPR can be bounded by $\alpha + (1-\alpha)e^{-n\epsilon_0^2 + d}$, where $d$ is the VC dimension of the hypothesis class. If $d$ is huge, then I'm pretty sure this bound can be arbitrarily larger than 1, whereas $\alpha < 1$.
- Towards better connecting the theory to the main paper: Theorem A.3 gives a required batch size $m$ to achieve a bounded FPR. It would be really interesting to evaluate $m$ for typical ML models and see if the result is practical. Is this telling us we need to batch up 94303 samples before making an effective decision about domain shift? Or is it saying we only need 20?


## Experiments
1.  The experiments are run in "{10,20,50}-shot scenarios". I wasn't totally clear on what this means. Does this mean that the model gets to see a batch of, e.g., 10 test datapoints before guessing whether there's model drift? It seems like this runs counter to the run in real time idea of the paper. In practice, I'd imagine that we want to make predictions one at a time. And then we have to make a "domain shift yes/no" guess when each datapoint is observed. How does the current experimental setup fit into this picture? Or are there really scenarios where we can wait to make a prediction until we've seen 50 test points?

2. In the entire experiment depicted in Figs 2 and 3, it seems like the existing BSSD method is outperforming the proposed method. This doesn't seem to be discussed in the paper -- so what is this experiment showing that's good or promising about the proposed method?

3. Figure 2b seems to be showing really bad performance of the proposed method, as it's true positive rate is nearly zero across the board. The text seems to indicate that this is a good thing because there's actually not much domain shift happening here. But in that case shouldn't there be no true positives at all? How are other methods getting high true positive rates if there is nothing to correctly flag?

4. The x-axis in Fig 3 isn't described. I'm guessing this is an amount of domain shift, but it's not clear to me what the numbers mean.



## Overall Comments
1. Some references to specific Appendicies seem to be pointing to the wrong place (e.g., saying that baselines are described in B.4, but they're actually described in B.1). Please check the appendix references to make sure they're all pointing to the right place.
2. "D3m does not trade off prediction accuracy" -- it does require the addition of a Bayesian head to the model, which I would assume could affect the accuracy.
3. $\tau$ was never defined in the main text
4. The introduction makes it sound like there are few restrictions on the model class, but the paper seems to be entirely about classification models.
5. I think the claim that "we answer all desiderata" it misleading, as one of the disiderata is that the method has a low FPR. I strongly suspect it is impossible to guarantee this in practice, but this claim makes it sound like the proposed method completely solves this issue.

# Strengths

I think with a properly motivated algorithm (along the lines of the theory in the appendix), this paper has the potential to be very strong. Proving conditions under which not only OOD detection is possible but a practically meaningful form of OOD (when the model performance is actually worse) is very impactful for applications, and I think this paper has the possibility to not only do that, but also give an explicit algorithm for doing so. I think it's especially strong that the algorithm has a mechanism to explicitly control its false positive rate, which I think will be important in practical settings; if I had to guess, I would say that hospital staff are going to be pretty susceptible to alarm fatigue from a black box ML system, so setting the FPR to be guaranteed low is important.

---

> ### Author Rebuttal · Authors · 2025-07-31
>
> Thank you very much for your very thorough dissection of our paper and insightful comments! We appreciate that you find our algorithm is “properly motivated”, with “potential to be very strong”. We hope to address your concerns and questions in the following.
>
> # Weaknesses
>
> > “I didn't see any discussion of why readers should think the proposed method would work [...]”.
>
> > The theoretical discussion in the supplement has some intuition of why the proposed algorithm will work. [...]
>
> Thank you for highlighting this critical point. The underlying intuition is that given a base classifier model, the disagreement rate is a quantity that differs in and out of distribution only when the base model will underperform on a OOD batch. The reason we track this discrepancy is due to Definition 2, Line 949, 3rd inequality $err(h;Q_f) > err(h; P_f)$ where the algorithm’s calibration step is generating empirical values for the RHS, while the Deployment step compares an empirical estimate of the LHS to an empirical distribution of $err(h; P_f)$. If we believe Definition 2 is sound, then Idealized D3M works because it’s designed to empirically track D-PDD, while D3M is a crude approximation.
>
> We agree that sampling over-approximates the set of interest. It’s likely that our sampling produces hypotheses that would’ve achieved higher error than the base classifier on the training set. Empirically, we find that although this may be the case, the strength of disagreement is more significant on deteriorating OOD data. Guarantees on significant disagreement gaps between ID and OOD despite sampling beyond the set of interest are currently beyond what the theory provides.
>
> > some of the theoretical development needs to be moved to the main text, as it's what ties the entire paper together [...]
>
> We agree with your assessment of the importance of the theoretical binds. For this reason, we are editing in a theoretical insights subsection along with an informal definition of D-PDD as well as an informal lemma linking D3M to the tracking of D-PDD, while referring readers to the Appendix for a more formal treatment.
>
> > There are no experients that get at the heart of what's going on with the algorithm. I think synthetic experiments are an excellent opportunity to really probe what's going on behind the scenes of an algorithm  [...]
>
> Thank you for raising this point! We could demonstrate a proof of concept for the elevated OOD disagreement rates by the following experiment: we learn a posterior over decision boundaries on the synthetic Two Moons dataset in 2D. For ID and OOD samples, we illustrate how likely it is to sample a decision boundary that disagrees on those samples with respect to the mean decision boundary, building intuition for the discrepancy in disagreement rates.
>
>
> > The theoretical results hinge on assuming the model class is correctly specified, which is a major assumption that is rarely true in practice.
>
> Indeed, this is an important limitation with regards to results. However, we encourage readers to view the theory as a perspective on how one might go about modeling the deteriorating OOD problem, as well as the insights insights which we derive and track directly in D3M (tracking $err(h;Q_f) > err(h;P_f)$, for instance).
>
> > Assumes that  datapoints are available at once to make a decision of prediction on. Again, it's not clear to me how practical this setting is.
>
> > The paper seems to propose waiting to make PDD predictions until a large batch of test points has been accumulated. Why is this realistic? How would this work in time-sensitive situations?
>
>
> An example of a practical setting would be D3M undergoing a silent trial at a hospital network, where patient data come in sequentially to fill up a batch before being processed with D3M. This is in contrast to paradigms of decision making where a decision is expected after an observation is recorded.
>
> > In Lemma A.1:  was not defined. [...]
>
> Our revised manuscript will provide the definition of $h$ following Definition 2.
>
> Regarding $Q$ being a point mass, this violates the TV condition unless $P$ already concentrates nearly all of its mass on it. Our lemma explicitly assumes a bounded TV shift and a nontrivial disagreement gap, ensuring that the difference in disagreement cannot be solely attributed to small distributional changes.
>
> We edited the supremum to be over $A \in \mathcal{F}$ of the underlying $\sigma$-algebra.
>
> We respectfully note that the reverse direction (D-PDD → PDD) is indeed concluded: the final inequality shows that if the disagreement gap satisfies $err(f,Q_h​)−err(f,P_h​)\geq 2(\kappa+\epsilon)$, then it necessarily implies $err(f, Q_g​)> err(f, P_g​)$, which is precisely the PDD condition under $g=g’$.
>
> > Definition 4, 5, Eq (6)
>
> Our manuscript has been revised with the suggested edits and clarifications.
>
> > Thm A.3 proves a bound on the FPR of the algorithm.
>
> > Towards better connecting the theory to the main paper
>
> We appreciate the engagement with Theorem A.3. Indeed, the bound derived there is likely loose and primarily serves as a proof of concept—a theoretical guarantee that, in principle, our method can control the FPR under distribution shift. While the bound scales with the VC dimension of the hypothesis class and may be vacuous for large models, its main role is to motivate and justify the design of our algorithm. Empirically, we find that our method performs well with small batch sizes, suggesting that the theoretical sample complexity is conservative but directionally aligned with practice.
>
> ## Experiments
>
> > The experiments are run in "{10,20,50}-shot scenarios" [...]
>
> As per our discussion with Reviewer **Ktyz**, these are the sizes of the subset of data receiving a verdict from D3M. Importantly, methods in the literature typically give a verdict per point, whereas our setup’s goal is to give a verdict per batch or per data subset. We envision D3M as a monitor to a predictive model in deployment rather than a decision-making agent. Inputs on which the model predicts are bundled (one may imagine a moving-window of the last $m$ samples). A D3M flag would signify that a corrective action is in order, i.e. a re-calibration, a fine-tuning, a domain-adaptation modification, or even a potential retraining.
>
> > BBSD [...]
>
> BBSD is indeed outperforming our model on the IM dataset for the age shift scenario. We suspect it is due to a strong covariate shift, where BBSD is known to perform really well. As for why other distance-based monitors such as MMD-D and Rel-MH aren’t achieving the same performance, we suspect they might be overfitting their representations to the training set.
>
> > Figure 2b [...]
>
> Thanks for catching this! The y-axis is supposed to be labeled FPR@5%, our manuscript is since fixed. Distance-based learners are particularly susceptible to this type of FPR since they pick up on covariate shifts. We argue based on Figure 2a that this scenario is not a deteriorating shift, and therefore good monitors should resist flagging this change.
>
> > The x-axis in Fig 3 [...]
>
> Thank you for the catch! These are the mixing coefficients between patients of different age groups as described in Lines 1315-1323. We will expand Fig3’s caption to be more comprehensive.
>
> ## Overall Comments
>
> > Some references to specific Appendicies seem to be pointing to the wrong place
>
> Thank you very much! We have proof-read our manuscript carefully.
>
> > "D3m does not trade off prediction accuracy" [...]
>
> The classification accuracy is affected but only so slightly. Please refer to corresponding section of the response to Reviewer **TFQc** for a closer look.
>
> > I think the claim that "we answer all desiderata" it misleading [...]
>
> Our empirical observations show that a low FPR is indeed achieved in practice. Intuitively, if the deployment sample came from the same training distribution, then its disagreement rate is beyond the $1-\alpha$ quantile with probability $\alpha$. In practice, FPRs may be slightly higher due to $\Phi$ being an empirical estimate and further approximation errors.
>
> > $\tau$
>
> Thank you for pointing this out. We have included its definition.
>
> # Questions
>
> > How does the idealized algorithm that the theory is about relate to the actual algorithm? [...]
>
> As per the above and as you have commented, the Bayesian sampling essentially over-approximates beyond the confines of the subset of hypotheses of interest. Another alternative is to try to optimize for $h$ directly via gradient descent with access to the training dataset in a similar fashion to Detectron (See **TFQc**). Users of D3M should view the theoretical results as **qualitative guidance rather than quantitative guarantees**, since the idealized D3M with idealized assumptions (mainly optimization oracle) do not fully hold in practice.
>
> > When will the algorithm fail? Are there any indicators or situations that users should look out for?
>
> Empirically during testing, we found that when the temperature is high, while the ID FPR remains consistent empirically, the deteriorating OOD TPR is heavily impacted (See Response to Reviewer **Ktyz**).
>
> Theoretically, a regime that our theory isn’t equipped to handle is Regime 2 (Line 1110), where we analyze a toy example and propose that well-trained base classifiers (low generalization error) are less likely to fall into this situation.
>
> > How can users reduce the variability [...]
>
> We found that lowering the temperature below 1 helps reduce variance significantly at the cost of TPR, and are looking for hyperparameters and perhaps design changes that would rectify this.
>
> # Closing Remarks
>
> Thank you once again for your helpful, attentive, and detailed feedback. We hope that our rebuttal addresses your questions and concerns, and we kindly ask that you consider a fresher evaluation of our paper if you are satisfied with our responses. We are also more than happy to answer any further questions that arise.

---

> > ### Comment · Reviewer_88gc · 2025-08-06
> >
> > Thank you for the replies to all the reviews I think moving some of the theoretical discussion / results to the main text will be beneficial. In particular, the flow of "in theory, when the model is correct and we have perfect access to all other models with error rate within $\epsilon$ of our model, our method works; in practice over-sample the set of within-$\epsilon$-error; and here we show it working in experiments" is compelling. I just have two further questions to further develop this chain of logic:
> >
> > 1. Sorry for not being more clear on this, but on the Bayesian oversampling: I meant to ask *why* we should expect the proposed method to be oversampling. Is there evidence of this in the literature? Or an experiment(s) that could demonstrate this?
> > 2. Supposing that we are oversampling, what is the consequence of this oversampling? Can this consequence be phrased in terms of getting a higher false negative rate or false positive rate? I don't think super deep experiments are needed to justify this; I just think some argument about why it's OK to oversample is needed, since that seems to be a key point in the logic of the proposed method.
> >
> > I think having some supporting evidence for both these points is the final key step to flesh out the chain of logic supporting the proposed method.
> >
> > Thank you!

---

> > > ### Author Response · Authors · 2025-08-07
> > > **Response to Reviewer 88gc**
> > >
> > > Thank you once again for your time and effort put into the reviewing of our submission and rebuttal, we greatly appreciate it! We hope to address your final comments and suggestions in the below.
> > >
> > > > In particular, the flow of "in theory, when the model is correct and we have perfect access to all other models with error rate within  of our model, our method works; in practice over-sample the set of within--error; and here we show it working in experiments" is compelling.
> > >
> > > Thank you for this suggestion. We agree that this chain of reasoning makes our algorithm more compelling. For this reason, we shall make this clear in the theoretical insights subsection discussed in our rebuttal above.
> > >
> > > > I meant to ask why we should expect the proposed method to be oversampling. Is there evidence of this in the literature? Or an experiment(s) that could demonstrate this?
> > >
> > > Thank you for your clarifications. We report results of the following ablation setup on UCI Heart Disease and CIFAR-10 (the setup for Camelyon17 is too computationally expensive to generate enough independent runs for confidence calculations within the given time frame). In the following, let $\mathcal{H}_p$ denote the subset of models within $\epsilon$-error of interest. During calibration (generation of $\Phi$) instead of sampling labels and computing maximum disagreement rates for bootstrapped unsupervised ID datasets, we sample labels for this bootstrapped set concatenated with the original training set along with the true training labels. For the MC sample resulting in the maximum disagreement rate (only computed on the bootstrapped set), we compute the accuracy against the true labels of the training set.
> > >
> > > For $|\Phi| = 1000$ independent runs and each dataset, we report the mean and standard deviation of training accuracy scores, the mean model’s validation score, and the percentage drop.
> > >
> > > | Dataset         | MaxDisRate Acc. (Train) | Validation Acc. (ID Valid) | % Difference |
> > > |----------------|--------------------------|-----------------------------|--------------|
> > > | UCI            | 0.65 ± 0.02              | 0.76 ± 0.02                 | -11%         |
> > > | CIFAR-10/10.1  | 0.55 ± 0.00              | 0.70 ± 0.02                 | -15%         |
> > >
> > > We remark that for no oversampling to occur, the sampled labels’ training accuracy should be at least as good as the ID validation accuracy. Since we have significant drops on these above datasets, we conclude that the sampled decision boundaries of D3M most likely lie beyond the desired set $\mathcal{H}_p$.
> > >
> > > We further remark that a single sample of labels from a Bayesian model is not representative of the model’s predictive capabilities as the latter relies on averaging over large samples of labels. We therefore make the preliminary assessment that models achieving the maximum disagreement rate are “oversampling”, i.e. they do not fall in $\mathcal{H}_p$.
> > >
> > > In addition, the second sampling from the categorical distribution given by logits further derails our empirical D3M from the idealized D3M as with respect to the training data, this can be seen as a noisy assignment of labels for a training accuracy computation.
> > >
> > > In contrast, Detectron [1] optimizes a joint loss term trading off between a training error minimization and a target unsupervised dataset’s error maximization (with respect to its pseudolabels). As this tradeoff is tunable, it’s possible to enforce stronger adherence to hypotheses within $\mathcal{H}_p$ at the heavy price of requiring the training set to be at arm’s length. In our work, we argue that this is heavily undesirable especially for the deployment stage. This comparison will be detailed in a new Appendix section dedicated to ablations requested by all reviewers of this work.
> > >
> > > > Supposing that we are oversampling, what is the consequence of this oversampling? Can this consequence be phrased in terms of getting a higher false negative rate or false positive rate?
> > >
> > > We believe that intuitively, D3M is ID-oversampling “by the same amount” during calibration and deployment, akin to a miscalibrated balance scale. Oversampling on an ID deployment sample would result in disagreement rates resembling those in $\Phi$, thus incurring no more false positives. For a deteriorating OOD deployment sample, the effect of oversampling exacerbates disagreement rates but only to the extent that it exacerbates disagreement rates for bootstrapped ID samples during the calculation of $\Phi$.
> > >
> > > However, we suspect oversampling might result in a wider dispersion of $\Phi$, meaning the $1-\alpha$ quantile might be pushed closer to 1, resulting in more false negatives.
> > >
> > > Once again, thank you for your response and for taking the time to reconsider your evaluation! We truly appreciate your support and are glad our clarifications were helpful!
> > >
> > > ## References
> > >
> > > [1] Ginsberg, T., Liang, Z., & Krishnan, R. G. (2022). A Learning Based Hypothesis Test for Harmful Covariate Shift. arXiv, 2212.02742.

---

> > > > ### Comment · Reviewer_88gc · 2025-08-07
> > > >
> > > > This extra experiment seems great! I would recommend including this experiment in the paper, along with adding some discussion of the consequences of oversampling.
> > > >
> > > > To me, this completes the chain of logic motivating / justifying the proposed method, so with this included, I've raised my score to vote for acceptance. I'll write in some final comments into the final justification box.

---

> > > > > ### Author Response · Authors · 2025-08-07
> > > > > **Response to Official Comment by Reviewer 88gc**
> > > > >
> > > > > Thank you so much!! We will be sure to include these results in the paper with discussions related to our claims of oversampling. We greatly appreciate your help in refining the chain of logic for D3M, and we're especially grateful for your thoughtful guidance throughout the review process. We now feel the paper presents a much more coherent and complete story, thanks to your insights.
> > > > >
> > > > > Thank you once again for taking the time to reconsider your evaluation, we truly appreciate your support!

---

### Official Review · Reviewer_TFQc · 2025-07-02

**Clarity:** 2
**Significance:** 2
**Originality:** 3
**Rating:** 4
**Confidence:** 3

**Summary:**

This paper proposes methodology, D3M, for monitoring distribution shifts that may lead to performance degradation of deployed classifiers.

The central claim is that D3M addresses two desiderata that are not satisfied by any single prior method: (1) no need to access training data at deployment time and (2) only flag *deterioriating* shifts.

The method works by leveraging uncertainty estimates from a Bayesian neural network and monitoring for high variance/disagreement in the predictions.

The D3M method is competitive with Detectron (which requires training data at deployment time), though trails by substantial margins on some datasets. They also claim in writing that D3M has a number of efficiency benefits over alternative approaches.

**Questions:**

- I’m confused by the columns in Table 1. How can D3M be both deteriorating and non-deteriorating?
- How does the use of VBLL compare with other Bayesian approaches?
- Other typos and clarifications:
    - There is a duplicated “operating operating” in the first sentence of the abstract.
    - “Underlines in our results indicate that D3M is close to the best baseline.” What is “close” here? It seems somewhat arbitrary given the differences in performance and nearly 10 points in some cases.
    - Please, clarify what exactly an N-shot scenario is. Also explain that the {10, 20, 50} in Table 2 correspond to shots in the caption.

**Ethical Concerns:**

["NO or VERY MINOR ethics concerns only"]

**Final Justification:**

Prior to the rebuttal, my main concern was that the paper made several claims regarding the efficiency and scalability of the method that were not substantiated quantitatively. I think this is a major concern given that the proposed method's only advantage over prior work (e.g. detectron) is scalability, not quality. The author's response provided some quantification in terms of training and calibration FLOPs. So I raised my score, assuming that these will be integrated into the final results.

**Limitations:**

Yes

**Quality:**

2

**Strengths And Weaknesses:**

**Strengths**

- The problem addressed in this paper (monitoring for distribution shifts) is very important in safety critical settings like medicine. The paper evaluates on interesting and realistic benchmarks from the medical domain.
- The proposed methodology has at least one advantage over prior work like Detectron: it does not requiring training data at deployment time.
- The method is competitive with prior work like Detectron.

**Weaknesses**

- In Section 2.3, the authors make a number of claims about D3M without concrete evidence or experiments:
    - “D3M does not trade-off prediction accuracy.” To support this claim, the authors should provide some experiments demonstrating that the prediction accuracy with D3M is indeed equivalent to or better than the prediction of accuracy of other classifiers. However, no such experiments or results are included in the paper.
    - “Potential for edge deployment.” The authors claim that D3M is particularly well-suited for edge deployments but do not provide any analysis of edge hardware and how their method will map onto it. Without a concrete analysis, this claim is speculative.
    - “While VBLL allows more efficient sampling compared to to a fully Bayesian network, the price to pay is the lack of diversity of the sampled predictions.” To support this claim, I would expect to see a comparison of the diversity of predictions when using VBLL and a fully Bayesian network. Without it, I cannot judge the importance of the proposed remedies (*e.g.* temperate scaling).
- In Section 3.2, the authors claim that “D3M is significantly more scalable and practical for real-world deployment [than Detectron]”. While it is a clear advantage that D3M doesn’t require training data, the authors do not provide any concrete evidence of the scalability of D3M and Detectron. I would recommend that the authors include some quantitative measure of cost (*e.g.* FLOPs) that demonstrates that D3M is indeed more scalable than Detectron.
- The presentation of the results on the IM dataset [Temporal Shift] could be improved.
    - I’m quite confused by the Y-axis in Figure 2b. The label says TPR@5%, but this is in a setting with only a non-deteriorating shifts, so the true positive rate is undefined?
    - The legend in Figure 2 and Figure 3 could be improved. I’m not sure what some of the abbreviations correspond to. For example, is DC the abbreviation for detectron? **

---

> ### Author Rebuttal · Authors · 2025-07-31
>
> Thank you very much for the helpful comments! We are encouraged that you find our problem “very important in safety critical settings”, and that D3M is “competitive with prior work”. We hope to address your concerns and questions in the following.
>
> # Weaknesses
>
> > “D3M does not trade-off prediction accuracy.” [...]
>
> Certainly! We hereby provide an ablation comparing the D3M base model (D3M) against the same feature extractor architecture coupled with a linear head (Lin), trained side-by-side for UCI Heart Disease, CIFAR-10, and Camelyon17, all else being equal. We report accuracy, F1, AUROC, and Matthew's correlation (for CIFAR-10, binary metrics are computed independently for each class and then averaging the results equally across all classes, without weighting by class frequency):
>
> ### UCI (MLPModel)
>
> | Model | Accuracy      | F1 Score      | AUROC        |  MCC         | Acc. Drop OOD |
> |-------|---------------|---------------|--------------|--------------|--------------|
> | D3M   | 0.76 ± 0.02   | 0.75 ± 0.02   | 0.86 ± 0.01  |0.51±0.04 | -0.11±0.02 |
> | Lin   | 0.77 ± 0.01   | 0.76 ± 0.01   | 0.87 ± 0.01  |0.52±0.02 | _ |
>
> ### CIFAR-10/10.1 (ConvModel)
>
> | Model | Accuracy      | F1 Score      | AUROC        | MCC | Acc. Drop OOD |
> |-------|---------------|---------------|--------------|--------------|--------------|
> | D3M   | 0.70 ± 0.02   | 0.70 ± 0.02   | 0.96 ± 0.00  |0.67±0.02 | -0.13±0.02 |
> | Lin   | 0.72 ± 0.01   | 0.72 ± 0.01   | 0.96 ± 0.00  | 0.69 ± 0.01| _ |
>
> ### Camelyon17 (ResNetModel)
>
> | Model | Accuracy      | F1 Score      | AUROC        | MCC | Acc. Drop OOD |
> |-------|---------------|---------------|--------------|--------------|--------------|
> | D3M   | 0.94 ± 0.01   | 0.94 ± 0.01   | 0.98 ± 0.00  |0.88±0.01 | -0.05±0.00 |
> | Lin   | 0.94 ± 0.00   | 0.94 ± 0.00   | 0.98 ± 0.00  |0.88±0.01 | _ |
>
> We find that indeed, D3M trades off little to no prediction performance. We hypothesize further that this tradeoff is conditional on the generalizability of the representations used before the last layer, as MLPModels and ConvModels are trained from scratch, while the ResNet backbone in the Camelyon17 experiments are finetuned from the ImageNet models, although a lot more experiments are needed to validate this hypothesis.
>
> > “Potential for edge deployment.” [...]
>
> We appreciate your observation regarding our claim about edge deployment. We agree that our discussion in this section was speculative and should have been more clearly framed as such. Our intent was to highlight that D3M’s source-free deployment mechanism offers practical advantages for local inference in e.g. hospital environments, where dedicated compute clusters may be unavailable or undesirable due to privacy and latency concerns. We acknowledge that further empirical study is needed to validate these claims, and we will revise the manuscript to reflect this more cautiously.
>
> > “While VBLL allows more efficient sampling compared to a fully Bayesian network, the price to pay is the lack of diversity of the sampled predictions.” To support this claim, I would expect to see a comparison of the diversity of predictions when using VBLL and a fully Bayesian network. [...]
>
> > How does the use of VBLL compare with other Bayesian approaches?
>
>
> We found that without sampling once more from the logits in order to Monte-Carlo (MC) sample labels with which the base model can disagree with, the maximum disagreement rates achieved across $10000$ runs hover around 5% for both ID and deteriorating OOD samples, making them nearly indistinguishable by D3M. We found that sampling candidate disagreeing predictions from softmaxed and temperature-scaled logits rather than argmaxing allows us to bring the ID disagreement rates in $\Phi$ to around 30%-50%, where deteriorating OOD samples often score 5%-10% higher, making them distinguishable.
>
> Under a rough comparison, for query size 100 and 5000 MC samples, fully Bayesian networks with logit argmaxing naturally achieve ~30% ID disagreement rate on the UCI task and ~37% ID disagreement rate on CIFAR-10. Importantly, while fully Bayesian models can reach comparable disagreement rates even without additional sampling tricks, they do so at significantly greater computational cost: D3M + VBLL trains $\Phi$ in 1 second while fully Bayesian D3M takes 17 minutes for UCI, for instance. In our setting, the goal is to highlight disagreement gaps between ID and deteriorating OOD data—not to maximize disagreement itself. Temperature scaling and softmax sampling already achieve this effectively, making added Bayesian complexity unnecessary.
>
> For a more detailed discussion regarding this difference, please see our response to Reviewer **Vf94**. A complete report with comparisons of fully Bayesian networks to D3M + VBLL across all datasets across independent seeds *is on the way* and will be promptly added to a future revised version of the manuscript.
>
> > In Section 3.2, the authors claim that “D3M is significantly more scalable and practical for real-world deployment [than Detectron]”.  [...]
>
> Certainly! The following is a table listing the FLOPs at different computational stages of the algorithms. We assume a query size of 100, and the same hyperparameters as reported in the Appendix section.
>
> | Algorithm | Dataset | Base Model Training      | Calibration   | **Total** |
>  |-------|-------|---------------|---------------|---------------|
> | D3M   | UCI |  0.23 GFLOPs | 9.00 GFLOPs | **9.23 GFLOPs**|
> | Detectron | UCI | 0.15 GFLOPs | 18.69 GFLOPs | **18.84 GFLOPs** |
> | D3M   | CIFAR-10/10.1 |  1.56 PFLOPs | 0.13 PFLOPs | **1.69 PFLOPs** |
> | Detectron | CIFAR-10/10.1 | 1.56 PFLOPs | >1.95 PFLOPs | **>3.51 PFLOPs** |
> | D3M   | Camelyon17 |  12.02 PFLOPs | 0.74 PFLOPs | **12.76 PFLOPs** |
> | Detectron | Camelyon17 | 12.02 PFLOPs | >11.04 PFLOPs | **>23.06 PFLOPs** |
>
> Eventually, as the feature extractors grow in size, we notice that for D3M the cost is dominated by the training phase, whereas Detectron expends at least nearly the same compute during the calibration phase as its training phase. This is due to Detectron requiring finetuning on held-out subsets while D3M simply freezing the model in evaluation mode and sampling from the VBLL layers. Furthermore, gradient computations of Detectron need to be replicated on the deployment sample.
>
> > I’m quite confused by the Y-axis in Figure 2b. The label says TPR@5%, but this is in a setting with only a non-deteriorating shifts, so the true positive rate is undefined?
>
> Our apologies for this mistake! In figure 2b, the label should be saying FPR@5%. We have adjusted the manuscript accordingly. Indeed, since this is a non-deteriorating shift (following Figure 2a), we want models to resist flagging. Covariate shifts are present as evidenced by their flagging from baseline naive covariate shift detectors.
>
> > The legend in Figure 2 and Figure 3 could be improved. I’m not sure what some of the abbreviations correspond to. For example, is DC the abbreviation for detectron? **
>
> We have revised the manuscript to make the legend correspondences with baseline algorithms clearer. DC here refers to a Domain Classifier (per Rabanser et. al. 2019), i.e. a separate model trained on the same inputs to discern between ID and OOD data (this, of course, requires labeled OOD data). A detailed description of the Domain Classifier will be included in the Appendix.
>
> # Questions
>
> > I’m confused by the columns in Table 1. How can D3M be both deteriorating and non-deteriorating?
>
> Indeed, these two columns refer to whether the algorithms 1. flags deteriorating shifts, and 2. resist flagging non-deteriorating shifts. Importantly, D3M achieves the latter, and provably so under certain theoretical conditions, detailed in Appendix A.
>
>
> > There is a duplicated “operating operating” in the first sentence of the abstract.
>
> Thank you! This mistake is fixed.
>
> > “Underlines in our results indicate that D3M is close to the best baseline.” What is “close” here? It seems somewhat arbitrary given the differences in performance and nearly 10 points in some cases.
>
>  We wanted to use the underlines to highlight situations where most baselines severely underperform while D3M is nearing the performance of the top baseline, though as you highlighted, this is inconsistent with CIFAR-10, query size 20, where D3M gets outperformed by both Detectron and Deep Ensemble at roughly the same performance. For this reason, we have removed the underlines.
>
> > Please, clarify what exactly an N-shot scenario is. Also explain that the {10, 20, 50} in Table 2 correspond to shots in the caption.
>
> Certainly! We appreciate you pointing out the lack of clarity. The N-shot scenarios correspond to the size of the unknown deployment dataset. Remote hospitals with scarcely an influx of patients would be more interested in the 10,20,50-deployment size scenarios, while major hospitals would already be enjoying high TPR from D3M.
>
> Due to this confusion, as well as “few-shot” colloquially referring to single digit shots, we have henceforth decided to adopt the terminology of “query size” to refer to the quantity above, and will edit the manuscript accordingly as to further avoid misunderstandings.
>
> # Closing Comments:
>
> Ablations reported in this rebuttal as well as other rebuttals will be published in a revised version of the manuscript.
>
> Thank you once again for your helpful feedback. We hope that our rebuttal addresses your questions and concerns, and we kindly ask that you consider a fresher evaluation of our paper if you are satisfied with our responses. We are also more than happy to answer any further questions that arise.

---

> > ### Comment · Reviewer_TFQc · 2025-08-04
> >
> > Thank you to the authors for responding to my questions and concerns.
> >
> > When integrating new results into the paper, I recommend adding clear quantification of their claims in the text.
> > For example, turn statements like "significantly more scalable" into "up to 10x FLOP reduction during training and 2x overall."
> >
> > Given that these new results will be integrated into the paper, I've raised my score.

---

> > > ### Author Response · Authors · 2025-08-06
> > > **Response to Reviewer TFQc**
> > >
> > > Thank you once again for your time and effort put into the reviewing of our submission and rebuttal, we greatly appreciate it! We hope to address your final comments and suggestions in the below.
> > >
> > > > When integrating new results into the paper, I recommend adding clear quantification of their claims in the text. For example, turn statements like "significantly more scalable" into "up to 10x FLOP reduction during training and 2x overall."
> > >
> > > Thank you! We will clearly quantify the scalability mentioned in our work where relevant as well as add the FLOPS in a separate appendix section dedicated to all ablations requested by all reviewers of this work to support our claims.
> > >
> > > > Given that these new results will be integrated into the paper, I've raised my score.
> > >
> > > Once again, thank you for your thorough feedback and for taking the time to reconsider your evaluation. We truly appreciate your support and are glad our clarifications were helpful!

---

### Official Review · Reviewer_ktyz · 2025-07-03

**Clarity:** 4
**Significance:** 3
**Originality:** 3
**Rating:** 5
**Confidence:** 4

**Summary:**

The paper presents a methodology to address post-deployment deterioration of model performance due to domain shift in the data. The authors propose a train, calibrate, and deploy strategy where, with the base model, there is a variational bayesian last layer (VBLL) to model the posterior predictive distribution over the target classes. The VBLL layer helps to identify deployment time deterioration of the model based on the disagreement rate over K drawn samples from the variational posteriors. The method is evaluated on publicly available datasets (i.e., vision, healthcare) and shows comparable performance with the baselines in detecting domain shifts in data.

**Questions:**

1) How does the VBLL layer impact classification accuracy?
2) How can the VBLL layer be extended to a regression problem?

**Ethical Concerns:**

["NO or VERY MINOR ethics concerns only"]

**Final Justification:**

Yes, please incorporate k-fold cross-validation results on the dataset in the appendix of the final draft to showcase the effectiveness of the framework.

“Functionally, this means that patients’ static covariates as well as their labs, medications, and medical interventions have drifted over time, especially into the COVID years.”  – Please clearly mention this in the paper. In this version, it reads like “Monitoring results on artificially shifted test data from the IM dataset”.

The ablation on VBLL and the Linear classification layer is good. Please include it in the appendix. Also, I find the ablation on the temperature parameter intuitive.


I would suggest including a short discussion on how to extend VBLL to a regression model in the appendix (as a future direction).

The rebuttal addresses most of my concerns with this paper. I am increasing my score to 5 (accept). Thanks!

**Limitations:**

1) Evaluation can be more thorough. Please select a new temporal dataset for evaluating the method
2) Need to proofread the manuscript

**Paper Formatting Concerns:**

No paper formatting issue.

**Quality:**

3

**Strengths And Weaknesses:**

Strengths:
1) The codebase is available, structured well, and documented for review.
2) The introduction of the paper is well placed and motivates the approach.
3) The algorithm is clearly explained with appropriate figures and descriptions
4) Section 2.3: Algorithmic insights is well explained
5) Experiments are conducted on publicly available datasets
6) The related work is properly structured.

Weakness:
1) The abstract has two “operating”. Needs to proofread the manuscript.
2) Evaluation on the UCI Heart Disease, CIFAR 10.1, and Camelyon 17 datasets can be more thorough with holdout participants or a dataset subset.
3) Temporal shifts are only shown for the UCI IM dataset with an artificial shift in the test data (i.e., it’s not evident in the original data). This compromises the motivation of the problem. I strongly suggest evaluating on another temporal dataset with an inherent temporal shift.
4) The impact on the target classification due to the VBLL layer is not clearly discussed
5) Please provide an ablation study on the parameters: temperature and alpha.

---

> ### Author Rebuttal · Authors · 2025-07-31
>
> Thank you very much for the helpful comments! We feel encouraged that you found our codebase as well as the introduction and the related work well structured! We are glad to hear that you found our algorithm clearly explained! In the following, we hope to address your questions and concerns.
>
> # Weaknesses
>
> > The abstract has two “operating”
>
> Thank you! We edited the manuscript accordingly.
>
> > Evaluation on the UCI Heart Disease, CIFAR 10.1, and Camelyon 17 datasets can be more thorough with holdout participants or a dataset subset.
>
> For all experiments, we construct D3M’s Phi training and ID splits by randomly splitting the original validation set. In each independent iterate of the experiment, random participants in the ID dataset as well as in the OOD dataset are selected from which run statistics contribute to the reported figures in the manuscript, thus each independent run “holds out” both ID and OOD participants. Could you perhaps mean a k-fold validation by splitting ID and OOD data into bundles and cross-validating TPRs? We are happy to run further experimentation to showcase the effectiveness of the framework if you could please elaborate on your suggestions on evaluation!
>
> > Temporal shifts are only shown for the UCI IM dataset with an artificial shift in the test data (i.e., it’s not evident in the original data). This compromises the motivation of the problem. I strongly suggest evaluating on another temporal dataset with an inherent temporal shift.
>
> Our apologies for the inadequate clarity in the experiments’ descriptions. The IM dataset (name withheld) is a private electronic health records (EHR) dataset from a network of hospitals, comprising of static and dynamic features for patients’ hospital visits. This dataset is temporal by nature as data splits are arranged per half-years starting from 2017, where shifts in patient features become even more pronounced during the COVID period of 2020-2021. For the predictive task of 14-day mortality, our D3M base models trained on data before 2018 surprisingly do not underperform on later splits (Fig. 2a), therefore we make the assessment that this shift is non-deteriorating.
>
> Furthermore, we also claim that an important feature of this dataset is its inherent temporal covariate shift, the evidence of which is in the elevated FPR of divergence/distance-based detectors of Deep Kernel MMD (MMD-D) and Relative Mahalanobis Distance (Rel-MH). Functionally, this means that patients’ static covariates as well as their labs, medications, and medical interventions have drifted over time, especially into the COVID years.
>
> > The impact on the target classification due to the VBLL layer is not clearly discussed
>
> > How does the VBLL layer impact classification accuracy?
>
>
> Thank you for pointing this out. We hereby provide an ablation comparing the D3M base model (D3M) against the same feature extractor architecture coupled with a linear head (Lin), trained side-by-side for UCI Heart Disease, CIFAR-10, and Camelyon17, all else being equal. We report accuracy, F1, AUROC, and Matthew's correlation (for CIFAR-10, binary metrics are computed independently for each class and then averaging the results equally across all classes, without weighting by class frequency):
>
> ### UCI (MLPModel)
>
> | Model | Accuracy      | F1 Score      | AUROC        |  MCC         |
> |-------|---------------|---------------|--------------|--------------|
> | D3M   | 0.76 ± 0.02   | 0.75 ± 0.02   | 0.86 ± 0.01  |0.51±0.04 |
> | Lin   | 0.77 ± 0.01   | 0.76 ± 0.01   | 0.87 ± 0.01  |0.52±0.02 |
>
> ### CIFAR-10/10.1 (ConvModel)
>
> | Model | Accuracy      | F1 Score      | AUROC        | MCC |
> |-------|---------------|---------------|--------------|--------------|
> | D3M   | 0.70 ± 0.02   | 0.70 ± 0.02   | 0.96 ± 0.00  |0.67±0.02 |
> | Lin   | 0.72 ± 0.01   | 0.72 ± 0.01   | 0.96 ± 0.00  | 0.69 ± 0.01|
>
> ### Camelyon17 (ResNetModel)
>
> | Model | Accuracy      | F1 Score      | AUROC        | MCC |
> |-------|---------------|---------------|--------------|--------------|
> | D3M   | 0.94 ± 0.01   | 0.94 ± 0.01   | 0.98 ± 0.00  |0.88±0.01 |
> | Lin   | 0.94 ± 0.00   | 0.94 ± 0.00   | 0.98 ± 0.00  |0.88±0.01 |
>
> We notice that side by side, $FE_\theta + VBLL_\theta$ is slightly worse than its linear head counterpart. We argue, however, that the sampling benefits provided by the VBLL layer enables the functioning of D3M. When compared to multiple heads (Deep Ensemble baseline) for deteriorating OOD detection, our method achieves noticeably higher few-shot TPR.
>
> > Please provide an ablation study on the parameters: temperature and alpha.
>
> Certainly! The following table is the result of an ablation on the temperature parameter at query size $100$.
>
> ### UCI
> | temp | fpr_id | tpr  | dis_rate_id | dis_rate_ood |
> |------|--------|------|------------------|-------------------|
> | 1    | 0.01   | 0.85 | 0.4359           | 0.4919            |
> | 2    | 0.07   | 0.49 | 0.5478           | 0.5785            |
> | 3    | 0.07   | 0.32 | 0.5864           | 0.6089            |
> | 5    | 0.09   | 0.12 | 0.6280           | 0.6381            |
> | 10   | 0.08   | 0.19 | 0.6520           | 0.6615            |
> | 20   | 0.04   | 0.11 | 0.6679           | 0.6721            |
>
> ### CIFAR-10/10.1
> | temp | fpr_id | tpr  | dis_rate_id | dis_rate_ood |
> |------|--------|------|------------------|-------------------|
> | 1    | 0.14   | 0.98 | 0.4309           | 0.5189            |
> | 2    | 0.06   | 1.00 | 0.6405           | 0.7267            |
> | 3    | 0.06   | 0.96 | 0.7683           | 0.8289            |
> | 5    | 0.05   | 0.78 | 0.8837           | 0.9136            |
> | 10   | 0.05   | 0.34 | 0.9505           | 0.9619            |
> | 20   | 0.07   | 0.09 | 0.9744           | 0.9762            |
>
> ### Camelyon17
> | temp | fpr_id | tpr  | dis_rate_id | dis_rate_ood |
> |------|--------|------|------------------|-------------------|
> | 1    | 0.09   | 0.99 | 0.1696           | 0.2561            |
> | 2    | 0.12   | 1.00 | 0.2891           | 0.3904            |
> | 3    | 0.09   | 1.00 | 0.3801           | 0.4712            |
> | 5    | 0.07   | 1.00 | 0.4774           | 0.5471            |
> | 10   | 0.08   | 0.61 | 0.5803           | 0.6118            |
> | 20   | 0.05   | 0.32 | 0.6295           | 0.6483            |
>
>
> By increasing the temperature, samples are more diverse and disagreements are more pronounced, evidenced by the elevated mean disagreement rates. This not only hurts OOD-TPR but also increases FPRs, as the “base ID disagreement rate” distribution is further skewed toward 1, making it harder to differentiate between ID disagreement rates and deteriorating OOD disagreement rates. The correct temperature balancing sampling diversity with reasonable ID disagreement rates (and thus ID disagreement thresholds) is thus a hyperparameter that we tune. In general, values between 1 and 5 are preferred.
>
> We respectfully note that $\alpha$ is a user-defined significance level, akin to a tolerance in classical hypothesis testing. Thus, varying $\alpha$ does not meaningfully reflect the behavior or robustness of the algorithm itself, but rather tunes the system’s operating point: a higher $\alpha$ increases sensitivity but also false positives, while a lower $\alpha$ increases specificity. As such, varying $\alpha$ does not reflect the robustness of the method, but rather the practitioner’s risk tolerance — similar to choosing a 95% vs. 99% confidence level depending on deployment needs.
> We provide results at a reasonable default $\alpha=0.1$ and practitioners may adjust it depending on the criticality of their deployment context.
>
> # Questions
> > How does the VBLL layer impact classification accuracy?
>
> Please see above.
>
> > How can the VBLL layer be extended to a regression problem?
>
> We believe several works in the literature replace the categorical likelihood with a Gaussian likelihood, i.e. the VBLL layer would output a predictive distribution over real-valued targets—typically by learning a distribution over the regression weights and using Bayesian linear outputs to produce a mean and variance for the predicted value. The ELBO objective can then be adapted to use a Gaussian likelihood while retaining variational inference for the weights, allowing the model to capture epistemic uncertainty in regression.
>
> In terms of deteriorating OOD detection for regression, we believe D3M has nice analogies with already-existing confidence region methods. We believe “the extent of disagreement” in classification is thus analogous to a $(1-\alpha)%$ confidence region following a posterior distribution in Bayesian regression, perhaps reducing D3M to a sampling-based approach to OOD detection for regression.
>
> # Closing Comments:
>
> Ablations reported in this rebuttal as well as other rebuttals will be published in a revised version of the manuscript.
>
> Thank you once again for your valuable feedback. We hope that our rebuttal addresses your questions and concerns, and we kindly ask that you consider a fresher evaluation of our paper if you are satisfied with our responses. We are also more than happy to answer any further questions that arise.

---

> > ### Comment · Reviewer_ktyz · 2025-08-05
> > **The rebuttal addresses my concerns with the paper. Please incorporate the new changes in the final draft**
> >
> > Yes, please incorporate k-fold cross-validation results on the dataset in the appendix of the final draft to showcase the effectiveness of the framework.
> >
> > “Functionally, this means that patients’ static covariates as well as their labs, medications, and medical interventions have drifted over time, especially into the COVID years.”  – Please clearly mention this in the paper. In this version, it reads like “Monitoring results on artificially shifted test data from the IM dataset”.
> >
> > The ablation on VBLL and the Linear classification layer is good. Please include it in the appendix. Also, I find the ablation on the temperature parameter intuitive.
> >
> >
> > I would suggest including a short discussion on how to extend VBLL to a regression model in the appendix (as a future direction).
> >
> > The rebuttal addresses most of my concerns with this paper. I am increasing my score to 5 (accept). Thanks!

---

> > > ### Author Response · Authors · 2025-08-06
> > > **Response to Reviewer ktyz**
> > >
> > > Thank you once again for your time and effort put into the reviewing of our submission, we greatly appreciate it! We hope to address your final comments and suggestions in the below.
> > >
> > > > Yes, please incorporate k-fold cross-validation results on the dataset in the appendix of the final draft to showcase the effectiveness of the framework.
> > >
> > > Thank you! We will append the k-fold results (as described in the rebuttal) to the appendix of the final draft to showcase the effectiveness of our framework.
> > >
> > > > “Functionally, this means that patients’ static covariates as well as their labs, medications, and medical interventions have drifted over time, especially into the COVID years.” – Please clearly mention this in the paper. In this version, it reads like “Monitoring results on artificially shifted test data from the IM dataset”.
> > >
> > > Thank you for highlighting this. In the final version, we will clearly mention in the main paper concrete evidence of the patients' static and dynamic covariates naturally shifting over time, especially into the COVID years, as per our discussion above. Further, all anonymized details about the IM dataset will be published as well per the dataset's terms of usage agreements.
> > >
> > > > The ablation on VBLL and the Linear classification layer is good. Please include it in the appendix. Also, I find the ablation on the temperature parameter intuitive.
> > >
> > > Thank you! A new appendix section shall be dedicated entirely to all ablations requested by all reviewers of this work which will comprise results, analyses, and discussions of their implications and limitations to our framework.
> > >
> > > > I would suggest including a short discussion on how to extend VBLL to a regression model in the appendix (as a future direction).
> > >
> > > Thank you once again for raising this insightful point in your review. We are thinking of how to completely concretize the regression version as this could be related to some prior work on uncertainty quantification and uncertainty quantification-based monitor designs for Bayesian regression. In the final draft, an appendix subsection (Appendix A.X) shall be dedicated to discussing this.
> > >
> > > > The rebuttal addresses most of my concerns with this paper. I am increasing my score to 5 (accept). Thanks!
> > >
> > > Thank you so much for your thoughtful feedback and for taking the time to reconsider your evaluation of our work! We truly appreciate your support and are glad our clarifications were helpful!

---

### Official Review · Reviewer_vf94 · 2025-07-07

**Clarity:** 3
**Significance:** 3
**Originality:** 3
**Rating:** 4
**Confidence:** 3

**Summary:**

This paper tackles the challenge of knowing when a machine learning model’s performance is getting worse after deployment especially in real-world settings where data changes over time and labels are not available. The authors introduce D3M, a method that checks for model disagreement to detect harmful changes in data. Unlike typical drift detectors, D3M aims to avoid false alarms when the model is still performing well and to reliably catch cases where performance is truly dropping. The method is supported by theoretical guarantees and shows strong results on both standard benchmarks and a large internal medicine dataset, making it promising for use in critical applications.

**Questions:**

1. Line 1: operating repeated twice
2. Can the authors provide the definitions of deteriorating and non-deteriorating shifts directly in the introduction?
3. Can the authors explain as to why measuring disagreement via a variational approach that too only for the last layer is a useful method for measuring disagreement? In my opinion, wouldn't there be limited diversity or limited choice of hypothesis class exploration if only the last layer is trained in that fashion?
4. How do the results change (qualitative explanation is sufficient) if there are multiple classification heads instead of the variational training?
5. After T rounds, can you provide a graph of how the base model (ID) threshold changes? Is it more or less very similar to each other?
6. In addition to metrics such as TPR, FPR, can the authors also evaluate a unifying metric such as Matthew's correlation coefficient in order to holistically characterize performance?
7. Can the authors also qualitatively compare their work to conformal prediction and risk control? Conformal prediction can also be an easy to adopt strategy to explain model failures and even identify distribution shifts.

**Ethical Concerns:**

["NO or VERY MINOR ethics concerns only"]

**Final Justification:**

The authors have sufficiently addressed my concerns and I feel their results are very comprehensive.

**Limitations:**

Yes

**Quality:**

3

**Strengths And Weaknesses:**

Strengths - The writing is very clear and the paper is well-structured. The paper offers theoretical justification and broad empirical evaluation.

Weaknesses - See questions

---

> ### Author Rebuttal · Authors · 2025-07-31
>
> Thank you so much for the helpful comments! We are encouraged that you find our writing clear and the paper well-structured. We hope to address your questions and concerns in the following.
>
> # Questions
>
> > Line 1: operating repeated twice
>
> Thank you for pointing this out! We’ve removed the repeated word and proof-read carefully.
>
> > Can the authors provide the definitions of deteriorating and non-deteriorating shifts directly in the introduction?
>
> We will qualitatively introduce post-deployment deterioration prior to discussing the key challenges with its monitoring, thank you for the suggestion!
>
>
> > Can the authors explain as to why measuring disagreement via a variational approach that too only for the last layer is a useful method for measuring disagreement? In my opinion, wouldn't there be limited diversity or limited choice of hypothesis class exploration if only the last layer is trained in that fashion?
>
> In theory, the diversity should be severely limited. The ultimate key to the method is in fact a two-stage sampling. Each Monte Carlo (MC) realization corresponds to a set of logits sampled from the variational posterior.
>
> From here, one may argmax the logits to get labels for each MC realization, for each sample in our batch. In our testing, we found that even with up to $10,000$ MC realizations, the sampled logits are not diverse enough to the extent that their argmaxes largely are the same. Disagreement rates hover at less than 5% on both ID and OOD batches, making them practically indistinguishable.
>
> Thus, we sample *once again* from softmaxed, temperature-weighted logits of each MC realization in order to get more variability, allowing us to achieve 30%-50% ID disagreement rates across various experiments. Of course, all operations here are parallelized, making the computational costs acceptable, even comfortable.
>
> With this in mind, consider now a fully Bayesian implementation. Indeed, even without the two-stage sampling above, we achieve acceptable ID disagreement rates. However, this comes at the cost of heavy computational overhead. One either reparametrizes layers, making each forward pass’s output essentially a sample of logits, requiring sequential forward passes to collect MC logits, or one must lift all weight tensors to include a sample dimension and ensure every downstream operation — including residual connections, batch norm, and other architectural components — is broadcast-compatible.
>
> Importantly, while this is realizable, it offers no benefits to our two-stage sampling strategy. We find that when temperatures are tuned to yield ID disagreement rates in the 30% to 50% range, deteriorating OOD disagreement rates are easily discernable as on those inputs, D3M tends to disagree 5%-10% better, thus easily achieving high TPR.
>
> In summary, the limited choice of hypothesis classes is adequately rectified by the double-sampling described above and in the manuscript, which we found to be empirically sufficient for the purposes of D3M’s 2nd and 3rd stages.
>
> > How do the results change (qualitative explanation is sufficient) if there are multiple classification heads instead of the variational training?
>
> This is the architecture and design philosophy of Deep Ensemble adapted to monitoring model deterioration. We report in Table 2 that this monitor underperforms compared to other baseline methods as well as D3M. While leveraging head disagreements is a sensible insight, we believe D3M is more appropriate when unlabeled test data is scarce.
>
> > After T rounds, can you provide a graph of how the base model (ID) threshold changes? Is it more or less very similar to each other?
>
> We report the ID thresholds for $\alpha=0.1$ across $10$ independent runs. Columns are the rounds $t \leq T=1000$ from which the $1-\alpha$ quantile is computed for $\Phi$.
>
> | Dataset   | $t=5$          | $t=50 $         | $t=100 $        | $t=500$          | $t=T=1000 $        |
> |-----------|-------------|-------------|-------------|--------------|--------------|
> | UCI Heart Disease       | 0.457±0.011 | 0.468±0.007 | 0.467±0.004 | 0.468±0.004  | 0.470±0.000  |
> | CIFAR-10/10.1   | 0.452±0.013 | 0.464±0.009 | 0.463±0.008 | 0.465±0.005  | 0.463±0.005  |
> | Camelyon17  | 0.306±0.011 | 0.312±0.004 | 0.312±0.004 | 0.310±0.000 | 0.310±0.000 |
>
> The above shows that although some variability is exhibited for early $t \in T$, the threshold stabilizes with more independent rounds, as evidenced by the decrease in standard deviation. Importantly, even as early as 5 samples, the ID threshold is already stabilized.
>
> > In addition to metrics such as TPR, FPR, can the authors also evaluate a unifying metric such as Matthew's correlation coefficient in order to holistically characterize performance?
>
> For tasks with known OOD deterioration we desire monitors to identically output 1 (positive flag). Therefore, the most important metric by which we compare detectors is the TPR. One may choose to view detection as a meta-learning problem wherein the sole input to the detector is the task, and the metric by which to assess the detector is how reliably it identifies OOD deterioration. The converse reasoning is true for FPR.
>
> Regarding the base classifier’s performance however, we report their accuracy, F1, AUROC, Matthew’s Correlation, as well as the accuracy drop when testing on deteriorating OOD held-out data below. We additionally provide classification performance comparisons with a Linear head instead of a VBLL layer, all else being equal, trained side to side.
>
> ### UCI (MLPModel)
>
> | Model | Accuracy      | F1 Score      | AUROC        |  MCC         | Acc. Drop OOD |
> |-------|---------------|---------------|--------------|--------------|--------------|
> | D3M   | 0.76 ± 0.02   | 0.75 ± 0.02   | 0.86 ± 0.01  |0.51±0.04 | -0.11±0.02 |
> | Lin   | 0.77 ± 0.01   | 0.76 ± 0.01   | 0.87 ± 0.01  |0.52±0.02 | _ |
>
> ### CIFAR-10/10.1 (ConvModel)
>
> | Model | Accuracy      | F1 Score      | AUROC        | MCC | Acc. Drop OOD |
> |-------|---------------|---------------|--------------|--------------|--------------|
> | D3M   | 0.70 ± 0.02   | 0.70 ± 0.02   | 0.96 ± 0.00  |0.67±0.02 | -0.13±0.02 |
> | Lin   | 0.72 ± 0.01   | 0.72 ± 0.01   | 0.96 ± 0.00  | 0.69 ± 0.01| _ |
>
> ### Camelyon17 (ResNetModel)
>
> | Model | Accuracy      | F1 Score      | AUROC        | MCC | Acc. Drop OOD |
> |-------|---------------|---------------|--------------|--------------|--------------|
> | D3M   | 0.94 ± 0.01   | 0.94 ± 0.01   | 0.98 ± 0.00  |0.88±0.01 | -0.05±0.00 |
> | Lin   | 0.94 ± 0.00   | 0.94 ± 0.00   | 0.98 ± 0.00  |0.88±0.01 | _ |
>
> The drops in OOD accuracy is symptomatic of model deterioration.
>
> > Can the authors also qualitatively compare their work to conformal prediction and risk control? Conformal prediction can also be an easy to adopt strategy to explain model failures and even identify distribution shifts.
>
> Thank you for this suggestion! Conformal prediction (CP) methods typically rely on a held-out calibration set with labeled data to construct valid predictive intervals, making them challenging to apply in the fully unsupervised post-deployment setting we consider. However, in a lot of applied scenarios, labels do eventually *become* available either via explicit labeling or one of several outcomes realized within a said time period. The deployment-label free advantage of D3M wears off over time in a practical setting when compared to CP in this regard.
>
> We also highlight that D3M is uniquely robust to non-deteriorating shifts that may still be flagged by CP methods due to covariate changes. While CP methods can signal increased uncertainty, they do not directly distinguish between the harmfulness of shifts or offer the same guarantees on FPRs.  We believe that the scope and assumptions of D3M differ from those of standard CP and risk control approaches.
>
> # Closing remarks
> The manuscript will be updated with all ablation results presented in this rebuttal.
>
> Thank you once again for your helpful feedback. We hope that our rebuttal addresses your questions and concerns, and we kindly ask that you consider a fresher evaluation of our paper if you are satisfied with our responses. We are also more than happy to answer any further questions that arise.

---

### Note · Authors · 2025-08-13

We thank all Reviewers vf94, ktyz, TFQc, and 88gc for their **deep involvement** in our work. Their reviews have resulted in a significantly refined submission. Their inputs have strengthened the narrative of the work and we have the ablations to justify the efficacy of our proposed framework. We highlight the core changes made to our manuscript following the discussion period.

# 1. Restructuring the Narrative in the Main Body

Per Reviewer 88gc, we included a subsection with Lemmas relating D3M to its idealized version, justifying how D3M approximately tracks the Disagreement-based Post-Deployment Deterioration (D-PDD) criterion. In particular, we refined the narrative to reflect that in idealized theoretical scenarios, the algorithm provably works, while the D3M approximation scales and performs well in practice from oversampling the desired hypotheses set.

# 2. Ablations in the Appendix

Reviewer-recommended ablations are compiled in the Appendix with detailed discussions of their importance and support for main claims:

1. D3M's VBLL vs. Linear Head (vf94, ktyz, TFQc): Experiments across all open datasets comparing accuracy, F1, AUROC, and Matthew's correlation confirm minimal performance drops when implementing VBLL over linear heads.

2. Computational Costs (TFQc): FLOP comparisons show D3M achieves up to 10x training reduction and 2x overall reduction versus Detectron , requiring no training data once the base model completes training.

3. Temperature and ID Threshold Stability (ktyz, vf94): Ablations on temperature parameter (ktyz) and ID disagreement threshold stability across rounds (vf94) demonstrate that higher temperatures reduce TPR by narrowing the ID-OOD disagreement gap. Optimal temperature is task-dependent and requires careful tuning.

4. D3M's VBLL vs. Fully Bayesian (vf94, TFQc): Results on UCI and CIFAR-10 show fully Bayesian networks provide no TPR/FPR advantage over VBLL for OOD detection, while requiring orders of magnitude more wall-clock time. D3M's double sampling adequately addresses limited hypothesis classes for the (2) Calibration and (3) Deployment stages.

5. D3M vs. Idealized/Theoretical D3M (88gc): Accuracy plots of max disagreement rate decision boundaries versus validation accuracy confirm D3M oversamples beyond the desired hypothesis class. We detail how this empirical-theoretical discrepancy could increase false negatives through higher $\Phi$ dispersion without increasing false positives.

---

### Decision · Program_Chairs · 2025-09-17

**Decision:**

Accept (poster)

**Comment:**

This paper proposes D3M, a method to detect post-deployment deterioration due to data distribution shift. It adds a variational bayesian last layer and measures disagreement among output samples. This work tackles an important safety-critical scenario, does not require training labels, and is able to detect deteriorating shifts from non-deteriorating shifts in contrast to standard OOD settings. It provides theoretical analysis, and compares the practical and idealized version empirically. Experiments on public datasets and a large-scale medical dataset demonstrate its efficacy and efficiency.

Reviewers were critical of multiple aspects, including the lack of theoretical foundation, unsupported claims and missing ablations. The rebuttal with additional theoretical justification and experiments addressed almost all the concerns. As a result, all reviewers agree on the acceptance of this submission.

Some concerns remain, though they are not considered rejection-worthy:
- Not significant scalability improvement in absolute term compared to the baseline (though the relative reduction of FLOPs is large)
- Strong assumptions in the theoretical justification, admitted by authors